# Concerted measurements of free amino acids at the Cape Verde Islands: High enrichments in submicron sea spray aerosol particles and cloud droplets

Nadja Triesch[1], Manuela van Pinxteren[1], Anja Engel[2] and Hartmut Herrmann[1]

[1]Leibniz-Institute for Tropospheric Research (TROPOS), Atmospheric Chemistry Department (ACD), Leipzig 04318, Germany
[2]GEOMAR Helmholtz Centre for Ocean Research, Kiel 24105, Germany

*Correspondence to*: Hartmut Herrmann (herrmann@tropos.de)

**Abstract.**

Measurements of free amino acids (FAA) in the marine environment to elucidate their transfer from the ocean into the atmosphere to marine aerosol particles and to clouds were performed at the MarParCloud campaign at the Cape Verde islands in autumn 2017. According to physical and chemical specifications such as the behaviour of air masses, particulate MSA

concentrations and MSA/sulfate ratios as well as particulate mass concentrations of dust tracers, aerosol particles predominantly of marine origin with low to medium dust influences were observed. FAA were investigated in different compartments: they were examined in two types of seawater -underlying water (ULW) and in the sea surface microlayer (SML)- as well as in ambient marine size-segregated aerosol particle samples at two heights (ground based at the Cape Verde Atmospheric Observatory (CVAO) and at 744 m height at the Mt. Verde) and in cloud water using concerted measurements.

The $\sum$FAA concentration in the SML varied between 0.13-3.64 µmol L$^{-1}$, whereas it was between 0.01-1.10 µmol L$^{-1}$ in the ULW; also, a strong enrichment of $\sum$FAA (EF$_{SML}$: 1.1-298.4, average of 57.2) was found in the SML. In the submicron (0.05-1.2 µm) aerosol particles at the CVAO, the composition of FAAs was more complex and higher atmospheric concentrations of $\sum$FAA (up to 6.3 ng m$^{-3}$) compared to the supermicron (1.2-10 µm) aerosol particles (maxima of 0.5 ng m$^{-3}$) were observed. The total $\sum$FAA concentration (PM$_{10}$) was between 1.8-6.8 ng m$^{-3}$ and tended to increase during the campaign. Averaged

$\sum$FAA concentrations on the aerosol particles at the Mt. Verde were lower (submicron: 1.5 ng m$^{-3}$, supermicron: 1.2 ng m$^{-3}$) compared to the CVAO. A similar percentage contribution of $\sum$FAA to dissolved organic carbon (DOC) in the seawater (up to 7.6 %) and to water-soluble organic carbon (WSOC) on the submicron aerosol particles (up to 5.3 %) indicated a related transfer process of FAA and DOC in the marine environment.

Considering solely ocean-atmosphere transfer and neglecting atmospheric processing, high FAA enrichment factors were

found in both aerosol particles in the submicron range (EF$_{aer(\sum FAA)}$: 2·10$^3$-6·10$^3$) and medium enrichment factors in the supermicron range (EF$_{aer(\sum FAA)}$: 1·10$^1$-3·10$^1$). In addition, indications for a biogenic FAA formation were observed. One striking finding was furthermore the high and varying FAA cloud water concentration (11.2-489.9 ng m$^{-3}$) as well as enrichments

(EF$_{CW}$: $4 \cdot 10^3$ and $1 \cdot 10^4$ compared to the SML and ULW, respectively), which were reported here for the first time. The abundance of inorganic marine tracers (sodium, methane-sulfonic acid) in cloud water suggests an influence of oceanic sources on marine clouds. Finally, the varying composition of the FAA in the different matrices shows that their abundance and ocean-atmosphere transfer are influenced by additional biotic and abiotic formation and degradation processes. Simple physico-chemical parameters (e.g. surface activity) are not sufficient to describe the concentration and enrichments of the FAA in the marine environment. For a precise representation in organic matter (OM) transfer models, further studies are needed to unravel their drivers and understand their composition.

**Keywords**

amino acids, organic matter, seawater, sea surface microlayer, size-segregated aerosol particles, cloud water, transfer, enrichment factor, Cape Verde Atmospheric Observatory (CVAO)

## 1. Introduction

Amino acids contribute to the global nitrogen and carbon cycle and to the atmosphere-biosphere nutrient cycle (Zhang and Anastasio, 2003;Wedyan and Preston, 2008). They can be divided into free single amino acids (FAA) and combined amino acids (CAA), which include proteins, peptides or other combined forms (Mandalakis et al., 2011). Amino acids are produced in the ocean and are reported to be in the upper layer of the ocean, the sea surface microlayer (SML) (Kuznetsova et al., 2004;Reinthaler et al., 2008;van Pinxteren et al., 2012;Engel and Galgani, 2016). The SML, as the direct interface between the ocean and the atmosphere, may play an important role as a source of organic matter (OM) in aerosol particles within the marine environment (Cunliffe et al., 2013;Engel et al., 2017;Wurl et al., 2017). Specific organic groups of compounds, including nitrogenous OM (Engel and Galgani, 2016) can be strongly enriched in the SML. From the ocean, amino acids as part of the class of proteinaceous compounds can be transferred into atmospheric particles via bubble bursting (Kuznetsova et al., 2005;Rastelli et al., 2017). These proteinaceous compounds are often analyzed as sum parameter 'proteins' using an analytical staining method with Coomassie blue developed by Bradford (1976) and often applied in previous studies (Gutiérrez-Castillo et al., 2005;Mandalakis et al., 2011;Rastelli et al., 2017). Despite their attribution to proteins the FAAs are better utilizable forms of nitrogen instead of proteins for an aquatic organism such as phytoplankton and bacteria (Antia et al., 1991;McGregor and Anastasio, 2001).

Due to their structure and hygroscopic properties, amino acids can act as both ice-forming particles (INP) (Wolber and Warren, 1989;Szyrmer and Zawadzki, 1997;Pandey et al., 2016;Kanji et al., 2017) as well as cloud condensation nuclei (CCN) (Kristensson et al., 2010) in the atmosphere when amino acids such as arginine and asparagine can exist as metastable droplets instead of solid particles at low relative humidity; this showed a laboratory study (Chan et al., 2005). In general, previous studies have shown that amino acids in aerosol particles can have both natural and anthropogenic sources. Having being detected in volcanic emissions (Scalabrin et al., 2012) and during biomass burning events (Chan et al., 2005;Feltracco et al.,

2019), amino acids can be produced by plants, pollens, fungi, bacterial spores and algae (Milne and Zika, 1993;Zhang and Anastasio, 2003;Matos et al., 2016). Nevertheless, they are useful indicators for aerosol particle age and origin (Barbaro et al., 2011;Matsumoto and Uematsu, 2005;Scalabrin et al., 2012). Based on a cluster and factor analysis, Scalabrin et al. (2012) suggested two possible sources for the amino acids in the ultrafine Arctic aerosol particles. First, the authors mentioned the

regional development (isoleucine, leucine, threonine) and long-range transport (glycine) of amino acids from marine areas; secondly, the influence of local sources such as of marine primary production (proline, valine, serine, tyrosine, glutamic acid). A different approach of Mashayekhy Rad et al. (2019) investigated the atmospheric proteinogenic aerosol particles in the Arctic and attributed them to different sources based among others on the reactivity of the distinct amino acids. The authors differentiated here between long-range transport (glycine), terrestrial and marine aerosol particles (proline, valine, serine,

tyrosine) and coastal and marine phytoplankton and bacteria (isoleucine, leucine and threonine) as important sources for amino acids (Mashayekhy Rad et al., 2019). In fact, previous studies have assigned individual amino acids to specific marine biogenic sources and used them as biomarkers. Hammer and Kattner (1986) reported correlations between aspartic acid, diatoms and zooplankton in seawater. GABA (γ-aminobutyric acid) was referred to as an indicator for the microbiological decomposition of OM (Dauwe et al., 1999;Engel et al., 2018) and is used as a microbiological proxy in aerosol particles. To facilitate the

comparison of amino acids in different studies, one possibility is to group them as regards their physio-chemical properties of amino acids ('hydropathy index' (Kyte and Doolittle, 1982)) as Pommié et al. (2004) suggested based on the partition coefficient between water and ethanol. This divides them into hydrophilic, neutral and hydrophobic amino acids as discussed in Barbaro et al. (2015) for FAA in Antarctic aerosol particles. They also observed that hydrophilic FAA in the Antarctic were predominant in locally produced marine aerosol particles, while hydrophobic amino acids prevailed in aerosol particles

collected at the continental station.

Although the study and characterization of amino acids are of paramount importance for atmospheric scientists, the true role and the fate of amino acids in the atmosphere are still poorly understood (Matos et al., 2016). Despite several studies of FAAs also conducted in the marine environment, there is still a huge uncertainty to the question whether FAAs are of marine origin or not. Matsumoto and Uematsu (2005) showed that the long-range transport of land-derived sources largely contributes to the

amino acid concentration in the North Pacific. On the other hand, based on a positive correlation between amino acids in seawater and the atmosphere, Wedyan and Preston (2008) pointed out the particulate amino acids in the Southern Ocean to be of marine origin. These findings are likely due to regional varying source strengths, given different meteorological and biological conditions, which require further measurements in distinct marine regions necessary. Unfortunately, measurements are lacking that regard the abundance and molecular composition of amino acids in both seawater and size-segregated aerosol

particles, especially in the tropical Atlantic Ocean.

So, the aim of the present study is to investigate the occurrence of FAA in the marine environment regarding all important compartments; i.e. the ULW, the SML, the aerosol particles and finally cloud water in the remote tropical North Atlantic Ocean at the Cape Verde Atmospheric Observatory (CVAO). Their abundance, origin and possible transfer from the seawater as well as their transport within the atmosphere are studied in particular. Therefore, the FAA are measured on a molecular level and

divided into hydrophilic (glutamic acid, aspartic, GABA), neutral (serine, glycine, threonine, proline, tyrosine) and hydrophobic compounds (alanine, valine, phenylalanine, isoleucine, leucine) according to their hydropathy index. Especially the similarities and differences between the amino acid composition in submicron (0.05-1.2 µm) and supermicron (1.2-10 µm) aerosol particles are elucidated. Finally, the potential of individual FAA as proxies or tracers for specific sources of aerosol particles and cloud water in the tropical marine environment is outlined.

## 2. Experimental

### 2.1 Study area

Within the framework of the MarParCloud (Marine biological production, organic aerosol particles and marine clouds: a Process chain) project with contribution of MARSU (MARine atmospheric Science Unravelled: Analytical and mass spectrometric techniques development and application), a field campaign was performed at the CVAO from 13 September to 13 October 2017. This remote marine station is located on the northeast coast of the island of São Vicente, directly at the ocean (16°51''49'N, 24°52'02'E) which Carpenter et al. (2010) and Fomba et al. (2014) described in more detail. In accordance with the classification of Longhurst (2007), the ocean around the Cape Verde Islands belongs to the region "North Atlantic Tropical Gyral Province (NATR)", which is described as the region with the lowest surface chlorophyll in the North Atlantic Ocean having a greater annual variability than seasonality. During this campaign, concerted measurements were performed including the sampling of size-segregated aerosol particles at the CVAO and seawater sampling at the ocean site (~16°53'17'N, ~24°54'25'E). The location was carefully chosen with minimal influence of the island and located in wind direction to the CVAO as shown in Fig. S1. Here, aerosol particle sampler and cloud water sampler were installed at the mountain station on the top of the Mt. 'Monte Verde' (MV, 16°52'11'N, 24°56'02'W). van Pinxteren et al. (2020) provide further details on the MarParCloud campaign.

### 2.1.1 Seawater sampling

The seawater samples were taken from a fishing boat, starting from Bahia das Gatas, São Vicente. A glass plate with a sampling area of 2500 cm$^2$ was vertically immersed into the seawater and then slowly drawn upwards to take the SML. The surface films adhered to the surface of the glass plate and were removed with Teflon wipers directly into a bottle. This glass plate approach is described in detail by Cunliffe (2014). The ULW was sampled in a depth of 1 m into a plastic bottle fitted on a telescopic rod. To avoid influences from the SML, the bottles were opened underwater at the intended sampling depth. All seawater samples were stored in plastic bottles at -20 °C until the time of analysis. All materials for the seawater sampling were pre-cleaned with a 10 % HCl solution and high purity water.

### 2.1.2 Aerosol particles sampling

Size-segregated aerosol particles were sampled using five stage Berner-type impactors (Hauke, Gmunden, Austria) at the top of a 30 m sampling tower at the CVAO since this location best represents the conditions above the ocean pursuant to previous

studies. The internal boundary layer (IBL), which can form when air passes a surface with changing roughness (i.e. the transfer from open water to island) is mainly beneath 30 m (Niedermeier et al., 2014). Moreover, aerosol particles were sampled on the top of the Mt. MV (744 m a.s.l.). To avoid the condensation of atmospheric water on the aerosol particle sampling substrate, a conditioning unit consisting of a 3 m long tube was installed between the impactor inlet and the sampling unit. By heating the sampled air, the high relative humidity of the ambient air before collecting the aerosol particles was set to 75-80 %. The temperature difference between the ambient air at the impactor inlet and the sampled air after the conditioning unit was below 9 K (van Pinxteren et al., 2020). The Berner impactors were operated with a flow rate of 75 L min$^{-1}$ for 24 h and pre-baked aluminium foils (350°C for two hours) were used as substrate material. The five stage Berner impactor includes stage 1 (B1): 0.05-0.14 µm, stage 2 (B2): 0.14-0.42 µm, stage 3 (B3): 0.42-1.2µm, stage 4 (B4): 1.2-3.5 µm and stage 5 (B5): 3.5-10 µm. When it comes to the segregated aerosol particle samples, our study differentiates between the ones of submicron size (B1, B2, B3), the ones of supermicron size (B4, B5) as well as the ones of PM$_{10}$ (B1-5). After the sampling, the aluminium foils were stored in aluminium boxes at -20 °C until the time of analysis. It needs to be pointed out that the Berner impactors ran continuously, thus the impactor on the MV sampled aerosol particles also during cloud events. However, due to the pre-conditioning unit, the cloud droplets were efficiently removed before the aerosol particles were collected on the aluminium foils.

### 2.1.3     Cloud water sampling

At the MV station, an Acrylic glass Caltech Active Strand Cloud water Collector version 2 (CASCC2) according to Demoz et al. (1996) was used to sample cloud water. During a 'cloud event' the bottles were changed every 2-3 h, whereas on the other days the sampling time was e.g. overnight (every 12 h). For each sampling, the used Teflon rods were pre-cleaned with a 10 % HCl solution followed by high purity water. The liquid water content (LWC) of the cloud was measured continuously by a particle volume monitor (PVM-100, Gerber Scientific, USA). Finally, the collected cloud water was sampled in pre-cleaned plastic bottles and stored at -20 °C until the time of analysis.

### 2.2 Analyses
### 2.2.1     Seawater analyses

For the DOC/TDN content and the analysis of inorganic ions, the seawater samples were first filtered (0.45 µm syringe filter) and then quantified with a TOC-VCPH analyser (Shimadzu, Japan) or an ion chromatography (ICS3000, Dionex, Sunnyvale, CA, USA) as described in van Pinxteren et al. (2017). As the seawater samples must first undergo a desalination step for the FAA analysis, 32 mL (SML samples) or 48 mL (ULW samples) were desalinated using Dionex™ OnGuard™II Ag/H cartridges (Thermo Fisher Scientific™, Waltham, Massachusetts, USA). The volume of the desalinated samples was reduced to several µL using a vacuum concentrator at T=30 °C (miVac sample Duo, GeneVac Ltd., Ipswich, United Kingdom) with a recovery rate of >86 %. 0.2 µm syringe filters (Acrodisc-GHP; 25 mm, Pall Corporation, New York, USA) enabled the filtering of the enriched samples; then, a derivatization was performed with the AccQ-Tag™ precolumn derivatization method (Waters,

Eschborn, Germany). Besides, the FAA analysis includes the determination of glycine (Gly), L-alanine (Ala), L-serine (Ser), L-glutamic acid (Glu), L-threonine (Thr), L-proline (Pro), L-tyrosine (Tyr), L-valine (Val), L-phenylalanine (Phe), L-aspartic acid (Asp), L-isoleucine (Ile), L-leucine (Leu), L-methionine (Met), L-glutamine (Gln) and γ-aminobutyric acid (GABA) (purity ≥ 99 %, Sigma-Aldrich, St. Louis, Missouri, USA). An ultra-high performance liquid chromatography with electrospray ionization and Orbitrap mass spectrometry (UHPLC/ESI-Orbitrap-MS) performed the analytical measurements of the derivatized FAAs. The UHPLC system (Vanquish Horizon UHPLC system, Thermo Fisher Scientific™, Waltham, Massachusetts, USA) was coupled to an ESI-Orbitrap mass spectrometer (Q Exactive™ plus, Thermo Fisher Scientific™, Waltham, Massachusetts, USA) therefore. The samples were subsequently separated through an ACQUITY UPLC® HSS T3 column (Waters, Eschborn, Germany) with the dimensions 1.8 µm, 2.1 x 100 mm at a constant temperature of 30 °C and a detection in positive mode. The eluent composition consisted of (A) 0.2 vol % acetic acid in high purity water (Millipore Elix 3 and Element A10, Merck Millipore, Darmstadt, Germany) and (B) acetonitrile (Optima® LC/MS Grade, Fisher Scientific, Hampton, New Hampshire, USA). The flow rate of the eluent was 0.3 mL min$^{-1}$ and the eluent gradient program was 5 % B for 1 min, 5 % B to 100 % B in 16 min, 100 % B for 2 min constant, in 0.1 min from 100 % B to 5 % B and the 5 % B was then kept constant for 3.9 min. This analytical procedure can be used for amines, too, as described in van Pinxteren et al. (2019). The FAA concentrations were determined via external calibration. Since no chiral column was used in the UHPLC separation, we cannot differentiate between L- and D- amino acids in our ambient samples. Each seawater sample was measured as a duplicate with relative standard deviation <10 % and under consideration of the blank samples for seawater. They consist of high purity water, which was filled in pre-cleaned plastic bottles and handled the same as the seawater samples. The limit of quantification (LOQ) of the individual FAAs in seawater samples is in good agreement with the FAA analysis in seawater samples (e.g. Kuznetsova et al. (2004)) and listed in Table S1. The LOQs were mostly below 10 nmol L$^{-1}$, however, GABA and Met exhibited LOQs with 24.2 nmol L$^{-1}$ and 16.8 nmol L$^{-1}$, respectively (due to high blank values). A quantification of some FAAs in seawater, mainly in the ULW with its generally lower FAA concentrations compared to the SML, is therefore partly limited.

### 2.2.2 Aerosol particle filter analyses

For analysing the size-segregated aerosol particle samples, the substrate material of each stage was extracted in 3 mL high purity water (Millipore Elix 3 and Element A10, Merck Millipore, Darmstadt, Germany). The aqueous particle extracts were divided into aliquots for the analysis of water-soluble organic carbon (WSOC)/total dissolved nitrogen (TDN), inorganic ions and amino acids. The aliquots for WSOC/TDN were first filtered with a 0.45 µm syringe filter and then determined by a TOC-VCPH analyser (Shimadzu, Japan) as described in van Pinxteren et al. (2012). For the analysis of inorganic ions, the aliquots (250 µL) were filtered (0.45 µm syringe filter) and investigated using ion chromatography (ICS3000, Dionex, Sunnyvale,CA, USA) as outlined in Mueller et al. (2010). The aliquot (1.5 mL) of the aqueous particle extracts for FAA analysis was reduced to several µL with a vacuum concentrator at T=30 °C (miVac sample Duo, GeneVac Ltd., Ipswich, United Kingdom), filtered using 0.2 µm syringe filters and derivatized as well as analyzed using the UHPLC/ESI-Orbitrap-MS method as explained in

section 2.2.1 for seawater samples. FAA concentrations were calculated via external calibration; each sample was measured in duplicate with a relative standard deviation <10 % and under consideration of field blanks. For generating field blanks, pre-baked aluminium foils without active sampling, were cut and prepared the same as field samples, including extraction and measurements for WSOC/TDN, inorganic ions and amino acids analysis. All values presented here for aerosol particle samples

are field blank corrected. The LOQs of the individual FAAs in aerosol particle samples are listed in Table S1 and are in good agreement with the sensitivity of other analytical methods for FAAs in aerosol particles (e.g. Matsumoto and Uematsu (2005)). Although a variance in LOQs between the individual FAAs is apparent, FAAs with relatively high LOQs (39.5 pg m$^{-3}$) on aerosol particles such as Ala, GABA, Asp in submicron and supermicron aerosol particles could be quantified (as discussed in section 3.2 and 3.4).

The analysis of mineral dust tracers on nucleopore foils sampled with the Berner impactor was performed with the Total Reflection X-Ray Fluorescence S2 PICOFOX (Bruker AXS, Berlin, Germany) equipped with a Mo-X-ray source on polished quartz substrates as can be seen in Fomba et al. (2013). The particulate mass (PM) of the aerosol particle samples was determined by weighing the filter before and after sampling (van Pinxteren et al., 2015). Back trajectory analyses provided information regarding the origins of air masses. Seven-day back trajectories were calculated on an hourly basis within the

sampling intervals, using the NOAA HYSPLIT model (HYbrid Single-Particle Lagrangian Integrated Trajectory, http://www.arl.noaa.gov/ready/hysplit4.html, 26.11.16) in the ensemble mode at an arrival height of 500 m ± 200 m (van Pinxteren et al., 2010); van Pinxteren et al. (2020) provide more details. The calculated backward trajectories are representative for both aerosol particle sampling stations (CVAO and MV)**.**

### 2.2.3    Cloud water analyses

The cloud water samples were processed the same as seawater samples for the analysis of DOC/TDN and inorganic ions (section 2.2.1). For the amino acid analysis, the volume of cloud water samples (44 mL) was reduced to several mL using a vacuum concentrator at T=30 °C (miVac sample Duo, GeneVac Ltd., Ipswich, United Kingdom). After the filtration with 0.2 µm syringe filters (Acrodisc-GHP; 25 mm, Pall Corporation, New York, USA), an aliquot of the prepared cloud water was

derivatized based on the AccQ-Tag™ precolumn derivatization method (Waters, Eschborn, Germany). The analytical measurements of the derivatized FAA were performed with UHPLC/ESI-Orbitrap-MS (section 2.2.1). The cloud water samples were measured as duplicates with a relative standard deviation <10 %. Via external calibration the amino acid concentrations under consideration of the cloud water blanks were calculated. The blank samples of cloud water were generated by rinsing the pre-cleaned Teflon rods with high purity water after its installation in the cloud water sampler. Then, the blank

samples were handled the same as the field cloud water samples including the derivatization and analytical separation as described in section 2.2.1. Overall, the LOQs of the individual FAAs in cloud water samples are in good agreement with the reported sensitivity of the FAA analysis in cloud water (Bianco et al., 2016) and listed in Table S1. Since the LOQs of the FAAs in cloud water are below 0.3 ng m$^{-3}$ and often below 0.06 ng m$^{-3}$, a limitation of the FAA composition in cloud water due to the LOQs is rather unlikely despite the variance of FAA concentrations (11.2-489.9 ng m$^{-3}$) in cloud water (section 3.3).

To calculate the atmospheric concentration of FAA in cloud water, the measured concentrations were multiplied with the measured liquid water content (LWC) of the clouds as Fomba et al. (2015) applied beforehand.

### 2.2.4 Enrichment factors

The enrichment factor in the SML ($EF_{SML}$) was calculated by dividing the concentration of the analyte in the SML with the concentration of the analyte in the ULW using the following equation (1):

$$EF_{SML} = \frac{c\ (analyte)_{SML}}{c(analyte)_{ULW}} \qquad (1)$$

Accordingly, both an enrichment in the SML with $EF_{SML} > 1$ and a depletion in the SML with $EF_{SML} < 1$ are indicated.

The FAA concentration in the ULW was assumed to be based on the concentration (LOQ/2) of individual amino acids for seawater samples from the same campaign day when individual FAA could be quantified in the SML samples, but not in the corresponding ULW ones due to FAA values below the LOQs (listed in Table S1). For the calculation of this estimated $EF_{SML}$, specially marked in the following, the concentration 25.2 nmol $L^{-1}$ was used for $c(analyte)_{ULW}$ in equation (1).

To calculate the enrichment factor of the individual analytes in different matrices (M), the concentration of the analyte in matrix 1 ($M_1$) relative to the sodium ($Na^+$) concentration in $M_1$ was divided by the analyte concentration in matrix 2 ($M_2$) relative to the $Na^+$ concentration in $M_2$ using equation (2):

$$EF_{M1} = \frac{c\ (analyte)_{M1}/c\ (Na^+)_{M1}}{c\ (analyte)_{M2}/c\ (Na^+)_{M2}} \qquad (2)$$

The aerosol enrichment factor ($EF_{aer}$) were calculated in each of the five Berner stages ($B_x$ with x = 1-5 as $M_1$) using the respective analyte and $Na^+$ concentration in relation to the SML or the ULW as $M_2$. For this purpose, the aerosol particle concentrations, typically sampled in a 24-hour interval, were combined with SML/ULW concentrations, which had been collected during the aerosol particle sampling period. The analyte concentration in each size class of size-segregated aerosol particle samples (B1-5) was combined with the analyte concentration in SML/ULW. The calculation of the $EF_{aer}$ was limited to the availability of data in both matrices – size-segregated aerosol particles and SML/ULW samples. The $EF_{aer}$ could only be calculated if both the analyte concentration and the sodium concentration could be quantified in the size-segregated aerosol particles and the corresponding SML/ULW samples. To calculate the enrichment factor in cloud water ($EF_{CW}$), the concentration of the analyte and of $Na^+$ in the cloud water were considered as $M_1$ and those of the SML or the ULW as $M_2$. The determination of $EF_{aer}$ was possible for n=3 samples both on the basis of SML and ULW. The $EF_{CW}$ could only be determined for n=1 sample though basing on the SML and ULW measurements. Section 3.4 discusses both the $EF_{aer}$ and the $EF_{CW}$ in more detail.

## 3. Results and Discussion

### 3.1 Seawater samples

#### *Free amino acids in seawater samples*

FAA were measured in the seawater as a source region of FAA on primary marine aerosol particles. Fig. 1b shows the measured

5 ∑FAA concentration in the SML and the ULW samples together with their enrichment factor $EF_{SML}$ (Eq. 1).

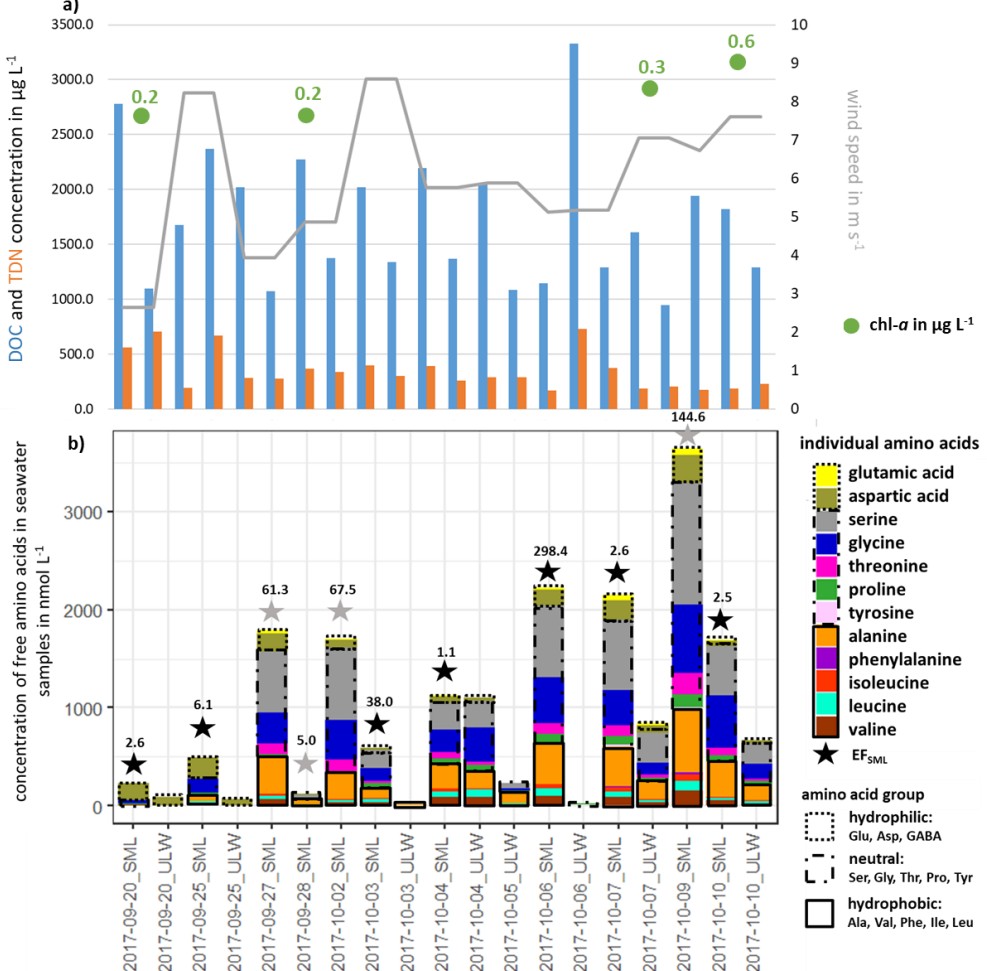

**Figure 1: a) DOC, TDN and chl-*a* concentration in seawater and windspeed and b) individual FAA concentration in the seawater samples and the enrichment factor $EF_{SML}$ of ∑FAA; $EF_{SML}$ based on measurements (black stars), $EF_{SML}$ based on LOQ/2 estimation (grey stars)**

10 ∑FAA included all investigated amino acids (listed in 2.2.1) except for Met, Gln and GABA. As discussed in section 2.2.1, GABA and Met have the highest LOQs of the analytical method used here, which may be one reason why these two analytes could not be quantified in the seawater samples (ULW and SML).

Looking at the percentage composition within the ULW (10.1 % hydrophilic, 57.0 % neutral, 32.8 % hydrophobic) and the SML (10.6 % hydrophilic, 61.7 % neutral, 27.7 % hydrophobic), the values are similar to each other. However, the concentration of ∑FAA varied between 0.01-1.10 µmol L$^{-1}$ in the ULW and between 0.13-3.64 µmol L$^{-1}$ in the SML. Interestingly, in the second half of the campaign, the ∑FAA concentration was higher than in the first part. Previous studies in

different oceanic areas (Kuznetsova and Lee, 2002;Kuznetsova et al., 2004;Reinthaler et al., 2008;van Pinxteren et al., 2012;Engel and Galgani, 2016) have already reported a general strong variability of ∑FAA concentration, especially in the SML. Reinthaler et al. (2008) concluded that the SML in the open ocean is a highly variable environment with high concentrations of dissolved FAA and their high enrichment in the SML, but without clear diel variations in their concentrations. Nevertheless, the variance of the ∑FAA concentrations in the SML or ULW observed here could neither be explained by the

variance of DOC or TDN values, nor by wind speed and chl-*a* concentrations (see Fig. 1, Table S2, S5), since no correlation between these parameters and the concentration or enrichment of FAA was found. This is consistent with other publications which observed that the amino acid concentration in seawater is not related to environmental parameters such as wind, humidity and light (Kuznetsova et al., 2004;van Pinxteren et al., 2012). The results of the individual FAA concentrations in seawater (ULW, SML) and their EF$_{SML}$, listed in Table S3, show clear differences between the individual amino acids and the amino

acid classes. The most highly enriched amino acids in the SML are the neutral ones with values of up to 203 compared to the hydrophilic (EF$_{SML}$: 2-98) and hydrophobic (EF$_{SML}$: 1-96) amino acids. This may be related to the fact that Ser, Thr and Gly as part of the neutral amino acids, are main components of cell wall proteins (Hecky et al., 1973). The direct release of FAAs through cell lysis and the associated destruction of the cell wall can thus explain the increased enrichment of neutral amino acids in the SML. Our study confirmed that the SML is often non-uniformly enriched with FAAs as outlined from previous

observations (Kuznetsova and Lee, 2002;Reinthaler et al., 2008;van Pinxteren et al., 2012;Engel and Galgani, 2016). Different factors, such as the transport of FAA from the ULW to the SML, the in-situ production by an extracellular hydrolysis of CAA or a direct release of FAA by cell lysis probably cause the observed enrichment of FAA in the SML. Kuznetsova and Lee (2002) showed that the rapid extracellular hydrolysis of CAA in the SML was not the cause of the non-uniformly enrichment in SML. Moreover, they suggested that the intracellular pools of organisms rich in DFAA and DCAA compared to seawater

can be leached out by stressed microorganisms, resulting in the release of DFAA which in turn influences the pools of both DFAA and DCAA in seawater. Based on previous studies, the transportation and releasing mechanisms seem most likely to be the reasons for the observed enrichment of FAA. However, further experiments are required to finally elucidate the most important drivers causing the enrichment.

Altogether, it can be concluded that there is some variability within the FAA concentration in the SML and in the ULW, with

a clear trend of its strong enrichment in the SML. The fact that the FAA concentrations were in accordance with the ones measured at the same location in 11/2013 (0.64 µmol L$^{-1}$, Table S4), supports the suggestion that the FAA concentrations reported here can be considered representative of the NATR region as part of the North Atlantic Ocean. These concentrations are generally similar comparing them to FAA concentrations in other marine regions (Kuznetsova and Lee, 2002;Reinthaler et al., 2008). Reinthaler et al. (2008) considered concentrations of dissolved FAA of 0.02-0.13 µmol L$^{-1}$ (ULW) and of 0.43-

11.58 µmol L$^{-1}$ (SML) in the subtropical Atlantic Ocean as well as values of 0.07-0.60 µmol L$^{-1}$ (ULW) and of 0.77-3.76 µmol L$^{-1}$ (SML) in the western Mediterranean Sea. Consequently, the FAA concentrations in the NATR region, with its very low surface chlorophyll and a greater annual variability than seasonality, are in the same order of magnitude compared to other marine regions (i.e. subtropical Atlantic and western Mediterranean Sea (Reinthaler et al., 2008)).

*Contribution of FAA to DOC and TDN content in seawater*

DOC and TDN concentrations and their enrichment in the SML (EF$_{SML}$) are listed in Table S5. The contribution of ∑FAA to DOC or TDN in seawater had been calculated (taking into account the carbon and nitrogen content of the amino acids, Table S6) and is also listed in Table S5. The carbon content of ∑FAA contributed to the DOC with values between 0.1-7.6 %. in the

seawater samples and a median of 2.4 % (n=17), differing between 2.8 % (n=11) in the SML and 1.8 % (n=6) in the ULW samples. Looking at the nitrogen content from ∑FAA to TDN in the seawater samples, 0.1-42.4 % of the TDN consisted of ∑FAA with a median of 8.3 % (n=18). In the SML, ∑FAA contributed on average with 11.9 % (n=11) whereas they contributed in the ULW with 3.2 % (n=7) to TDN. The observed daily variations within the contribution of ∑FAA to DOC/TDN, result from the daily variations of ∑FAA concentration in seawater (Fig. 1) and of DOC/TDN (Table S5). In the

SML of the Atlantic Ocean and the western Mediterranean Sea, the DFAA contributed with ~ 12 % of the DOC and ~ 30 % of the dissolved organic nitrogen (DON) (Reinthaler et al., 2008). Our results regarding the contribution to DOC were of the same order of magnitude, but slightly lower than those of Reinthaler et al. (2008).

## 3.2 Size-segregated aerosol particles

**3.2.1 Size-segregated aerosol particles at the CVAO**

*First indications of aerosol particle origin*

To obtain a first indication of the particle origin, that might help to explain the differences in the particle composition concerning amino acids, the particles were associated with the origin of the air masses and with marine and dust tracers. Overall, the CVAO station experienced north-easterly trade winds during this campaign, which are typical for this season

within this region (Fomba et al., 2014;van Pinxteren et al., 2020). According to physical and chemical specifications such as the air mass origins, particulate MSA concentrations and MSA/sulfate ratios as well as particulate mass concentrations of dust tracers, aerosol particles predominantly of marine origin with low to medium dust influences were observed. The dust and marine tracers of the aerosol particles considered here are discussed in more detail in SI (Table S8 and in 'aerosol particles: dust and marine tracers'). Further information on the classification of the air masses and distinct concentrations of dust tracers

are given in the overview paper of this campaign (van Pinxteren et al., 2020).

*Free amino acids in size-segregated aerosol particles: Concentrations*

The lower panel of Fig. 2b shows the atmospheric concentration of FAAs in each Berner stage at the CVAO whereas the upper panel represents the concentration in the submicron, the supermicron and PM aerosol particle size range. In the submicron

aerosol particles, the concentration of ∑FAA was between 1.3 ng m$^{-3}$ (1/10/2017) and 6.3 ng m$^{-3}$ (7/10/2017). Whilst the concentration ∑FAA varied between 0.2 ng m$^{-3}$ (6/10/2017) and 1.4 ng m$^{-3}$ (22/09/2017) in the supermicron size range, the highest atmospheric concentrations of ∑FAA were found in the submicron aerosol particles (mean of 3.2 ng m$^{-3}$) compared to the supermicron ones (mean of 0.6 ng m$^{-3}$). Daily variations of the ∑FAA content on the investigated size-segregated aerosol

particle samples were observed: the ∑FAA tended to increase slightly along the campaign. OM parameterization studies showed that wind speed and chl-*a* concentrations were most important parameters for the regulation of the OM production in sea spray aerosol particles (Gantt et al., 2011;Rinaldi et al., 2013;van Pinxteren et al., 2017). Correlations between the ∑FAA concentrations of the size-segregated aerosol particles (considered as submicron, supermicron and PM$_{10}$) and the wind speed were not observed for here reported data (Fig. 2, Table S2). However, the available wind speed and wind direction data

represented an average value of 24 hours. Therefore, shortly pronounced changes in the wind speed that might have affected the amino acids transfer would not have been visible in the averaged wind speed value. The major source of bubbles are whitecaps or breaking waves, that occur when the wind speed exceeds 3-4 m s$^{-1}$ (Blanchard, 1975), which was continuously reported during the campaign. Hence, the high wind speeds together with the constantly observed breaking waves indicated that the wind intensity in this region might be consistently sufficient to transfer the amino acids from the ocean into the

atmosphere. No significant correlation could be observed between the ∑FAA concentration of size-segregated aerosol particle samples (submicron, supermicron and PM$_{10}$) and the chl-*a* concentration in seawater. Nevertheless, the increasing chl-*a* concentration along the campaign (Fig. 2, Table S2) could be a reason for the slight increase in the concentrations of ∑FAA in seawater and on submicron aerosol particles, indicating a possible connection between ocean and atmosphere, e.g. the transfer of amino acids from the ocean into the atmosphere.

Overall, the concentrations reported here agree well with other FAA studies on marine aerosol particles. Matsumoto and Uematsu (2005) found averaged total concentrations of dissolved FAA with 4.5 ng m$^{-3}$ on aerosol particles (average of < 2.5 µm and > 2.5 µm) in the western North Pacific Ocean. Moreover, Wedyan and Preston (2008) observed an average concentration of dissolved FAA of 2.5 ng m$^{-3}$ on total suspended particles (TSP) during a transect ship cruise in the Atlantic Ocean. For Antarctic aerosol particles, the observed mean total FAA concentration on size-segregated aerosol particle samples

(< 0.49-10 µm) at the Mario Zucchelli Station was 4.6 ng m$^{-3}$ (Barbaro et al., 2015). Hence, regarding the sum of FAA, a striking similarity was found between FAA concentrations in different parts of the ocean that probably underlay different influences (e.g. pristine region in the Southern Ocean, continental-influenced aerosol particles in the North Pacific Ocean).

*Free amino acids in size-segregated aerosol particles: Composition*

∑FAA included all investigated amino acids (listed in 2.2.1) except for Met and Gln, analytes which were neither detected in the size-segregated aerosol particle samples. The most abundant FAA was Gly, which was consistently found in submicron and supermicron aerosol particles, followed by Ala and Ser. However, towards the end of the campaign (4/10/2017-7/10/2017), a high contribution of the hydrophilic FAAs GABA and Asp was detected (shown in the upper panel of Fig. 2b), which caused the slight increase of the total FAA concentration.

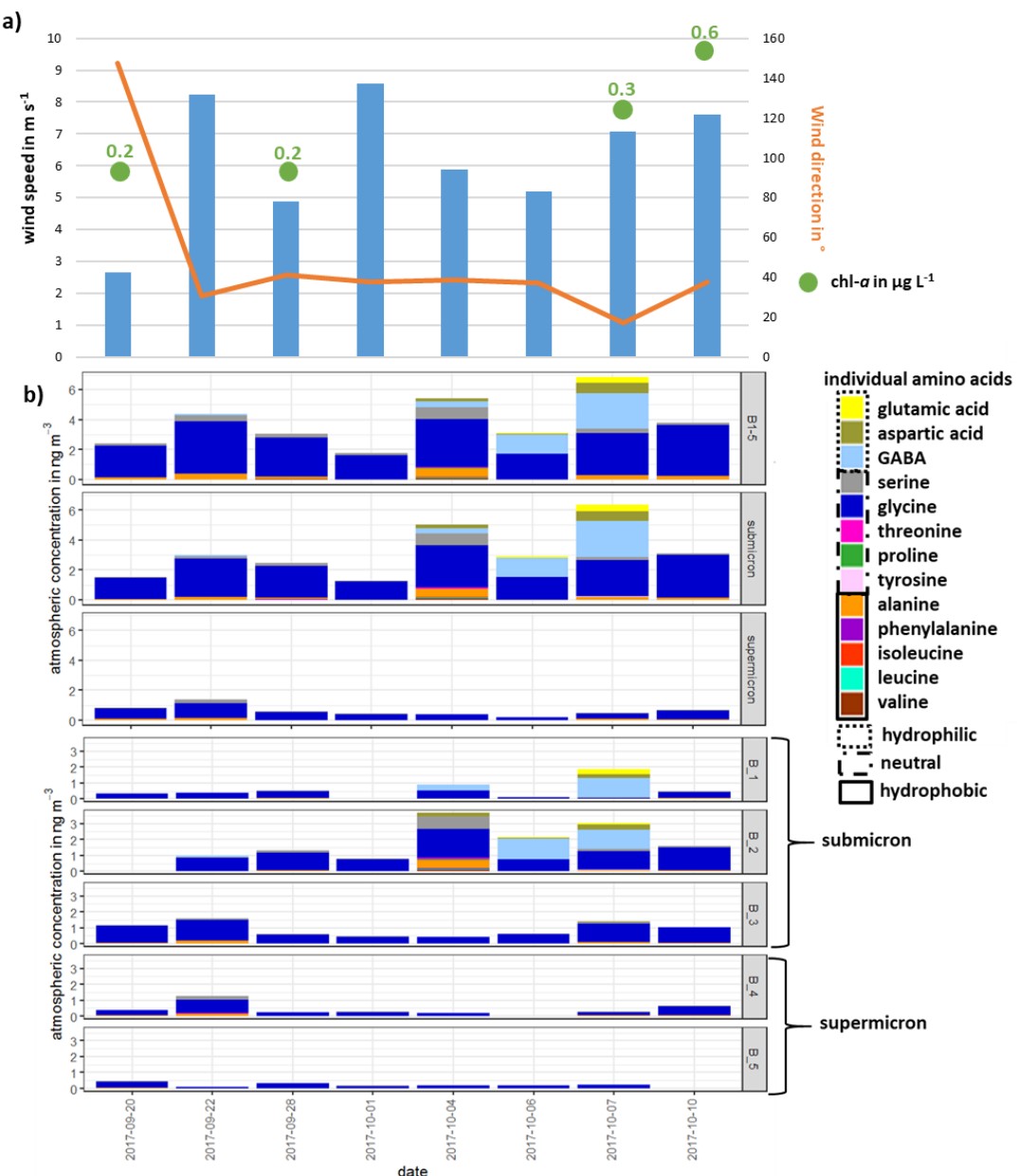

**Figure 2: a) overview of wind speed and wind direction at the CVAO and the chl-*a* concentration in seawater; b) atmospheric concentration of individual FAA: sum of all Berner stages (B1 - 5), in the submicron and supermicron size range (upper panel) and the atmospheric concentration of FAA in all individual Berner stages (lower panel) during the campaign at the CVAO**

5    The higher complexity in the FAA composition on the submicron aerosol particles could only be determined because the

analytical method applied here was able to quantify the individual molecular FAA species. Such differentiation would not be

possible with methods that determine the proteins as a sum parameter (e.g. the often applied Bradford method).

The high abundance of Gly in the aerosol particles is in good agreement with the Gly occurrence in other marine studies. Gly seems to be the dominant FAA, independent of whether the particles were sampled in the Arctic (Scalabrin et al., 2012), Antarctic (Barbaro et al., 2015) or in the North Pacific (Matsumoto and Uematsu, 2005) and whether they are attributed to a local marine source (Wedyan and Preston, 2008;Barbaro et al., 2015) or are rather continental or long-range influenced

(Matsumoto and Uematsu, 2005). Other abundant FAAs reported at the different locations are arginine (not analyzed here), Ser and Ala: the latter ones are also dominant FAAs found in the here reported study. Consequently, the usage of the major FAAs as chemical biomarkers seems to be restricted to some extend due to their lack of source-specifity. The high abundance of GABA found in the second half of the campaign has neither been partly regarded (i.e. included as a standard compound) in some marine studies, nor yet been reported in ambient marine aerosol particles, but seems to be special for this location.

However, the reasons for the high concentration of hydrophilic FAAs within these respective sampling days remain unclear, since no change in the environmental parameters determined (e.g. wind speed, wind direction, chl-*a* concentration, Fig. 2a) was observed. In addition, we considered further FAA physico-chemical parameters such as the octanol-water partition coefficient ($K_{OW}$), the topological polar surface area (TPSA), which describes the surface activity, and the density (Table S9) to describe the concentration changes. However, no statistically relevant correlations between the FAA concentration or

composition and physico-chemical parameters were found here either. Our observations could not clarify possible additional (i.e. non-marine) sources leading to the higher concentrations and complexity in the FAA composition. The dynamics behind the varying FAA concentration and composition at this location seem to be complex.

Following this hydropathy classification, the submicron aerosol particles consisted on average of 5 % hydrophobic, 15 % hydrophilic and 80 % neutral amino acids, while the supermicron aerosol particles contained on average only 7 % hydrophobic

and 93 % neutral amino acids (Table S7). During the campaign, an increase in the contribution of hydrophilic amino acids was observed with a maximum of 55 % on 7/10/2017. Barbaro et al. (2015) reported that hydrophilic components were predominant (60 %) in locally produced marine Antarctic aerosol particles, whereas hydrophobic compounds were rather dominate aerosol particles collected at the continental station (23 % and 27 %). The relatively high content of hydrophilic FAAs during certain periods of the campaign points at least at some influence of local oceanic sources.

*Contribution of FAA to WSOC and WSON*

In consideration of the carbon or nitrogen content of the amino acids (Table S6), the contribution of ∑FAA to WSOC and water-soluble organic nitrogen (WSON) in the size-segregated aerosol particles was calculated (Table S10). In the submicron size range, ∑FAA contributed up to 5.3 % (average 1.1 %) to WSOC, while in the supermicron range, ∑FAA only contributed

up to 0.04 % to WSOC. Looking at ∑FAA's total contribution to WSOC ($PM_{10}$), 0.7 % of WSOC consists of ∑FAA, which is in good agreement with the value of the study by Mandalakis et al. (2011). Considering the nitrogen content of the amino acids, ∑FAA contributed to the estimated WSON (WSON = 25 % of measured TDN concentrations according to Lesworth et al. (2010)) with an average of 0.4 % in the submicron and of 0.05 % in the supermicron size range. The observed daily variations of the contribution of ∑FAA to WSOC/WSON were derived from the daily variations of the atmospheric

concentration of $\sum$FAA (Fig. 2) and of WSOC/ WSON (Table S10). In summary, $\sum$FAA contributed up to 5.3 % to WSOC and to 1.8 % to WSON when it comes to the submicron aerosol particles (7/10/2017) and up to 0.15 % to WSOC and to 0.1 % to WSON for the supermicron aerosol particles. These percentages were in the same order of magnitude as for other organic compound groups, e.g. amines. van Pinxteren et al. (2019) showed that amines contributed on average 5 % to the submicron
WSOC content on marine aerosol particles. Especially, the percentage of $\sum$FAA to WSOC (up to 5.3 %) in the submicron aerosol particles demonstrated that FAA comprised a substantial fraction of submicron WSOC in marine aerosol particles.

### 3.2.2 Size-segregated aerosol particles at the MV

From the MV samples, FAAs and additional parameters such as PM, WSOC, sodium and MSA were investigated. The results
are listed in Table S11. The submicron aerosol particles at the MV had an averaged $\sum$FAA concentration of 1.5 ng m$^{-3}$ (0.8-1.9 ng m$^{-3}$) and were about three times lower compared to the $\sum$FAA concentration at the CVAO. The $\sum$FAA concentration in the supermicron aerosol particles at the MV (1.2 ng m$^{-3}$; 0.2-2.9 ng m$^{-3}$) was similar to the respective concentrations at the CVAO. Additional online measurements of particle size number distributions (PSND) at the CVAO and the MV, described in Gong et al. (2020) were in good agreement with one another during cloud-free times. This indicated that, for cloud-free
conditions, the aerosol particles measured at ground level (30 m) within the IBL, which is mainly below 30 m (Niedermeier et al., 2014), represented the aerosol particles at cloud level. Thus, the aerosol particles within the marine boundary layer (MBL) were well mixed and the Mt. Verde was most of the time within the (MBL) (van Pinxteren et al., 2020). However, as described above, the Berner measurements were (continuously) taken during cloud-free as well as during cloud times. The concentration and composition of the aerosol particles can therefore be affected by the clouds that formed and disappeared consistently
during the sampling period of the aerosol particles at the Mt. Monte Verde (for further details on the frequency of the cloud events see Gong et al. (2020) and van Pinxteren et al. (2020)). There was also no rain during the entire campaign. Furthermore, ageing processes may occur during the upwind of the aerosol particles from the CVAO to the MV station, which takes about 4 h considering an average vertical wind of 5 cm s$^{-1}$ (van Pinxteren et al., 2020). The particles at the MV exhibited lower particle masses, as well as lower concentrations of the aerosol particle constituents. The decrease in concentrations of $\sum$FAA,
PM, sodium, MSA and WSOC was reduced by a factor of three to four regarding the submicron aerosol particles. However, no uniform depletion ratio between their concentration at the CVAO and the MV was found for the supermicron aerosol particles (Table S11). While the PM of the supermicron particles was reduced by a factor of four at the MV (similar to the submicron aerosol particles), sodium and WSOC were depleted more strongly (factor of 11-12) compared to their respective concentrations at the CVAO. This suggests that the submicron particles were rather uniformly affected and depleted, likely by
cloud processes, while the supermicron particles were influenced by clouds, and potentially other sources, in a non-uniform way. Nevertheless, the abundance of the marine tracers (sodium, MSA), together with the presence of FAA in the aerosol particles (which mainly had a similar composition compared to the oceanic and ground-based particulate FAA) indicated an oceanic contribution to the aerosol particles at cloud level.

### 3.3  Cloud water samples

The concentration of FAA in cloud water (Fig. 3, Table S12) was, although varying, always significantly higher than the aerosol particles (Table S8) and several orders of magnitude above the LOQs (Table S1). The individual atmospheric concentration of FAA in cloud water was calculated based on the measured liquid water content (LWC) (section 2.2.3 and Table S12). The ∑FAA concentrations varied strongly between 11.2 and 489.9 ng m$^{-3}$ as shown in Fig. 3.

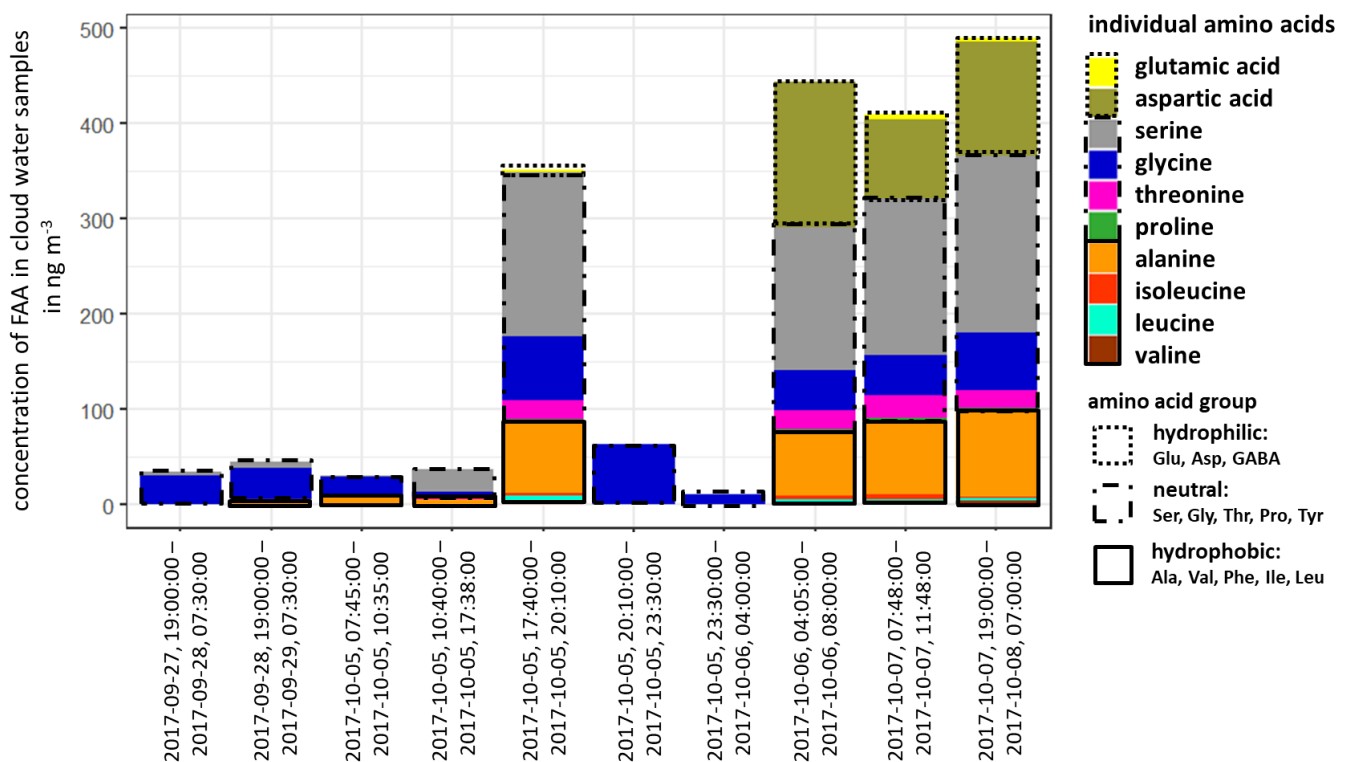

**Figure 3: Concentration of individual FAA in cloud water samples at the MV station in ng m$^{-3}$. The time represents the local start and end time of the cloud water sampling**

The inorganic marine tracers in cloud water (Na$^{+}$: 5.7 µg m$^{-3}$, MSA: 25.1 ng m$^{-3}$, Table S12) were also present in higher concentrations compared to the aerosol particle samples at the CVAO (submicron: Na$^{+}$: 72.3 ng m$^{-3}$, MSA: 6.0 ng m$^{-3}$) and the MV (submicron: Na$^{+}$: 17.0 ng m$^{-3}$, MSA: 1.8 ng m$^{-3}$, Table S11). The concentrations of cloud water sulfate (average: 2.9 µg m$^{-3}$, Table S12) and sodium were higher than in cloud water samples, collected at East Peak in Puerto Rico, which can be seen in Gioda et al. (2009). Our observed carbon concentration of FAA in cloud water at the MV station was between 17-757 µg C L$^{-1}$ and in the same order of magnitude as in a previous study of cloud water sampled on top of puy de Dôme mountain, inland of France (211±19 µg C L$^{-1}$, Bianco et al. (2016)), but showed a higher variance. Besides the concentration, the composition of FAA in cloud water also showed a high variability in the study presented here. In cloud water samples with ∑FAA <65 ng m$^{-3}$, Gly was usually dominant, followed by Ser. However, cloud water samples with ∑FAA >290 ng m$^{-3}$ showed a higher complexity in FAA composition, including the concentrations of Asp and Ala. Other abundant FAA were Thr, Leu and Ile. In

terms of the hydropathy classification, the first part of the campaign (27/09/2017-5/10/2017) was dominated by neutral FAAs, whereas a sudden increase of the hydrophilic FAAs was observed in its second part (06/10/2017-08/10/2017). Comparative studies on the FAA composition of cloud water in the marine environment are lacking, but especially in the second part of the campaign, it pointed to a local marine (biogenic) influence. The high concentrations of Asp might be related to diatoms and zooplankton in seawater (Hammer and Kattner, 1986). Scalabrin et al. (2012) reported local marine sources for Ile, Leu and Thr detected in aerosol particles, whereas Mashayekhy Rad et al. (2019) suggested coastal and marine phytoplankton and bacteria as possible sources for these amino acids. Therefore, the FAA composition might be related to an oceanic transfer via bubble bursting and/or microbial in-situ production. Interestingly, GABA, which was highly abundant on the aerosol particles, maybe due to biogenic production, was not present in the cloud water samples. The presence of the marine tracers (sodium, MSA) in cloud water supports a coupling to oceanic sources. In addition, the majority of low-level clouds were formed over the ocean and ocean-derived components are expected to have some influence on cloud formation (van Pinxteren et al., 2020). Nevertheless, contributions from the desert and other non-marine sources cannot be excluded.

The reason for the high concentrations of FAA in cloud water (compared to the oceanic and aerosol particle concentrations) remain speculative to date and will be subject of further studies. Altogether, the in-situ formation of FAA in cloud water by chemical abiotic processes in the cloud or by atmospheric biogenic formation, as proposed by Jaber Jaber et al. (2020), as well as by selective enrichment processes and pH-dependent chemical reactions might be potential additional sources besides aerosol particles.

### 3.4  Concerted measurements of FAA in the marine compartments (seawater, aerosol particles and cloud water)

Only a few studies which concern the simultaneous investigation of FAA in the marine compartments – seawater, aerosol particles and cloud water - using concerted measurements are present to date; most of them measured artificially generated aerosol particles. Kuznetsova et al. (2005) characterized proteinaceous compounds in marine ambient aerosol particles, in generated aerosol particles and in corresponding SML samples. Rastelli et al. (2017) investigated the transfer of OM (sum parameter for lipids, carbohydrates and proteins) from the ocean surface into marine aerosol particles under controlled conditions using a bubble-bursting experimental system. In previous studies, the transfer of microorganisms from the ocean to the aerosol particles could be reported (Aller et al., 2005;Pósfai et al., 2003) and even on submicron marine aerosol particles viruses and prokaryotes were present (Rastelli et al., 2017).

Within the here presented study, a simultaneous sampling of all marine matrices - seawater (ULW, SML), size-segregated aerosol particles (CVAO, MV) and cloud water samples - could be obtained for a period between 4/10/2017 and 7/10/2017 comprising 6 blocks of size-segregated aerosol particles (3 at the CVAO and 3 at the MV), 3 seawater samples (3 SML and 3 ULW) and one cloud water sample (7/10/2017; 7:48-11:48). For these sampling intervals, the fractional residence time of the air masses was mainly above water and the mass concentration of trace metals and inorganic marine tracers (sodium, MSA) (Table S8) strongly suggest a dominant marine origin of air masses. Sources other than marine (dust, continental) are, by contrast, of minor importance during this sampling period. The averaged values of these sampling days represent a case study

to combine and compare the FAAs in all matrices to investigate a possible transfer of FAAs from the ocean into the atmosphere and a possible transport of FAAs within the atmosphere. The comparability of the different matrices (e.g. seawater samples as a spot sample, aerosol particles samples covering a 24 h period) is discussed in Fig. S2.

The averaged FAA composition of this case study in all marine compartments is shown in Fig. 4. The high complexity of FAA observed in seawater was also found in the aerosol particles as well as in cloud water, and generally shows a high similarity between FAA in the different compartments. All marine compartments contained Gly, Ser, Glu and Ala as dominant species, i.e. representatives of the hydrophilic, neutral and hydrophobic groups. However, the percentage contribution of the individual FAAs to the ∑FAA varies within the different compartments.

Representatives of the hydrophilic, neutral, hydrophobic and aromatic amino acids are discussed below with respect to their distribution within the different marine matrices and with regard to a potential transfer. For a better comparison of the individual amino acids, the mean life time $\tau$ of the amino acids in the CVAO ('remote aerosol case') and in the MV ('remote cloud case') aerosol particle samples were considered as described in Table S13. The mean life time $\tau$ of the individual amino acids depends on the pH-dependent rate constant k and the OH radical concentration of the different atmospheric scenarios (SI, Eq. (3)).

### 3.4.1. Hydrophilic amino acids

The hydrophilic amino acids (Asp, Glu, GABA) comprised a significant fraction in the ULW and the SML, as well as in the (submicron) aerosol particles and in cloud water (Fig. 4a-d). They were not detected in the supermicron aerosol particles. A conspicuous finding is the high concentration of GABA, which is present exclusively in the submicron aerosol particles (B1 and B2: 0.05-0.42 µm) at the CVAO. Despite the relatively high LOQ of GABA in seawater (Table S1), a major abundance of GABA in seawater would be detectable. GABA is a metabolic product of the microbiological decarboxylation of Glu, which has been detected in all marine compartments. Active microbial enzymes on nasecent sea spray aerosol have recently been reported by Malfatti et al. (2019). The abundance of GABA on the submicron aerosol particles suggests that either GABA could have been produced by microbiological decarboxylation of Glu by present (marine) microorganisms on the aerosol particles, or that GABA was transferred from the seawater to the atmosphere. However, GABA could not be found in seawater (ULW and SML) and this is not related to the sensitivity of the analytical method. Hence, a very enhanced oceanic transfer of GABA would be needed to explain this finding. Such an enhanced transfer was, however, not observed for the other hydrophilic amino acids (Glu and Asp), their percentage composition was not strongly different regarding seawater and submicron aerosol particles at the CVAO. Unless the oceanic transfer of GABA is very different compared to other hydrophilic amino acids, this pathhway does not explain the high abundance of GABA on the submicron aerosol aprticles at the CVAO.

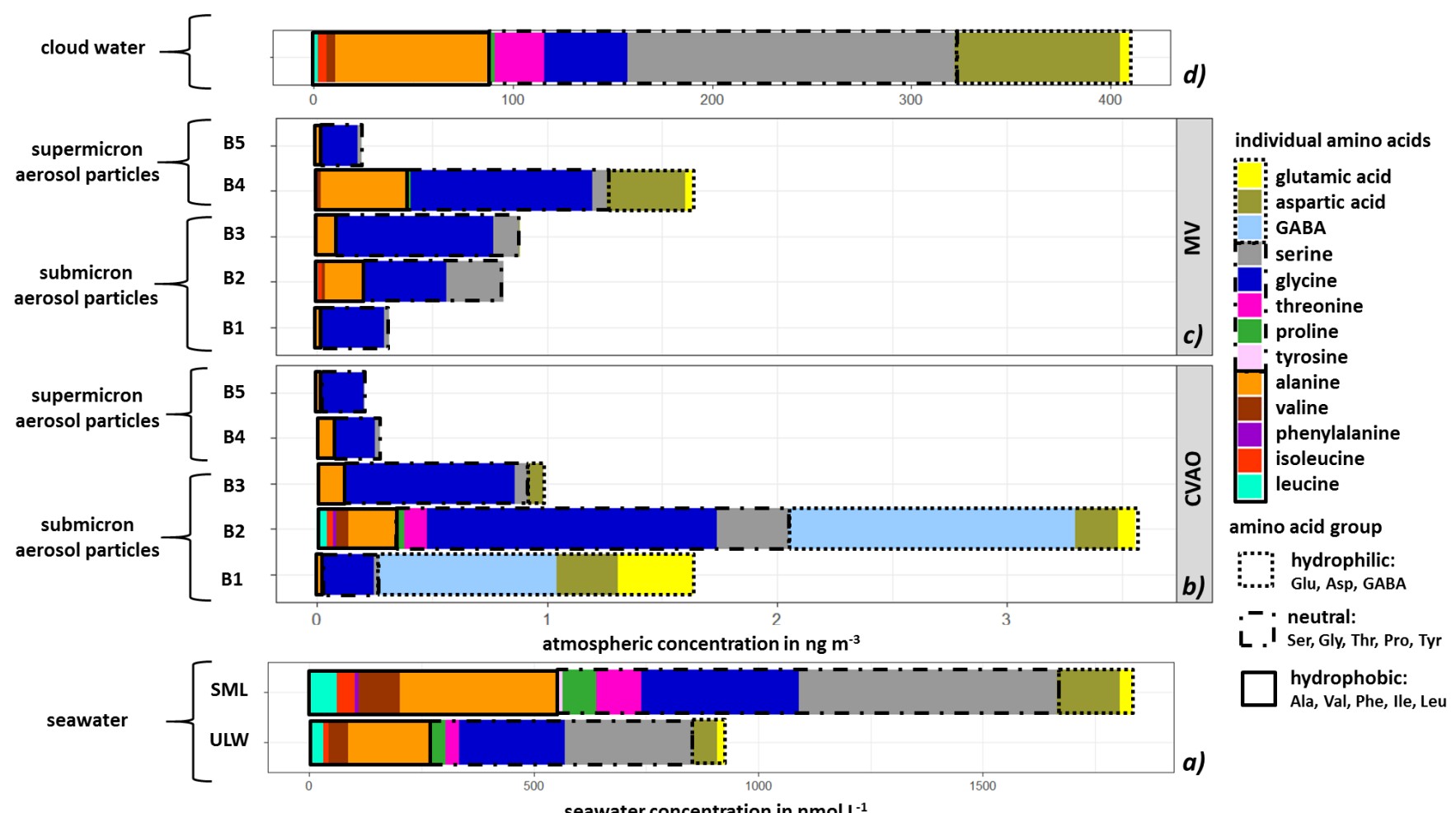

**Figure 4:** Case study: individual FAA concentration in a) seawater samples (ULW, SML) in nmol $L^{-1}$, in b) size-segregated aerosol particle samples at the CVAO and c) at the MV station (size range : 0-4 ng $m^{-3}$) and in d) cloud water sample (size range: 0-400 ng $m^{-3}$)

Together with the facts that GABA is a known indicator for the microbiological decomposition of OM (Dauwe et al., 1999;Engel et al., 2018), and microorganisms are known to be present on marine aerosol particles even in the submicron size range (Rastelli et al., 2017) the formation of GABA on the aerosol particles might be related to an in-situ formation. Interestingly, GABA was not detected in cloud water samples, although bacteria were found during the campaign in cloud water (van Pinxteren et al., 2020) whose presence has been reported in the literature (Jardine, 2009;Vaïtilingom et al., 2013;Jiaxian et al., 2019). It remains speculative whether GABA was degraded in cloud water despite its rather long lifetime (remote cloud case: 28.8 h, Table S13) or whether it was not produced by the bacteria in cloud water. Asp has been detected in all marine compartments and showed high cloud water concentrations. Correlations between Asp with diatoms and zooplankton have been reported for the marine environment (Hammer and Kattner, 1986). Hence, the occurrence of Asp in the marine environment can be attributed to a biogenic origin, whilst the high concentrations of Asp in cloud water (Fig. 3 & 4d) might be related to an oceanic source. In the Antarctic, Barbaro et al. (2015) attributed the hydrophilic amino acid fraction mainly to locally produced aerosol particles. According to the biogenic sources of the hydrophilic acids and their characteristics observed here, a local marine source for Asp and Glu, together with the biogenic formation of GABA on the aerosol particles the Cape Verde islands could be prevalent.

### 3.4.2 Neutral and hydrophobic amino acids

Neutral amino acids were generally the amino acid group with the highest concentration in all investigated marine compartments, accounting for more than 50% of the FAA total (Fig. 4a-d). Ser and Gly were the dominant representatives of this group. It is remarkable that especially the aerosol particles in the larger size range (e.g. supermicron aerosol particles: B4, B5) at both smapling stations are less complex in amino acid composition and almost exclusively dominated by Gly, folowed by Ser and Ala (Fig. 4b, 4c). Gly is discussed in the literature as a photochemical degradation product of other existing amino acids and this comparatively more stable amino acid (Gly) thus becomes a major component of the FAA composition (Barbaro et al., 2015). Compared to other amino acids, Gly and Ser have a very low atmospheric reactivity (McGregor and Anastasio, 2001) and therefore a higher mean lifetime τ (Gly: 0.48 h, Ser: 0.24 h; remote aerosol case, Table S13). Due to its atmospheric stability, Gly is proposed as an indicator for long-range transport (Barbaro et al. (2015) and references therein) and has a very low atmospheric reactivity (McGregor and Anastasio, 2001). However, our results clearly show that Gly and Ser are also present in seawater to a high extend, likely resulting from the siliceous exosceleton of diatom cell walls (e.g. Hecky et al. (1973)). Hence, besides long-range transport, a transfer from the ocean via bubble bursting might be an additional likely source of the stable, long-lived FAA in the atmosphere. The neutral amino acid Pro has been reported to be of biogenic origin in the marine environment and was detected in seawater (Fig. 4a), on submicron aerosol particles at the CVAO (Fig. 4b) and in cloud water (Fig. 4d). Fischer et al. (2004) demonstrated that Pro can be used to identify the presence of algal spores on aerosol particles and might thus be used as a tracer for an oceanic source. The presence of Pro in all marine compartments suggests a transfer from the ocean into the atmosphere up to cloud level. This is supported by the comparatively low atmospheric reactivity of Pro (remote aerosol case: 0.24 h, Table S13). Finally, the hydrophobic FAAs Ile, Leu and Thr were found in all

marine compartments in low concentrations. They are classified as relatively reactive amino acids and their abundance has been attributed to local or medium local sources consequently (e.g. Mashayekhy Rad et al. (2019)). Their low but constant abundance in all marine matrices again indicates a bubble-bursting transfer.

### 3.4.3 Aromatic amino acids

Aromatic FAAs as Phe and Tyr were present in seawater, but not on the aerosol particles , neither in cloud water samples. It could be assumed that these aromatic FAAs were either not transferred from the ocean into the atmosphere, or they reacted already after their transfer due to chemical transformation reactions, or they were not detected because of their low atmospheric concentration. The mean lifetimes $\tau$ of Phe (0.014 h) and Tyr (0.007 h) (Table S13) showed that both FAAs had a comparatively high atmospheric reactivity ($\tau < 1$ min) at remote aerosol case conditions. Hence, a rapid chemical reaction of these compounds is most likely. Moreover, previous studies reported low atmospheric concentrations of Tyr and Phe on aerosol particles. Barbaro et al. (2011) found Phe (0.5 ng m$^{-3}$) and Tyr (0.3 ng m$^{-3}$) with a contribution $< 1$ % to $\sum$FAA ($\sum$FAA: 42.5 ng m$^{-3}$) on TSP samples in urban background (Venice, Italy). In our study at the CVAO, the mean value of $\sum$FAA in PM$_{10}$ aerosol particles was 3.8 ng m$^{-3}$ (section 3.2). Assuming that Phe and Tyr were contributing to $\sum$FAA in a very small fraction as reported in Barbaro et al. (2011), their concentrations would be below the detection limit and could thus probably not be detected. It can be concluded that the aromatic FAAs could either not be quantified on aerosol particles due to the sensitivity of the analytical method used here or they react very quickly in the atmosphere and could therefore not be detected.

### 3.4.4 Transfer of amino acids from the ocean into the atmosphere

A high similarity regarding the FAA species within the different marine compartments could be observed, although some differences could also be identified (e.g. GABA). Together with the high concentration of ocean-derived compounds (Na$^+$, MSA) in the aerosol particles and cloud water, this indicates a coupling between the FAA in the ocean and the atmosphere. A quantitative metric for comparing compounds in the ocean and in the atmosphere is the EF$_{aer}$ (Eq. (2)). The concept is mainly applied to closed systems (e.g. Quinn et al. (2015), Rastelli et al. (2017)) because FAA formation or degradation pathways on the aerosol particles including biological or photochemical atmospheric reactions, and possible transport from other than marine sources are excluded in this parameter. However, for comparison purposes, it might be useful to calculate the EF$_{aer}$ also from open systems as done e.g. by Russell et al. (2010) or van Pinxteren et al. (2017). The averaged EF$_{aer}$ of $\sum$FAA in the individual Berner stages of the case study at the CVAO based on SML and ULW concentrations are shown in Fig. 5.

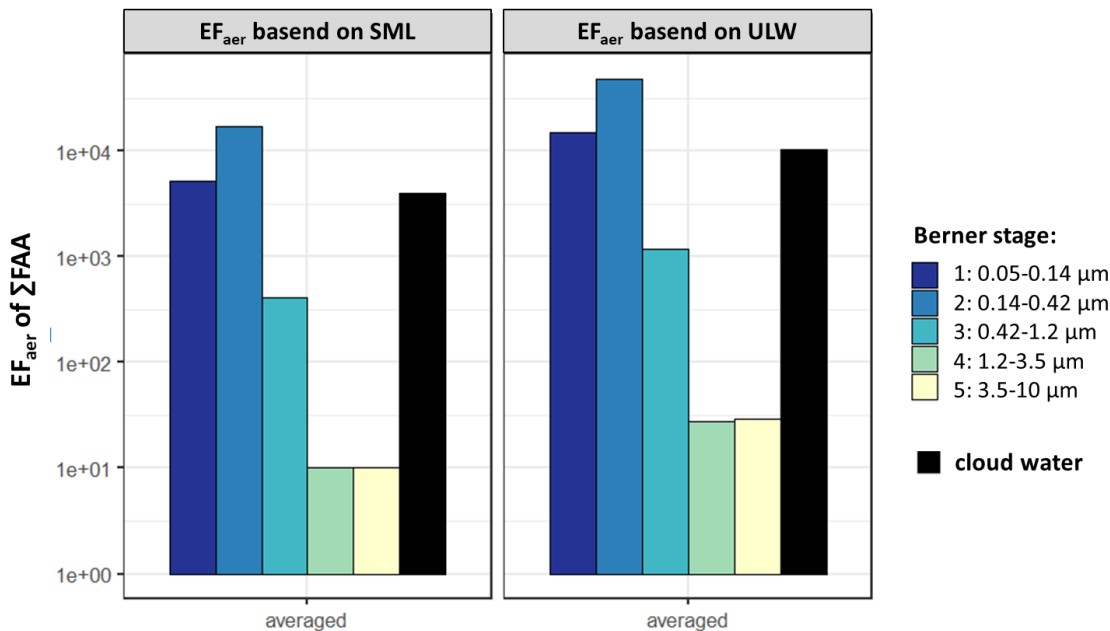

**Figure 5: The averaged aerosol enrichment factor (EF$_{aer}$) of ∑FAA in the size-segregated aerosol particle samples (Berner stage 1-5) at the CVAO and the cloud water enrichment factor (EF$_{CW}$ in black) based on SML (left) and on ULW (right) calculation (Eq. (2))**

The EF$_{aer}$ of ∑FAA, based on SML, were in the supermicron size range ($1·10^1$ (B5), $1·10^1$ (B4)) several orders of magnitude smaller than in the submicron range ($4·10^2$ (B3), $2·10^4$ (B2), $5·10^3$ (B1)). Furthermore, the calculated EF$_{aer}$, based on ULW, was up to one order of magnitude higher in the aerosol particles than the EF$_{aer}$, based on the SML. This is due to the different FAA concentrations in seawater (section 3.1), as the sodium values were very similar with 12.45 g L$^{-1}$ in the ULW and 12.53 g L$^{-1}$ in the SML. van Pinxteren et al. (2017) showed that the EF$_{aer}$ of the WSOC in the submicron marine ambient aerosol particles at the Cape Verdes ranged between $10^3$ and $10^5$. The averaged EF$_{aer}$ of the WSOC during our campaign in the submicron range was between $2·10^3$ and $1·10^4$ and between $3·10^2$ and $4·10^2$ in the supermicron range (Table S14) and in good agreement with van Pinxteren et al. (2017). Comparing the EF$_{aer}$ of ∑FAA ($1·10^1$-$2·10^4$) with the EF$_{aer}$ of WSOC ($1·10^1$-$2·10^4$) in the submicron range, both EF$_{aer}$ are in the same order of magnitude. Moreover, similar percentages of ∑FAA were observed for the DOC in the SML (up to 7.6%) (section 3.1) and for the WSOC in submicron aerosol particles (up to 5.3%) (section 3.2).

Previous studies have shown that OM ejected into the atmosphere during bubble bursting, results in the formation of sea spray aerosol particles containing OM similar to SML (Russell et al. (2010);Cunliffe et al. (2013) and references therein). Especially the film droplets have been reported to be enriched in OM and are suggested to transfer OM from the SML onto submicron aerosol particles (Wilson et al., 2015). The supermicron aerosol particles tend to form from the larger jet droplets and thus represent the ULW composition (Blanchard, 1975;Wilson et al., 2015). We cannot derive mechanistic transfer characterizations from the ambient measurements performed here. Nevertheless, the constant FAA enrichment in the SML

together with the strong FAA enrichment in the submicron aerosol particles strongly suggest that film droplets form the submicron particles. However, Wang et al. (2017) showed that jet drops (which transfer OM from the ULW) also have the potential to contribute significantly to the formation of submicron sea spray aerosol particles, so, jet droplets can also contribute to FAA formation.

Applying the concept of the enrichment factor to cloud water and calculating the $EF_{CW}$ (Eq. (2)), the $EF_{CW(\sum FAA)} = 4 \cdot 10^3$ (based on SML) and $1 \cdot 10^4$ (based on ULW) could be determined. As mentioned in section 3.3, several atmospheric processing (aging), oceanic transfer and biogenic-driven processes might contribute to this high enrichment and need to be addressed in future studies. The high FAA concentrations and enrichments might have implications on OM processing through clouds and are worth further studying.

## 4   Conclusion and Outlook

Concerted measurements i.e., simultaneous measurements of seawater, size-segregated aerosol particles and cloud water samples during the MarParCloud campaign at the CVAO and MV stations allowed to investigate FAAs on a molecular level, which are important contributors to marine OM. The similarities between the FAA composition in the seawater (SML) and on
the submicron aerosol particle samples, as described in section 3.4, indicated that a certain FAA contribution, in particular the hydrophilic amino acids Asp and Glu in the submicron aerosol particles at the CVAO, was probably caused by sea spray and might be transferred up to cloud level. The neutral and hydrophobic amino acids were also present in all marine compartments, suggesting some interconnections. Stable amino acids like Gly are often reported as long-range tracers, but their abundance in seawater and marine air masses prevailing during the sampling period suggest an (additional) oceanic source. The oceanic link
is supported by a high atmospheric concentration of ocean-derived compounds (sodium, MSA), a high fractional residence time of the air masses above water and a low-to-medium impact of other non-marine sources (based e.g. on the mass concentration of trace metals). In addition, some indications for the biological production of amino acids on the aerosol particles (GABA) were observed. Aromatic amino acids are either not transferred from the ocean into the atmosphere or react very quickly; in any case, they are present only in small concentrations close to the LOQ. By distinguishing between submicron
and supermicron aerosol particles, differences in the chemical composition of these aerosol particle size classes could be identified, which show a much higher complexity of the FAA composition in the submicron aerosol particles. FAAs were present in the size range for aerosol particles associated with CCN activity and cloud water, and might be connected to CCN activity due to their hygroscopicity and soluble character, but this effect was not investigated here and should be examined in future studies. In a simplified approach, considering only a possible transfer from the ocean onto the aerosol particles and
cloud water (neglecting e.g. atmospheric processing), the aerosol enrichment factor was calculated. A high FAA enrichment in the submicron aerosol particles of $EF_{aer(\sum FAA)}$: $2 \cdot 10^1$-$6 \cdot 10^3$ and a medium enrichment on supermicron aerosol particles $EF_{aer(\sum FAA)}$: $1 \cdot 10^1$-$3 \cdot 10^1$ were observed. Applying the same concept to cloud water, an enrichment of $4 \cdot 10^3$-$1 \cdot 10^4$ was obtained. The high FAA concentrations (11.2-489.9 ng m$^{-3}$) and enrichments in cloud water were reported here for the first time. Their composition, together with the high concentrations of inorganic marine tracers (sodium, MSA), indicate at least to some extend

an oceanic transfer and biogenic formation that remains subject to future work. Altogether, the varying composition of FAAs in the different matrices shows that their abundance and their enrichments in the SML and their atmospheric transfer are not determined by single environmental drivers (e.g. wind speed) and/or simple physico-chemical parameters (e.g. surface activity). The ocean-atmosphere transfer of FAAs is influenced by biotic and abiotic formation and degradation processes.

Further studies are required to unravel their drivers and understand their complex composition that, finally, have to be considered in OM transfer models. To the best of our knowledge, this study was the first that simultaneously analyzed the FAA in all marine compartments - seawater including the ULW and the SML, size-segregated aerosol particles and cloud water – in such detail to obtain indications on their sources and interconnections.

*Data availability.* The data are available through the World Data Centre PANGAEA under the following link: https://doi.pangaea.de/10.1594/PANGAEA.914220.

*Special issue statement.*

*Acknowledgements.* This work was funded by Leibniz Association SAW in the project "Marine biological production, organic aerosol particles and marine clouds: a Process Chain (MarParCloud)" (SAW-2016-TROPOS-2) and within the Research and Innovation Staff Exchange EU project MARSU (69089). The authors also thank Susanne Fuchs, Anett Dietze, Sontje Krupka, René Rabe and Anke Rödger for providing additional data and filter samples and all MarParCloud and MARSU project partners, especially Malena Manzi, for a good cooperation and support. Additional thanks to Khanneh Wadinga Fomba and

his support in context of mineral dust and cloud water analytics, to Thomas Schaefer regarding the kinetic analysis and to Tobias Spranger concerning data visualization. We further acknowledge the professional support provided by the Ocean Science Centre Mindelo (OSCM) and the Instituto do Mar (IMar).

*Author contributions.* NT wrote the manuscript with contributions from MvP, HH and AE. NT and MvP performed as part of

the MarParCloud campaign team the field sampling and NT the chemical measurements of amino acids. The chemical data evaluation was done by NT in consultation with HH and MvP and with the mentioned researcher in the acknowledgement. All authors discussed the results and further analysis after the campaign. All co-authors proofread and commented the manuscript.

*Competing interest.* The authors declare that they have no conflict of interest.

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
