# Peer review of "Concerted measurements of free amino acids at the Cape Verde Islands: High enrichments in submicron sea spray aerosol particles and cloud droplets"

_Atmospheric Chemistry and Physics, 2019_

## Referee Comment (RC1) · Anonymous Referee #3 · 21 May 2020

The manuscript of Triesch et al. focuses on the measurements of free amino acids (FAA) in different marine compartments (underlayer water samples, sea surface microlayer, size segregated aerosol samples at different heights, cloud water) at the Cape Verde Atmospheric observatory (CVAO) and at the Mt. Verde during September/October 2017. Further, through the case study authors discusses the possible transfer of specific FAA from the ocean to the atmosphere up to the cloud level. This work for the first time aims to provide a detail chemical analyses of FAA, of both the sea surface layers as well as of atmospheric samples (size segregated aerosols and cloud water), thus representing a promising approach to contribute to the fundamental state-of-the-art knowledge on the linkage between the ocean and atmosphere. In addition, this work reports on unique marine and atmospheric organic matter data from remote tropical areas, being rare in comparison to continental/coastal ones. Thus, my opinion is that this manuscript presents valuable data sets and after some revision, it will be definitely worth to publish.

**Major comments**

The authors should comment and discuss that some specific FAA were not detected in particular compartments possibly due to their high LOQ (in particular matrices) and the fact that maybe not enough material was available for their analysis. This is particularly relevant for the selected hydrophilic FAA in comparison to other FAA groups and should be considered when discussing the selective transfer of FAA groups from the ocean to the atmosphere.

The experiments have been done carefully and the quality of data is good. Authors comprehensively presented the experimental approach and obtained data within the manuscript. Important data are also shown within the SI material. However, some SI Tables are not easily comparable to each other. For example, in the present form Table S9 does not follow the Table S8 format. Comparing Tables S8 and S10, I am wondering why data for all size fractions are presented within Table S10 and only submicron and supermicron classification is done within the Table S8. I suggest to reorganize and to uniform SI Tables to follow the main text clearly and to enable the data comparison more easily.

Some sentences/paragraphs require major rephrasing. I found the reading of several statements rather unclear and I proposed some changes accordingly. However, my main concern lies in the sentence style used throughout the manuscript. I believe that the paper should be thoroughly edited.

Once introduced, abbreviations should be used further throughout the text.

**Specific and technical comments**

**Page 2**

L23 Skip …also..; it is confusing considering the previous paragraph

**Page 3**

L7-9 please rewrite;…into consideration *for the accurate prediction of marine organic matter transfer to the atmosphere* (van Pinxteren et al., 2017).

**Page 4**

L1-3 Repetition with the L27-34; I suggest to skip it or rewrite the overall paragraph to avoid repetition.

L14-15 Unclear/confusing sentence, please rewrite

L15 The abbreviations for the sampling sites (CVAO and MV stations) were introduced within the Introduction, thus please use it further throughout the text (e.g. .P4 L30, P5 L6, P12 L30…)

L19: I suggest avoiding abbreviations within the title (*Seawater sampling*). Types of samples could be specify within the following paragraph

L20: Please correct ..were taken *from* a fishing boat…

L22 and L23: Skip -pre-cleaned- as it is stated in L25-26 that all sampling material was pre -cleaned

**Page 5**

L16 I suggest to use the FAA abbreviation throughout the text

L22 Add…include *determination of* glycine (Gly)…

L34 Unclear sentence part (..in 0.1 min back to 5 % B and…); please rewrite

L34 Please correct: …This *analytics OR analytical procedure* can be used…

**Page 6**

L2-5 Unclear/confusing sentence, please rewrite

L6 Introduce the LOQ abbreviation here instead within L24

L13 Please add the method used for the trace metal determination

L13 Remove a space before -total-

L23 Rewrite to be clear that data obtained for the aerosol samples were blank corrected

**Page 7**

Please explain the calculation of $EF_{SML}$ based on LOQ/2 estimation (Figure 1) in the case of the missing ULW data within the 2.2.4 Section.

L28 I suggest to name the $EF_{aer}$ as the -aerosol enrichment factor- instead of the -enrichment factor aerosol- (see accordingly the Fig 5 caption)

**Page 8**

Authors should discuss if the observed variability of FFA concentrations in seawater samples (especially in the SML) was comparable to the variability of the overall DOC and/or TDN pools. It seems to me that high variability of DOC and/or TDN content actually caused the variability of their FAA constituents.

In the context of further FAA transfer discussion, it would be useful to follow the FFA classification into hydrophilic, neutral and hydrophobic groups as done for the atmospheric samples.

**Page 9**

L2 Add …Mediterranean Sea *of* 0.07-0.60 µmol L-1…

L4-6 Add …. in the SML or in the ULW *since* no correlation between…

L23-24 The sentence should be supplemented to indicate the particular oceanic regions with the comparable FFA levels as at the CVAO.

**Page 10**

L16 Add …of dissolved FAA ($PM_{10}$; sum of B1-5)…

L17-21 Please indicate the PM size fraction citing the Matsumoto and Uematsu (2005) as well as Barbaro et al, 2015

**Page 12**

Please explain why would the low percentage of hydrophobic FAA and higher percentages of hydrophilic FAA point to the local oceanic origin of FAA in aerosol size fractions. As marked previously, it would be useful to refer to the seawater FAA classification in this study.

L31-32 Unnecessary sentence, repetition

**Page 13**

L6-10 The PM mass concentrations of size segregated samples at the MV were substantially lower than those obtain at the CVAO. Thus, it could be expected that the levels of Na and MSA constituents will be accordingly lower at MV in comparison to CVAO. For the sake of comparison of aerosol tracer levels at different stations, it is more reasonable to consider the contribution of Na and MSA to the PM mass (of corresponding size fraction) instead of comparing the concentration levels.

L24-27 Authors should comment on the fact that high LOQ of particular FAA could resulted in their variabilities or selective determination in cloud water samples.

L28-32 to P14 L4-8 This paragraph deals with the possible dominant origin of FAA in the cloud water collected at the MV as within the 3.4.5. Section. I suggest skipping this paragraph

and combining the cloud water source discussion within the 3.4.5. Section to avoid the repetition.

L33-34 Please specify the study location of Gioda et al. (2009)

**Page 15**

The authors should comment and discuss that some specific FAA were not detected in particular compartments possibly due to their high LOQ (in all matrices) and/or the fact that not enough material was collected for their analysis. This is particularly relevant for selected hydrophilic FAA in comparison to other FAA classes/groups and should be considered when discussing the selective transfer of FAA groups from the ocean to the atmosphere.

**Page 17**

Although beyond the scope of this paper, I suggest discussing the potential connection of hydrophilic, hydrophobic and neutral FFA to CCN activity.

**Page 18**

L11-13 Please correct: For the calculated $EF_{aer}$, it should be noted that no further FAA formation or degradation pathways on the aerosol particles are considered, including biological or photochemical atmospheric reactions, *and* a possible transport from other than marine sources *is excluded*.

L14 remove -as-

**Page 17**

L7-9 Please correct: Previous studies showed that organic material ejected into the atmosphere during bubble bursting, *results* in the formation of sea spray aerosol particles containing similar organic…..

**Page 19**

L18 I suggest to change the title to -Origins of FFA in cloud water-, to be consistent with previous section

**SI material**

Page 2, L17 ..and *is* discussed in more…

---

## Referee Comment (RC2) · Matthew Pendergraft (Referee) · 19 Jun 2020

acp-2019-976 Submitted on 23 Oct 2019 Concerted measurements of free amino acids at the Cape Verde Islands: High enrichments in submicron sea spray aerosol particles and cloud droplets Nadja Triesch, Manuela van Pinxteren, Anja Engel, and Hartmut Herrmann

Generally, a referee comment should be structured as follows: an initial paragraph or section evaluating the overall quality of the preprint ("general comments"), followed by a

section addressing individual scientific questions/issues ("specific comments"), and by a compact listing of purely technical corrections at the very end ("technical corrections": typing errors, etc.).

In your evaluation, please take into account the different aspects mentioned on the ACP website at: https://urldefense.com/v3/__https://www.atmospheric-chemistry-and-physics.net/peer_review/review_criteria.html__;!!Mih3wA!RVxwT_2zwS6jjcQw8KkE9dEEIKQtZT2PEZGrh00gh-g3c8NhYkfIhXujsQz0_c9y$

General Comments

This paper describes the measurement of free amino acids in seawater (microlayer and underlying water), aerosol, and cloud water at Cape Verde Island. The types and abundances of individual amino acids present in different sample types are compared. Enrichment factors are calculated for amino acid concentrations between the the microlayer and underlying water, and between the water and aerosol. The contribution the amino acids make to the pools of organic carbon and total dissolved nitrogen in water and aerosol are presented.

The work largely presents an inventory of amino acids for this location and the given sample types. The concerted measurement of amino acids in underlying seawater, microlayer, aerosol, and cloud water are novel. The authors also mention that these are the first reported free amino acids concentrations for marine clouds. Enrichment factors between seawater and aerosol are useful for comparison but I urge the authors to elaborate a little on the caveats of their use in an open system, namely that it is unlikely that a high fraction of the aerosol (and cloud droplets) measured came from the location of the measured seawater (and even less likely they emerged from those waters when the water was sampled). Concentrations in the atmosphere are also subject to ageing. This differs from other work on sea spray aerosol in closed systems where all particles are emitted from the contained source waters and minimal ageing occurs on the nascent particles (at least upon collection).

[Figure]

This work could be strengthened by: 1. elaborating on the significance of amino acids in the ocean and atmosphere and how the scienific community benefits from their inventory. 2. investigating the drivers of their observations by further relationships amongst their datasets and incorporating other datasets, e.g. Did variations in measured aerosol volume drive variations in amino acid concentrations? Did primary productivity (via remotely sensed chlorophyll-a) influence FAA concentrations in water and air? 3. further exploration of commonalities amond and differences between their amino acid data across different sample types. 4. what implications can be drawn the analyses accomplished and those I suggest? Are there certain ratios of amino acids in seawater that hold constant in seawater samples but not in the aerosol and clouds? What could cause this? 5. better connecting to past work. How do the observations compare not just to the values reported in other work but to the conclusions drawn in other research? How do drivers of amino acid concentrations differ between the ocean and atmosphere? What are the surface activities of the amino acids and can they predict their transfer in sea spray aersosol?

Ultimately I would like the authors to demonstrate that they have done more than make some measurements and that we have gained new insight into the ocean-atmosphere system beyond an inventory of amino acids. The manuscript needs to set up what was done on top of our current understanding of sea spray aerosol formation and the transfer of different compounds/-classes into sea spray aersosol. By building on past research the present work is prepared to more clearly communicate its findings and to also contribute new knowledge to the field.

Introduction. page 3. line 22. Barbaro et al. (2015) investigated FAA in size-segregated Antarctic aerosol particles to gain information about FAA as possible tracers of primary biological production in Antarctic aerosol particles line 24. Although there are several studies in different marine regions, there is a lack of ambient measurements of FAA simultaneously in seawater and in size-segregated aerosol particles in the tropical Atlantic Ocean It would strengthen the justification for the work if it was stated why we

must know this information specifically for the Atlantic Ocean and atmosphere.

There is little effort made to investigate the drivers of amino acids data for the ULW and SML. The data are not plotted against other measurements. We are merely offered (page 9, line 19): "On the basis of previous studies, the transportation and the releasing mechanisms seems to be most likely for the observed enrichment of FAA." At this point, we were not given a strong motivation for the measurements and we haven't learned anything from them. Then it is explained that the data are in agreement with other datasets and then this one dataset is extrapolated to the entire North Atlantic Ocean. I wouldn't extrapolate data from one location to the entire North Atlantic Ocean.

Similar to the seawater measurements, the atmospheric research would be stronger if it explored the drivers of the amino acid data. Particularly lacking from the aerosol work are aerosol sizing measurements. In their absence, we do not know if the variability in amino acid concentrations in the air are driven by changes in particle concentrations. Similarly, amino acid concentrations in the air could be related to wind direction.

Other:

Did particle type vary across the different aerosol sampling periods?

If amino acids have already been measured in remote marine air/aerosol, how does the present manuscript advance our knowledge?

Specific Comments

Abstract. page 1. line 20. "The total concentration (PM10) was between 1.8–6.8'ng'm−3 and tended to increase during the campaign." Perhaps instead of "during the campaign", give the time period or relate to a potential/suspected driver of the observation.

Introduction.

page 2.

lines ∼8-12. The marine nitrogen cycle is alluded to but vaguely. "(T)he bulk DON pool CAN BECOME bioavailable" or "IS bioavailable"? What is remineralized nitrogen? Perhaps get straight to proteniaceous compounds and amino acids as being large members of the DON pool.

line 21. much more work has been done on amino acids as ice nucleating entities (and as antifreeze entities) than just Szyrmer and Zawadzki, 1997 and should be ackowledged because it gives importance to this work.

line 23. "Amino acids are also present and have been described in the marine environment." This has already been established. The first 3 sentences of this paragraph feel like we are going backwards. In the previous paragraph we went from the ocean to the atmosphere and now we are back to describing amino acids in the ocean.

line 28. (Engel et al., 2017) lacking other citations for the microlayer's importance in the ocean and sea spray aerosol.

page 3.

line 7. "However, chl-a concentration solely does not adequately describe the complete spectrum of biological activity (Quinn et al., 2014)" should reference a paper on marine microbiology, like Azam and Malfatti. 2007. Microbial Sctructuring of Marine Ecosystems. DOI: 10.1038/nrmicro1747

page 4.

line 13. "During this campaign, concerted measurements were performed including sampling of sizesegregated aerosol particles at the CVAO and seawater sampling at the ocean site (∼16°53ËĹ30ËĹN, ∼24°54ËĹ00ËĹËĹW). The location was carefully chosen with minimal influence of the island and located in wind direction to the CVAO" This second sentence is crucial to the rationale of the study and should be demonstrated. I'm surprised to not see a figure referenced here.

line 20. "The seawater samples were taken with a fishing boat, starting from Bahia

das Gatas, São Vicente." A study site map with water and aerosol locations and winds during sampling periods would be good to "connect the dots". Figure S1 does not do this.

page 6.

line 2. "Since no chiral column was used in the UHPLC separation, it is possible that not only L-amino acids, which were used as the standard, were quantified, but that the here presented concentrations were possibly quantified as the sum of the L- and D-amino acids." So why not simply report them as L- and D-amino acids instead of reporting them as L-amino acids but acknowledging that is not accurate?

2.2.4 Enrichment Factors

For calculating enrichment factors (EFs) between aerosol and the SML, I wonder if it would be more appropriate to first calculate the aerosol-ULW EF using equation 2 and then apply the SML-ULW EF (EF_SML) from equation 1 for that analyte. So there would be EFs for aerosol relative to the underlying water and relative to the SML and they would differ by the EF_SML (Eq.1). Ideally this would resolve the current inconsistency in invoking the Na+ concentration of the SML when comparing to aerosol to SML (Eq. 2) but not when comparing SML to ULW. The way it stands now, if you calculate aerosol EFs relative to the ULW and the SML they won't differ by the EF_SML and instead will differ by an EF_SML calculated using equation 2.

page 10.

line 1. This sentence is confusing: "In the study of Reinthaler et al. (2008) DFAA contributed with ∼12 % to DOC and with ∼ 30 % to dissolved organic nitrogen (DON) in the SML of the Atlantic ocean and the western Mediterranean Sea." Is it DFAA in seawater accounting for ∼12% of DOC (in seawater), and DFAA in SML accounting for ∼30% of DON in the SML?

line 28. The characterization of FAA into the hydrophyllic, hydrophobic, and neutral

classes is nice. What might be the drivers and implications?

page 11

line 11. "However, the presence of Glu, Asp and GABA as part of the hydrophilic species in the submicron aerosol particles (on 22/09/2017, 4/10/2017, 6/10/2017, 7/10/2017) strongly indicated a local oceanic origin." If amino acids can indicate aerosol type or source, this should be central to this work, explained in the introduction, and examined for each sample. Also, how did the amino acid profiles of the different sample types (and individual samples) compare? Were the same relative abundances consisten across the ULW, SML, and different aerosol size classes? Are all the amino acids measured commonly found in the ocean and are they exclusive to the ocean?

page 12

line 2. "Considering the amino acid classifications from Barbaro et al. (2015)), it can be concluded that the submicron aerosol particles with low averaged percentage of hydrophobic species (5 %) and higher percentages of hydrophilic species (4-55 %, mean of 15 %) could have local oceanic origin." Similar to previous comment. The different amino acid characterizations: hydropathy index of Kyte & Doolittle 1982; Pommie et al. 2004; and Barbaro et al. 2005 should be explained early on and would add value to the present work.

line 5. "This is supported by a predominant marine origin of the aerosol particles according to the air masses history, particulate MSA concentrations and MSA/sulfate ratios and particulate concentrations of dust tracers (Table S8)." It would be nice to lay this out because we make it this far into the manuscript wondering about the presence of aerosols from land in these samples. I would characterize the air masses early on. Do the back trajectories differ for the samples collected at 2 different elevations? And which back trajectories were used? Hysplit or Flexpart? Is this mentioned in the Experimental section?

[Figure]

line 7. "The higher complexity in the FAA composition on the submicron aerosol particles could only be determined because the analytical method applied here is able to quantify the individual molecular FAA species." Yes, that is good, and I would encourage the authors to leverage this resolution in their data. See previous comments regarding the comparison of amino acid "profiles" for different sample types and what do the presence of each amino acid tell us.

line 10. "The composition of FAA on the size-segregated aerosol particle samples with focus on the comparison of the submicron with the supermicron aerosol particles as well as the comparison of aerosol composition with the seawater composition will be discussed more detailed in section 3.4." Please simplify this sentence.

3.2.2 Size-segregated aerosol particles at the mountain station (MV) This seciton reports particulate matter (PM) masses for both the MV and CVAO stations. The Experimental section only reports particle volume measured for cloud water sampling at Mount Verde, and not at CVAO, with liquid water content (LWC) somehow derived. I am in favor of reporting what is measured directly. Here, is particle mass dervied from the particle volume measurements or from the mass of filter samples (at CVAO) and cloud water (at MV) recovered? I assume there was a particle volume monitor at both CVAO and MV. I encourage plotting all particle volume data against other particle measurements like FAA. Does FAA abundance track particle volume?

page 13

line 24. "In cloud water samples with $\Sigma$FAA <65 ng m-3, usually Gly was dominant followed by Ser. Cloud water samples with $\Sigma$FAA >290 ng m-3 showed a higher complexity in the FAA composition, especially towards the end of the campaign, including the appearance of Asp." Did the relative abundances of the FAA vary, indicating different FAA profiles, or were they similar, indicating a consistent FAA profile? What was the profile of hydrophobic, hydrophyllic, and neutral amino acids and how did that compare to the aerosol samples?

Were the sodium, sulfate, and MSA measurements made on cloud water also made on the aerosol samples? Other than FAA (in section 3.4) which other data are common to the two sample types - aerosol and cloud water - that would allow us to compare them?

page 15

line 10. "the reactivity/ mean life time $\tau$ of the amino acids" Please explain.

line 22. "The mean lifetime $\tau$ of Glu (remote aerosol case: 0.02 d" Thats 29 minutes. Is that considered long?

line 30. "The presence of GABA on the submicron aerosol particles pointed out that (marine) microorganisms were present on the aerosol particles and produced GABA via microbiological decarboxylation of Glu." Until the authors demonstrate that GABA cannot exist outside of a (marine) microorganism, this statement is unfounded. The opposite is a safe assumption: that any compound produced my marine microorganisms will also be found in the sewawater, either by adtive release by the living microorganism or via release of the dead microorganism (residence times will vary).

page 17

line 2. "the presence of bacteria in cloud waters has been reported in the literature (Jiaxian et al., 2019)." Microorganisms have been documented in the air since at least Darwin's HMS Beagle voyages so a few more citations here would be appropriate. A few of note:

Jardine, B. Between the Beagle and the barnacle: Darwin's microscopy, 1837-1854. Stud. Hist. Philos. Sci. Part A 40, 382–395 (2009).

Salisbury, J. H. On the Cause of Intermittent and Remittent Fevers. Am. J. Med. Sci. 51–75 (1866).

M. Vaïtilingom et al., Potential impact of microbial activity on the oxidant capacity and organic carbon budget in clouds. Proceedings of the National Academy of Sciences of

the United States of America. 110, 559–564 (2013).

3.4.2 Neutral and hydrophobic amino acids How do the surface activities vary across the different amino acids? Does abundance in aerosol correlate to surface activity? Or a combination of surface activity and reactivity/lifetime?

page 18.

line 13. "a possible transport from other than marine sources is included in this parameter." The language is (not on purpose) vague and should clearly state that the waters measured and used for the enrichment factors have not been demonstrated to be the source of the aerosols measured.

line 16. "Regarding the transfer of OM from the ocean into ambient aerosol particles, solely organic carbon as a sum parameter has been regarded to date and no distinction of single organic matter classes for ambient measurements has been performed."

Although not written altogether clearly, this statement seems to state that compound classes nor compounds have been resolved from ambient aerosol. This is false as there is a number of studies that have accomplished this. See:

Molecular diversity of sea spray aerosol particles: Impact of ocean biology on particle composition and hygroscopicity RE Cochran, O Laskina, JV Trueblood, AD Estillore, HS Morris, ... Chem 2 (5), 655-667

Quinn, P. K., Collins, D. B., Grassian, V. H., Prather, K. A., & Bates, T. S. (2015). Chemistry and Related Properties of Freshly Emitted Sea Spray Aerosol. Chemical Reviews. American Chemical Society. https://doi.org/10.1021/cr500713g

Figure 5. Please include in the figure the size range for each Berner stage.

page 19.

line 7. "Previous studies showed that organic material ejected into the atmosphere during bubble bursting, resulting in sea spray aerosol particles containing similar organic

material to that of the SML (Russell et al. (2010);Cunliffe et al. (2013) and references therein)." This - the basics of sea spray aerosol formation - need to be brought up later and the investigation of hydrophyllic/-phobic amino acids in differnt particle types needs to be established in this context.

line 29. "In situ-formation of FAA in cloud water, maybe due to biogenic formation or enzymatic degradation of proteins, selective enrichment processes as well as pH de-pendent chemical reactions might be potential sources." site Malfatti, F., Lee, C., Tinta, T., Pendergraft, M. A., Celussi, M., Zhou, Y., ... Prather, K. A. (2019). Detection of Active Microbial Enzymes in Nascent Sea Spray Aerosol: Implications for Atmospheric Chemistry and Climate. Environmental Science and Technology Letters, 6(3), 171–177. https://doi.org/10.1021/acs.estlett.8b00699

Technical Corrections

Introduction.

page 2. line 8. "surface global ocean" to "global surface ocean" line 16. "utilizable sources of nitrogen" to "utilizable FORMS of nitrogen"

page 3. line 31. "underline seawater" to "underlying seawater"

page 4. line 4. "as proxies or tracer" to "as proxies or tracerS"

In the Experimental section perhaps change "analytics" to "analyses"

2.2.1 Seawater sample analytics Was the standard addition method applied to samples to assess for recovery efficiency of the entire process?

page 6. line 23. "All here presented values" to "All values presented here" line 29. "The cloud water samples were operated the same as seawater samples". change "operated" to "handled" or "processed". line 31. "syringe filters filters" to "syringe filters" or "syringe tip filters"

page 10. line 12. Remove "It is obvious that" as it is confusing language. line 14-15.

Reword. Use "neither" instead of "both".

page 12 line 32. "aerosol particles" is redundant. aerosols are particles. say just "aerosol" or "particle", here and elsewhere.

page 17. line 7. "Neutral amino acis" to "acids" line 10. "A further explanation approach" remove "approach"

———————————————————

---

## Author Comment (AC1) · 12 Sep 2020

We thank the reviewer for the careful examination of the manuscript and the supporting information. In the docment "Author's response_ACP-2019-976-RC1", please find a point-by-point response to the questions and concerns. All references to the manuscript (e.g. page and line numbers) listed in our replies refer to the clean version of the manuscript (without track changes).

Please also note the supplement to this comment:

[Figure]

https://acp.copernicus.org/preprints/acp-2019-976/acp-2019-976-AC1-supplement.pdf

[Figure]

**Supplement:**

The manuscript of Triesch et al. focuses on the measurements of free amino acids (FAA) in different marine compartments (underlayer water samples, sea surface microlayer, size segregated aerosol samples at different heights, cloud water) at the Cape Verde Atmospheric observatory (CVAO) and at the Mt. Verde during September/October 2017. Further, through the case study authors discusses the possible transfer of specific FAA from the ocean to the atmosphere up to the cloud level. This work for the first time aims to provide a detail chemical analyses of FAA, of both the sea surface layers as well as of atmospheric samples (size segregated aerosols and cloud water), thus representing a promising approach to contribute to the fundamental state-of-the-art knowledge on the linkage between the ocean and atmosphere. In addition, this work reports on unique marine and atmospheric organic matter data from remote tropical areas, being rare in comparison to continental/coastal ones. Thus, my opinion is that this manuscript presents valuable data sets and after some revision, it will be definitely worth to publish.

We thank the reviewer for the careful examination of the manuscript and the supporting information. In the following, please find a point-by-point response to the questions and concerns. All references to the manuscript (e.g. page and line numbers) listed in our replies refer to the clean version of the manuscript (without track changes).

**Major comments**

*R#1-1)* The authors should comment and discuss that some specific FAA were not detected in particular compartments possibly due to their high LOQ (in particular matrices) and the fact that maybe not enough material was available for their analysis. This is particularly relevant for the selected hydrophilic FAA in comparison to other FAA groups and should be considered when discussing the selective transfer of FAA groups from the ocean to the atmosphere.

We agree with the reviewer's comment that a more detailed discussion of the LOQ of the individual analytes is needed and revised several parts of the manuscript accordingly.
In the SI (Table S1), we listed the LOQ of each FAA in the respective matrices (seawater, cloud water, aerosol particles). In the manuscript, we referred to these LOQs in a number of different places as described in the following:
In section "2.2.1 Seawater analyses" (page 6, line 20-23), we added: "The LOQs were mostly below 10 nmol $L^{-1}$, however, GABA and Met exhibited LOQs with 24.2 nmol $L^{-1}$ and 16.8 nmol $L^{-1}$, respectively (due to high blank values). A quantification of some FAAs in seawater, mainly in the ULW with its generally lower FAA concentrations compared to the SML, is therefore partly limited."
In addition, we mentioned that "The limit of quantification (LOQ) of the individual FAAs in seawater samples is in good agreement with the FAA analysis in seawater samples (e.g. Kuznetsova et al. (2004)) and listed in Table S1" (page 6, line 19-20)
In section "2.2.2 Aerosol particle filter analyses" (page 7, line 6-8), we added: "Although a variance in LOQs between the individual FAAs is apparent, FAAs with relatively high LOQs (39.5 pg $m^{-3}$) on aerosol particles such as Ala, GABA, Asp in submicron and supermicron aerosol particles could be quantified (as discussed in section 3.2 and 3.4)."
In section "2.2.3 Cloud water analyses" (page 7, line 31-33), we added "Since the LOQs of the FAAs in cloud water are below 0.3 ng $m^{-3}$ and often below 0.06 ng $m^{-3}$, a limitation of the FAA composition in cloud water due to the LOQs is rather unlikely despite the variance of FAA concentrations (11.2-489.9 ng $m^{-3}$) in cloud water (section 3.3)."
This possible restriction of the results by the LOQ was also discussed in the results and discussion section in the individual sections on seawater (section 3.1), aerosol particles (section 3.2), cloud water (section 3.3) and concerted measurements (section 3.4).

In section "3.1 Seawater samples" (page 10, line 1-3), we added "As discussed in section 2.2.1, GABA and Met have the highest LOQs of the analytical method used here, which may be one reason why these two analytes could not be quantified in the seawater samples (ULW and SML)."
In section "3.4.1 Hydrophilic amino acids" (page 18, line 19/20), we added "Although some limitations of GABA quantification (Table S1) in seawater exist, a relatively major abundance of it in seawater would not be restricted due to the high FAA concentrations."
In section "3.4.3 Aromatic amino acids" (page 21 ,line 1-3), we added "Assuming that Phe and Tyr were contributing to ∑FAA in a very small fraction as reported in Barbaro et al. (2011), their concentrations would be below the detection limit and could thus probably not be detected."

***R#1-2)*** The experiments have been done carefully and the quality of data is good. Authors comprehensively presented the experimental approach and obtained data within the manuscript. Important data are also shown within the SI material. However, some SI Tables are not easily comparable to each other. For example, in the present form Table S9 does not follow the Table S8 format. Comparing Tables S8 and S10, I am wondering why data for all size fractions are presented within Table S10 and only submicron and supermicron classification is done within the Table S8. I suggest to reorganize and to uniform SI Tables to follow the main text clearly and to enable the data comparison more easily.

We thank the reviewer for his positive judgement and his comment. We agree regarding the SI Tables. In order to simplify the comparability of the individual information in the SI, the SI tables (Table S2, S7, S8, S10 and S11) have been standardized in the distinction between submicron and supermicron aerosol particles. This provides a better overview of the measurement data and allow a clearer and easier to understand discussion.

***R#1-3)*** Some sentences/paragraphs require major rephrasing. I found the reading of several statements rather unclear and I proposed some changes accordingly. However, my main concern lies in the sentence style used throughout the manuscript. I believe that the paper should be thoroughly edited.

Following this comment, the manuscript and supporting information has been carefully revised by a professional English-speaking person. The main focus was on the comprehensibility of the sentences/paragraphs. In this context, the specific and technical comments were also implemented.

***R#1-4)*** Once introduced, abbreviations should be used further throughout the text.

We agree with the reviewer's comments regarding the abbreviations. Some of these are taken up again and explained in more detail in the following section "specific and technical comments" for abbreviations such as CVAO, MV and FAA. When revising the manuscript and the Supporting Information, care was taken to ensure that abbreviations (e.g. CVAO, MV, FAA) were used consistently after their introduction. A detailed overview can be found in the change tracking of both scripts attached.

**Specific and technical comments**

***R#1-5)*** **Page 2**
L23 Skip …also..; it is confusing considering the previous paragraph

Due to the revision of the introduction and also in line with the suggestions from reviewer 2, we concentrated on the FAA from the beginning and deleted general parts (as the one mentioned by the reviewer) from the manuscript.

***R#1-6)* Page 3**

L7-9 please rewrite;…into consideration *for the accurate prediction of marine organic matter transfer to the atmosphere* (van Pinxteren et al., 2017).

As mentioned above, due to the revision of the introduction and also following the suggestions from reviewer 2, this part was deleted from the manuscript.

***R#1-7)* Page 4**

L1-3 Repetition with the L27-34; I suggest to skip it or rewrite the overall paragraph to avoid repetition.

We agree with the reviewer's comment and have removed the detailed information on the sampling sites of the campaign in the introduction and concentrated on the different sampling approaches for seawater, aerosol particles and cloud water. The text now reads as follows in the Introduction (page 3, line 31 – page 4, line 5): "So, the aim of the present study is to investigate the occurrence of FAA in the marine environment regarding all important compartments; i.e. the ULW, the SML, the aerosol particles and finally cloud water in the remote tropical North Atlantic Ocean at the Cape Verde Atmospheric Observatory (CVAO). Their abundance, origin and possible transfer from the seawater as well as their transport within the atmosphere are studied in particular. Therefore, the FAA are measured on a molecular level and divided into hydrophilic (glutamic acid, aspartic, GABA), neutral (serine, glycine, threonine, proline, tyrosine) and hydrophobic compounds (alanine, valine, phenylalanine, isoleucine, leucine) according to their hydropathy index. Especially the similarities and differences between the amino acid composition in submicron (0.05-1.2 µm) and supermicron (1.2-10 µm) aerosol particles are elucidated. Finally, the potential of individual FAA as proxies or tracers for specific sources of aerosol particles and cloud water in the tropical marine environment is outlined."

***R#1-8)* L14-15 Unclear/confusing sentence, please rewrite**

We agree with the comments of the reviewer and have reworded the sentence and also added a figure of the sampling locations based on the comments of reviewer 2. It now reads (page 4, line 16-19): "During this campaign, concerted measurements were performed including the sampling of size-segregated aerosol particles at the CVAO and seawater sampling at the ocean site (~16°53′17′N, ~24°54′25′E). The location was carefully chosen with minimal influence of the island and located in wind direction to the CVAO as shown in Fig. S1."

***R#1-9)* L15 The abbreviations for the sampling sites (CVAO and MV stations) were introduced within the Introduction, thus please use it further throughout the text (e.g. .P4 L30, P5 L6, P12 L30…)**

We agree with the reviewer's comment and following the introduction we now use the abbreviations CVAO and MV.

***R#1-10)* L19: I suggest avoiding abbreviations within the title (*Seawater sampling*). Types of samples could be specify within the following paragraph**

Following the advice of the reviewer, we have changed the title of section 2.1.1 to "Seawater sampling" (page 4, line 23). As noted by the reviewer, the two collected seawater sample types (SML and ULW) are described in more detail in this section.

***R#1-11)* L20: Please correct ..were taken *from* a fishing boat…**

The sentence (page 4, line 24) was changed to "The seawater samples were taken from a fishing boat, starting from Bahia das Gatas, São Vicente."

*R#1-12)* L22 and L23: Skip -pre-cleaned- as it is stated in L25-26 that all sampling material was pre -cleaned

Following this comment, we have omitted the additional description of the sample bottles by "pre-cleaned". The text is now written (page 4, line 25-28) as "The surface films adhered to the surface of the glass plate and were removed with Teflon wipers directly into a bottle. This glass plate approach is described in detail by Cunliffe (2014). The ULW was sampled in a depth of 1 m into a plastic bottle fitted on a telescopic rod. To avoid influences from the SML, the bottles were opened underwater at the intended sampling depth."

**Page 5**
*R#1-13)* L16 I suggest to use the FAA abbreviation throughout the text

As mentioned in comment R#1-4, we agree to the continuous use of the abbreviation "FAA" and have integrated it in the manuscript and in the Supporting Information.

*R#1-14)* L22 Add…include *determination of* glycine (Gly)…

Following the comment of the reviewer, we included "determination of". It is now written (page 6, line 1-4) as "Besides, the FAA analysis includes the determination of glycine (Gly), L-alanine (Ala), L-serine (Ser), L-glutamic acid (Glu), L-threonine (Thr), L-proline (Pro), L-tyrosine (Tyr), L-valine (Val), L-phenylalanine (Phe), L-aspartic acid (Asp), L-isoleucine (Ile), L-leucine (Leu), L-methionine (Met), L-glutamine (Gln) and γ-aminobutyric acid (GABA) (purity ≥ 99 %, Sigma-Aldrich, St. Louis, Missouri, USA)."

*R#1-15)* L34 Unclear sentence part (..in 0.1 min back to 5 % B and…); please rewrite

We agree with the comment of the reviewer and rewrote this sentence part (page 6, line 12-14). Now it reads: "The flow rate of the eluent was 0.3 mL min 1 and the eluent gradient program was 5 % B for 1 min, 5 % B to 100 % B in 16 min, 100 % B for 2 min constant, in 0.1 min from 100 % B to 5 % B and the 5 % B was then kept constant for 3.9 min."

*R#1-16)* L34 Please correct: …This *analytics OR analytical procedure* can be used…

The sentence was corrected to "This analytical procedure can be used for amines, too, as described in van Pinxteren et al. (2019)." (page 6, line 14/15)

**Page 6**
*R#1-17)* L2-5 Unclear/confusing sentence, please rewrite

We agree with the comment of the reviewer and rewrote this sentence (page 6, line 15/16), which now reads: "Since no chiral column was used in the UHPLC separation, we cannot differentiate between L- and D- amino acids in our ambient samples."

*R#1-18)* L6 Introduce the LOQ abbreviation here instead within L24

Following the comment, we introduced the LOQ abbreviation on page 6, line 19/20. It is now written as: "The limit of quantification (LOQ) of the individual FAAs in seawater samples is in good agreement with the FAA analysis in seawater samples (e.g. Kuznetsova et al. (2004)) and listed in Table S1."

Moreover, we used only the LOQ abbreviation on page 7, line 4-6: "The LOQs of the individual FAAs in aerosol particle samples are listed in Table S1 and are in good agreement with the sensitivity of other analytical methods for FAAs in aerosol particles (e.g. Matsumoto and Uematsu (2005))."

*R#1-19)* L13 Please add the method used for the trace metal determination

The method for the trace metal determination was described in more detail on page 7, line 9-11: "The analysis of mineral dust tracers on nucleopore foils sampled with the Berner impactor was performed with the Total Reflection X-Ray Fluorescence S2 PICOFOX (Bruker AXS, Berlin, Germany) equipped with a Mo-X-ray source on polished quartz substrates as can be seen in Fomba et al. (2013)."

*R#1-20)* L13 Remove a space before -total-

We agree with the comment of the reviewer and removed the space before -total-. On page 6, line 27-29 it is written now: "The aqueous particle extracts were divided into aliquots for the analysis of water-soluble organic carbon (WSOC)/total dissolved nitrogen (TDN), inorganic ions and amino acids."

*R#1-21)* L23 Rewrite to be clear that data obtained for the aerosol samples were blank corrected

Following the comment, we rewrote this sentence. Now it reads: All values presented here for aerosol particle samples are field blank corrected." (page 7, line 4)

*R#1-22)* **Page 7**
Please explain the calculation of $EF_{SML}$ based on LOQ/2 estimation (Figure 1) in the case of the missing ULW data within the 2.2.4 Section.

To better describe this procedure, we added in section "2.2.4 Enrichment factors" an explanation for the calculation of $EF_{SML}$ based on LOQ/2 in the case of missing ULW data. Now it reads (page 8, line 11-14): "The FAA concentration in the ULW was assumed to be based on the concentration (LOQ/2) of individual amino acids for seawater samples from the same campaign day when individual FAA could be quantified in the SML samples, but not in the corresponding ULW ones due to FAA values below the LOQs (listed in Table S1). For the calculation of this estimated $EF_{SML}$, specially marked in the following, the concentration 25.2 nmol $L^{-1}$ was used for $c(analyte)_{ULW}$ in equation (1)."

*R#1-23)* L28 I suggest to name the $EF_{aer}$ as the -aerosol enrichment factor- instead of the -enrichment factor aerosol- (see accordingly the Fig 5 caption) 3

Following the comment of the reviewer we changed the name of $EF_{aer}$ to aerosol enrichment factor throughout the manuscript.

*R#1-24)* **Page 8**
Authors should discuss if the observed variability of FFA concentrations in seawater samples (especially in the SML) was comparable to the variability of the overall DOC and/or TDN pools. It seems to me that high variability of DOC and/or TDN content actually caused the variability of their FAA constituents.

To put the FAA data in context with the DOC/TDN data, we included the DOC/TDN data in Figure 1 and we evaluated a possible connection between FAA variability in seawater (ULW and SML) and DOC and TDN variability in seawater (ULW and SML). However, no statistical relevant correlation/link between FAA and DOC or TDN in terms of variability was found. The study of the

correlation of variability (FAA and DOC/TDN) was included in the revised manuscript (page 10, line 12-14) as follows: "Nevertheless, the variance of the ∑FAA concentrations in the SML or ULW observed here could neither be explained by the variance of DOC or TDN values, nor by wind speed and chl-*a* concentrations (see Fig. 1, Table S2, S5), since no correlation between these parameters and the concentration or enrichment of FAA was found."

*R#1-25)* In the context of further FAA transfer discussion, it would be useful to follow the FFA classification into hydrophilic, neutral and hydrophobic groups as done for the atmospheric samples.

We agree with the reviewer's comment and have extended the FAA's illustrations (Fig. 1, 2, 3, 4) and discussion in the different marine compartments not only to the individual amino acids but also to the amino acid groups.
In section "3.1 Seawater samples" (page 10, line 4-5) the text now reads: "Looking at the percentage composition within the ULW (10.1 % hydrophilic, 57.0 % neutral, 32.8 % hydrophobic) and the SML (10.6 % hydrophilic, 61.7 % neutral, 27.7 % hydrophobic), the values are similar to each other."
In section "3.3 Cloud water samples" the sentence "In terms of the hydropathy classification, the first part of the campaign (27/09/2017-5/10/2017) was dominated by neutral FAAs, whereas a sudden increase of the hydrophilic FAAs was observed in its second part (06/10/2017-08/10/2017)." was added on page 16, line 18 – page 17, line 2.

**Page 9**
*R#1-26)* L2 Add …Mediterranean Sea *of* 0.07-0.60 µmol L-1…

We agree with the reviewer's comment and added 'of'. Now it reads (page 11, line 3-5): "Reinthaler et al. (2008) considered concentrations of dissolved FAA of 0.02-0.13 µmol L$^{-1}$ (ULW) and of 0.43-11.58 µmol L$^{-1}$ (SML) in the subtropical Atlantic Ocean as well as values of 0.07-0.60 µmol L$^{-1}$ (ULW) and of 0.77-3.76 µmol L$^{-1}$ (SML) in the western Mediterranean Sea."

*R#1-27)* L4-6 Add …. in the SML or in the ULW *since* no correlation between…

The reviewer's comment was implemented in the new sentence. It now reads (page 10 line 12-14): "Nevertheless, the variance of the ∑FAA concentrations in the SML or ULW observed here could neither be explained by the variance of DOC or TDN values, nor by wind speed and chl-*a* concentrations (see Fig. 1, Table S2, S5), since no correlation between these parameters and the concentration or enrichment of FAA was found."

*R#1-28)* L23-24 The sentence should be supplemented to indicate the particular oceanic regions with the comparable FFA levels as at the CVAO.

We agree with the comment and have specified the region of study accordingly.
In section "2.1 Study area" the following sentences (page 4, line 13-16) were inserted: "In accordance with the classification of Longhurst (2007), the ocean around the Cape Verde Islands belongs to the region "North Atlantic Tropical Gyral Province (NATR)", which is described as the region with the lowest surface chlorophyll in the North Atlantic Ocean having a greater annual variability than seasonality."
In section "3.1 seawater samples" the discussion (page 10, line 32- page 11, line 7) reads as follows: "Altogether, it can be concluded that there is some variability within the FAA concentration in the SML and in the ULW, with a clear trend of its strong enrichment in the SML. The fact that the FAA concentrations were in accordance with the ones measured at the same location in 11/2013 (0.64 µmol L$^{-1}$, Table S4), supports the suggestion that the FAA concentrations reported here can be considered representative of the NATR region as part of the North Atlantic Ocean. These

concentrations are generally similar comparing them to FAA concentrations in other marine regions (Kuznetsova and Lee, 2002;Reinthaler et al., 2008). Reinthaler et al. (2008) considered concentrations of dissolved FAA of 0.02-0.13 µmol L$^{-1}$ (ULW) and of 0.43-11.58 µmol L$^{-1}$ (SML) in the subtropical Atlantic Ocean as well as values of 0.07-0.60 µmol L$^{-1}$ (ULW) and of 0.77-3.76 µmol L$^{-1}$ (SML) in the western Mediterranean Sea. Consequently, the FAA concentrations in the NATR region, with its very low surface chlorophyll and a greater annual variability than seasonality, are in the same order of magnitude compared to other marine regions (i.e. subtropical Atlantic and western Mediterranean Sea (Reinthaler et al., 2008))."

**Page 10**
*R#1-29)* L16 Add …of dissolved FAA (*PM10*; sum of B1-5)…

This sentence was omitted in the revision of the manuscript. The FAA's concentration discussion is now more focused on the distinction between submicron and supermicron aerosol particles and reads as follows (page 12, line 3-7): "In the submicron aerosol particles, the concentration of ∑FAA was between 1.3 ng m$^{-3}$ (1/10/2017) and 6.3 ng m$^{-3}$ (7/10/2017). Whilst the concentration ∑FAA varied between 0.2 ng m$^{-3}$ (6/10/2017) and 1.4 ng m$^{-3}$ (22/09/2017) in the supermicron size range, the highest atmospheric concentrations of ∑FAA were found in the submicron aerosol particles (mean of 3.2 ng m$^{-3}$) compared to the supermicron ones (mean of 0.6 ng m$^{-3}$)."

*R#1-30)* L17-21 Please indicate the PM size fraction citing the Matsumoto and Uematsu (2005) as well as Barbaro et al, 2015

Following the comment of the reviewer we defined the PM size fraction of both studies, Matsumoto and Uematsu (2005) and Barbaro et al, 2015, in more detail. Now it reads (page 12, line 23-28): "Matsumoto and Uematsu (2005) found averaged total concentrations of dissolved FAA with 4.5 ng m$^{-3}$ on aerosol particles (average of < 2.5 µm and > 2.5 µm) in the western North Pacific Ocean. Moreover, Wedyan and Preston (2008) observed an average concentration of dissolved FAA of 2.5 ng m$^{-3}$ on total suspended particles (TSP) during a transect ship cruise in the Atlantic Ocean. For Antarctic aerosol particles, the observed mean total FAA concentration on size-segregated aerosol particle samples (< 0.49-10 µm) at the Mario Zucchelli Station was 4.6 ng m$^{-3}$ (Barbaro et al., 2015)."

**Page 12**
*R#1-31)* Please explain why would the low percentage of hydrophobic FAA and higher percentages of hydrophilic FAA point to the local oceanic origin of FAA in aerosol size fractions. As marked previously, it would be useful to refer to the seawater FAA classification in this study.

This part was thoroughly revised.
In the Introduction (page 3, line 17-20), we stated "This divides them into hydrophilic, neutral and hydrophobic amino acids as discussed in Barbaro et al. (2015) for FAA in Antarctic aerosol particles. They also observed that hydrophilic FAA in the Antarctic were predominant in locally produced marine aerosol particles, while hydrophobic amino acids prevailed in aerosol particles collected at the continental station."
We intensively discussed the single FAA contributing to the hydrophilic fraction and the text now reads (page 14, line 21-27): "Following this hydropathy classification, the submicron aerosol particles consisted on average of 5 % hydrophobic, 15 % hydrophilic and 80 % neutral amino acids, while the supermicron aerosol particles contained on average only 7 % hydrophobic and 93 % neutral amino acids (Table S7). During the campaign, an increase in the contribution of hydrophilic amino acids was observed with a maximum of 55 % on 7/10/2017. Barbaro et al. (2015) reported that hydrophilic components were predominant (60 %) in locally produced marine Antarctic aerosol particles, whereas hydrophobic compounds were rather dominate aerosol particles collected at the continental station (23 % and 27 %). According to the conclusions by Barbaro et al. (2015), the

relatively high content of hydrophilic FAA found here points at least at some influence of local oceanic sources."

***R#1-32)*** L31-32 Unnecessary sentence, repetition

Following the comment of the reviewer, we removed that sentence.

**Page 13**
***R#1-33)*** L6-10 The PM mass concentrations of size segregated samples at the MV were substantially lower than those obtain at the CVAO. Thus, it could be expected that the levels of Na and MSA constituents will be accordingly lower at MV in comparison to CVAO. For the sake of comparison of aerosol tracer levels at different stations, it is more reasonable to consider the contribution of Na and MSA to the PM mass (of corresponding size fraction) instead of comparing the concentration levels.

We strongly revised this part and following the suggestions of the reviewer, we included and discussed ratios of FAAs to other aerosol particle constituents (PM, $Na^+$, MSA, WSOC) of the corresponding size fractions regarding CVAO and MV. The section (page 15, line 23-32) now reads: "The particles at the MV exhibited lower particle masses, as well as lower concentrations of the aerosol particle constituents. The decrease in concentrations of $\sum$FAA, PM, sodium, MSA and WSOC was reduced by a factor of three to four regarding the submicron aerosol particles. However, no uniform depletion ratio between their concentration at the CVAO and the MV was found for the supermicron aerosol particles (Table S11). While the PM of the supermicron particles was reduced by a factor of four at the MV (similar to the submicron aerosol particles), sodium and WSOC were depleted more strongly (factor of 11-12) compared to their respective concentrations at the CVAO. This suggests that the submicron particles were rather uniformly affected and depleted, likely by cloud processes, while the supermicron particles were influenced by clouds, and potentially other sources, in a non-uniform way. Nevertheless, the abundance of the marine tracers (sodium, MSA), together with the presence of FAA in the aerosol particles (which mainly had a similar composition compared to the oceanic and ground-based particulate FAA) indicated an oceanic contribution to the aerosol particles at cloud level."

***R#1-34)*** L24-27 Authors should comment on the fact that high LOQ of particular FAA could resulted in their variabilities or selective determination in cloud water samples.

We agree with this comment and have added on page 7, line 31-33 a statement to a possible limitation of the FAA composition in cloud water due to the LOQ.
It reads now: "Since the LOQs of the FAAs in cloud water are below 0.3 ng m$^{-3}$ and often below 0.06 ng m$^{-3}$, a limitation of the FAA composition in cloud water due to the LOQs is rather unlikely despite the variance of FAA concentrations (11.2-489.9 ng m$^{-3}$) in cloud water (section 3.3)."

***R#1-35)*** L28-32 to P14 L4-8 This paragraph deals with the possible dominant origin of FAA in the cloud water collected at the MV as within the 3.4.5. Section. I suggest skipping this paragraph 4 and combining the cloud water source discussion within the 3.4.5 section to avoid the repetition.

We agree with the reviewer and included the entire cloud water discussion in section "3.3 Cloud water samples". Section 3.4.5 has been deleted and section "3.4.4 Transfer of amino acids from the ocean into the atmosphere" now contains only some sentences on the enrichment factor in cloud water, which reads now (page 22, line 20-24): "Applying the concept of the enrichment factor to cloud water and calculating the EF$_{CW}$ (Eq. (2)), the EF$_{CW(\sum FAA)}$ = 4·10$^3$ (based on SML) and 1·10$^4$ (based on ULW) could be determined. As mentioned in section 3.3, several atmospheric processing (aging), oceanic transfer and biogenic-driven processes might contribute to this high enrichment

and need to be addressed in future studies. The high FAA concentrations and enrichments might have implications on OM processing through clouds and are worth further studying."

*R#1-36*) L33-34 Please specify the study location of Gioda et al. (2009)

Following the comment of the reviewer we specified the study location of Gioda et al., (2009). Now the text reads (page 16, line 11-13): "The concentrations of cloud water sulfate (average: 2.9 µg m$^{-3}$, Table S12) and sodium were higher than in cloud water samples, collected at East Peak in Puerto Rico, which can be seen in Gioda et al. (2009)."

**Page 15**
*R#1-37*) The authors should comment and discuss that some specific FAA were not detected in particular compartments possibly due to their high LOQ (in all matrices) and/or the fact that not enough material was collected for their analysis. This is particularly relevant for selected hydrophilic FAA in comparison to other FAA classes/groups and should be considered when discussing the selective transfer of FAA groups from the ocean to the atmosphere.

We agree with the reviewer and carefully discussed LOQ restrictions in several parts of the manuscript as follows: In the experimental part section "2.2. Analyses" an estimation was given for which amino acids due to the LOQ a restriction of the sensitivity in the marine compartment would be possible and thus no quantitative statement on the analyte would be possible.
This possible restriction of the results by the LOQs was also discussed in the results and discussion section in the individual sections on seawater (section 3.1), aerosol particles (section 3.2), cloud water (section 3.3) and concerted measurements (section 3.4). The detailed response and citations from the manuscript can be found under comment R#1-1.

**Page 17**
*R#1-38*) Although beyond the scope of this paper, I suggest discussing the potential connection of hydrophilic, hydrophobic and neutral FFA to CCN activity.

We included on page 2, line 27-30: "Due to their structure and hygroscopic properties, amino acids can act as both ice-forming particles (INP) (Wolber and Warren, 1989;Szyrmer and Zawadzki, 1997;Pandey et al., 2016;Kanji et al., 2017) as well as cloud condensation nuclei (CCN) (Kristensson et al., 2010) in the atmosphere when amino acids such as arginine and asparagine can exist as metastable droplets instead of solid particles at low relative humidity; this showed a laboratory study (Chan et al., 2005)."
However, we are not aware of a study that investigates amino acids within their hydrophilic, hydrophobic and neutral characteristics. As no CCN studies with regards to amino acids were performed here, we would like to not stress this topic too much in the manuscript.

**Page 18**
*R#1-39*) L11-13 Please correct: For the calculated EF$_{aer}$, it should be noted that no further FAA formation or degradation pathways on the aerosol particles are considered, including biological or photochemical atmospheric reactions, *and* a possible transport from other than marine sources *is excluded*.

We agree with the reviewer's note and have incorporated his correction comments for page 21. In the course of re-structuring parts of the manuscript according to the suggestions of reviewer 2, this part was changed to: "A quantitative metric for comparing compounds in the ocean and in the atmosphere is the EF$_{aer}$ (Eq. (2)). The concept is mainly applied to closed systems (e.g. Quinn et al. (2015), Rastelli et al. (2017)) because FAA formation or degradation pathways on the aerosol particles including biological or photochemical atmospheric reactions, and possible transport from other than marine sources are excluded in this parameter." (page 21, line 9-12)

***R#1-40)*** L14 remove -as-

In the course of re-structuring to address the comments of reviewer 2, this sentence was changed and reads now (page 21, line 13-15): "The averaged $EF_{aer}$ of $\sum FAA$ in the individual Berner stages of the case study at the CVAO based on SML and ULW concentrations are shown in Fig. 5."

**Page 17**
***R#1-41)*** L7-9 Please correct: Previous studies showed that organic material ejected into the atmosphere during bubble bursting, *results* in the formation of sea spray aerosol particles containing similar organic…..

We agree with the reviewer's comment and corrected the sentence on page 22, line 10-11. Now it reads: "Previous studies have shown that OM ejected into the atmosphere during bubble bursting, results in the formation of sea spray aerosol particles containing OM similar to SML (Russell et al. (2010);Cunliffe et al. (2013) and references therein)."

**Page 19**
***R#1-42)*** L18 I suggest to change the title to -Origins of FFA in cloud water-, to be consistent with previous section

Taking into account the review comment R1#-35, the section 3.4.5 was removed.

**SI material**
***R#1-43)*** Page 2, L17 ..and *is* discussed in more…

We agree with the reviewer's comment and corrected the sentence in the SI (page 3, line 5-6) to "It was generally low but increased during the campaign from 0.1 µg L$^{-1}$ to 0.6 µg L$^{-1}$ and is discussed in more detail by van Pinxteren et al., (2020)."

**Additional changes performed by the authors**

Due to the comments by reviewer 2 several parts of the manuscript were changed and adopted to the reviewer´s suggestions. This affected the following parts:
- The introduction: here the focus was more clearly placed on the FAA
- Discussion about the amino acid groups (hydrophilic, neutral and hydrophobic) in all marine compartments (seawater, aerosol particles and cloud water)
- Discussion about the concerted measurements of FAA in marine compartments
- Discussion about the aerosol enrichment factor
- Discussion and conclusion in general: stronger focus on the novelty value of the main findings shown here in the context of previous studies

When discussing the mean lifetime $\tau$ of individual amino acids (section 3.4 and Table S13), the unit of $\tau$ was changed from days (d) to hours (h).

In addition, the acknowledgement was also revised to thank the people from the OSCM. The added sentence is now as follows: "We further acknowledge the professional support provided by the Ocean Science Centre Mindelo (OSCM) and the Instituto do Mar (IMar)" (page 24, line 3-4)
The measured data were published on PANGAEA. The data availability statement was therefore updated and reads as follows: "Data availability. The data are available through the World Data Centre PANGAEA under the following link: https://doi.pangaea.de/10.1594/PANGAEA.914220."
(page 23, line 26/27)

The previous citation of van Pinxteren (submitted 2019) was updated to van Pinxteren et al. (2020) in the revised manuscript and supporting information.

**References**

van Pinxteren, M., Fomba, K. W., Triesch, N., Stolle, C., Wurl, O., Bahlmann, E., Gong, X., Voigtländer, J., Wex, H., Robinson, T. B., Barthel, S., Zeppenfeld, S., Hoffmann, E. H., Roveretto, M., Li, C., Grosselin, B., Daële, V., Senf, F., van Pinxteren, D., Manzi, M., Zabalegui, N., Frka, S., Gašparović, B., Pereira, R., Li, T., Wen, L., Li, J., Zhu, C., Chen, H., Chen, J., Fiedler, B., von Tümpling, W., Read, K. A., Punjabi, S., Lewis, A. C., Hopkins, J. R., Carpenter, L. J., Peeken, I., Rixen, T., Schulz-Bull, D., Monge, M. E., Mellouki, A., George, C., Stratmann, F., and Herrmann, H.: Marine organic matter in the remote environment of the Cape Verde islands – an introduction and overview to the MarParCloud campaign, Atmos. Chem. Phys., 20, 6921-6951, 10.5194/acp-20-6921-2020, 2020.

van Pinxteren, M. F., K. W.; Triesch, N.; Stolle, C.; Wurl, O.; Bahlmann E.; Gong, X.; Voigtländer J.; Wex, H.; Robinson, B.; Barthel, S.; Zeppenfeld, S.; Hoffmann, E. H.; Roveretto, M.; Li, C.; Grosselin, B.; Daele, V.; Senf, F.; van Pinxteren, D.; Manzi, M.; Zabalegui, N.; Frka, S.; Gašparović, B.; Pereira, R.; Li, T.; Xue, L.; Wen, L.; Wang, X.; Wang, W.; Li, J.; Chen, J.; Zhu, C.; Chen, H.; Chen, J.; Fiedler, B.; von Tümpling, W.; Read, K. A.; Punjabi, S.; Lewis, A. C.; Hopkins, J. R.; Carpenter, L. J.; Peeken, I.; Rixen, T.; Schulz-Bull, D.; Monge, M. E.; Mellouki, A.; George, C.; Stratmann, F.; Herrmann, H.: Marine organic matter in the remote environment of the Cape Verde Islands - An introduction and overview to the MarParCloud campaign, Atmos. Chem. Phys., submitted 2019.

---

## Author Comment (AC2) · 12 Sep 2020

We thank the reviewer for the careful examination of the manuscript and the supporting information. In the document "Author's response_ACP-2019-976-RC2", please find a point-by-point response to the questions and concerns. All references to the manuscript (e.g. page and line numbers) listed in our replies refer to the clean version of the manuscript (without track changes).

Please also note the supplement to this comment:

[Figure]

Creative Commons CC BY license icon

https://acp.copernicus.org/preprints/acp-2019-976/acp-2019-976-AC2-supplement.pdf

**Supplement:**

ACP-2019-976-RC2

We thank the reviewer for the careful examination of the manuscript and the supporting information. In the following, please find a point-by-point response to the questions and concerns. All references to the manuscript (e.g. page and line numbers) listed in our replies refer to the clean version of the manuscript (without track changes).

**General Comments**
R#2-1 a) This paper describes the measurement of free amino acids in seawater (microlayer and underlying water), aerosol, and cloud water at Cape Verde Island. The types and abundances of individual amino acids present in different sample types are compared. Enrichment factors are calculated for amino acid concentrations between the microlayer and underlying water, and between the water and aerosol. The contribution the amino acids make to the pools of organic carbon and total dissolved nitrogen in water and aerosol are presented.
The work largely presents an inventory of amino acids for this location and the given sample types.

We have strongly revised the manuscript an added a lot of additional interpretation which represents clearly more than an inventory of amino acids and is explained in more detail in R#2-2 point 4. We agree that the implications were not well enough elaborated. From the revised results, we derived the atmospheric implication that simple parameters alone cannot describe the abundance and enrichment of FAA in the diverse marine environments. This work shows that, for a proper representation of FAA in oceanic/atmospheric models, their drivers need to be better understood and more studies are needed to unravel their complex composition.

R#2-1 b) The concerted measurement of amino acids in underlying seawater, microlayer, aerosol, and cloud water are novel. The authors also mention that these are the first reported free amino acids concentrations for marine clouds. Enrichment factors between seawater and aerosol are useful for comparison but I urge the authors to elaborate a little on the caveats of their use in an open system, namely that it is unlikely that a high fraction of the aerosol (and cloud droplets) measured came from the location of the measured seawater (and even less likely they emerged from those waters when the water was sampled). Concentrations in the atmosphere are also subject to ageing. This differs from other work on sea spray aerosol in closed systems where all particles are emitted from the contained source waters and minimal ageing occurs on the nascent particles (at least upon collection).

We thank the reviewer for his comments. We agree with the reviewer that the caveats of the concept of enrichment factor between seawater and aerosol were not treated well enough in the manuscript. It is true that the concept of the aerosol enrichment factor originally originates from controlled tank experiments and therefore does not take into account e.g. ageing processes or long-range transport processes of the aerosol particles, which does not completely correspond to the real conditions of field investigations. However, having pointed this out, we believe that the $EF_{aer}$ is still a useful metric for comparison purposes. In recent studies, enrichment factors have been calculated from open systems and therefore provide a basis for comparative studies (e.g. Russell et al. (2010) or van Pinxteren et al. (2017)). In the revised version of our manuscript, we have underlined the uncertainties related to this concept and explained that the $EF_{aer}$ might be useful for comparison purposes, but should be treated with caution.
The same applies to the enrichment factor between cloud water and seawater, which was calculated in the same way. Again, in the revised version, we point out that this is simplified approach, which only considers a possible transfer from the ocean to the aerosol particles and cloud water (neglecting

e.g. atmospheric processing). To illustrate the $EF_{CW}$, we included the values (with respect to the SML and the ULW, respectively) in Figure 5 in the revised version.

As a result, we have very carefully addressed these points in the following parts of the revised manuscript: In section 3.4.4 it is written now (page 21, line 9-13): "A quantitative metric for comparing compounds in the ocean and in the atmosphere is the $EF_{aer}$ (Eq. (2)). The concept is mainly applied to closed systems (e.g. Quinn et al. (2015), Rastelli et al. (2017)) because FAA formation or degradation pathways on the aerosol particles including biological or photochemical atmospheric reactions, and possible transport from other than marine sources are excluded in this parameter. However, for comparison purposes, it might be useful to calculate the $EF_{aer}$ also from open systems as done e.g. by Russell et al. (2010) or van Pinxteren et al. (2017)."

In addition, we have revised the abstract and the conclusion accordingly: The abstract (page 1, line 29-31) now reads: "Considering solely ocean-atmosphere transfer and neglecting atmospheric processing, high FAA enrichment factors were found in both aerosol particles in the submicron range ($EF_{aer(\sum FAA)}$: $2 \cdot 10^3$-$6 \cdot 10^3$) and medium enrichment factors in the supermicron range ($EF_{aer(\sum FAA)}$: $1 \cdot 10^1$-$3 \cdot 10^1$)."

The conclusion (page 23, line 10-14) is adopted as follows: "In a simplified approach, considering only a possible transfer from the ocean onto the aerosol particles and cloud water (neglecting e.g. atmospheric processing), the aerosol enrichment factor was calculated. A high FAA enrichment in the submicron aerosol particles of $EF_{aer(\sum FAA)}$: $2 \cdot 10^1$-$6 \cdot 10^3$ and a medium enrichment on supermicron aerosol particles $EF_{aer(\sum FAA)}$: $1 \cdot 10^1$-$3 \cdot 10^1$ were observed. Applying the same concept to cloud water, an enrichment of $4 \cdot 10^3$-$1 \cdot 10^4$ was obtained."

Besides the concept of the enrichment factor, the revised manuscript deals with possible atmospheric processes/ aging, which are discussed in sections 3.4.1, 3.4.2 and 3.4.3 when discussing the individual amino acids or amino acid classes.

To address the representativeness of the aerosol particle measurements, we added the information about the aerosol particle measurements at the CVAO on the 30 m tower, which represent the conditions in the open ocean (and not just the surf zone), as follows (page 4, line 33 – page 5, line 2): "Size-segregated aerosol particles were sampled using five stage Berner-type impactors (Hauke, Gmunden, Austria) at the top of a 30 m sampling tower at the CVAO since this location best represents the conditions above the ocean pursuant to previous studies. The internal boundary layer (IBL), which can form when air passes a surface with changing roughness (i.e. the transfer from open water to island) is mainly beneath 30 m (Niedermeier et al., 2014)."

R#2-2) This work could be strengthened by: 1. elaborating on the significance of amino acids in the ocean and atmosphere and how the scienific community benefits from their inventory. 2. investigating the drivers of their observations by further relationships amongst their datasets and incorporating other datasets, e.g. Did variations in measured aerosol volume drive variations in amino acid concentrations? Did primary productivity (via remotely sensed chlorophyll-a) influence FAA concentrations in water and air? 3. further exploration of commonalities amond and differences between their amino acid data across different sample types. 4. what implications can be drawn the analyses accomplished and those I suggest? Are there certain ratios of amino acids in seawater that hold constant in seawater samples but not in the aerosol and clouds? What could cause this? 5. better connecting to past work. How do the observations compare not just to the values reported in other work but to the conclusions drawn in other research? How do drivers of amino acid concentrations differ between the ocean and atmosphere? What are the surface activities of the amino acids and can they predict their transfer in sea spray aersosol?

We thank the reviewer for this comment and the suggestions to improve the manuscript. We have addressed the points raised by the reviewer as shown in the following:

1. elaborating on the significance of amino
acids in the ocean and atmosphere and how the scienfic community benefits from
their inventory.

1) *Significance of amino acids*: Amino acids are important in the oceans and in the atmosphere for a number of reasons. Due to their chemical structure, amino acids contribute to both dissolved organic nitrogen (DON) and dissolved organic carbon (DOC) and are therefore important biologically available sources of nitrogen and carbon. Especially in oligotrophic regions, such as the NATR region investigated here, the amino acids are an important source of nutrients for bacteria and other microorganisms. Besides the biogenic formation of amino acids in seawater and probably also on aerosol particles (e.g. GABA, see R#2-31), amino acids (as parts of proteins) also play an important role as ice-forming particles (INP) (Wolber and Warren, 1989;Szyrmer and Zawadzki, 1997;Pandey et al., 2016;Kanji et al., 2017) or cloud condensation nuclei (CCN) (Chan et al., 2005;Kristensson et al., 2010) in the marine environment. Besides abiotic and biotic conversion processes of amino acids, secondary organic aerosol particles (SOA) can be formed by the reaction of glyoxal with amino acids (Haan et al., 2009). These are important reasons why amino acids in the marine environment have been studied both in seawater and on the aerosol particles in different regions.

With our concerted FAA measurements in all marine compartments (seawater (ULW, SML), size-segregated aerosol particles and cloud water), it was possible not only to investigate the individual compartments (section 3.1, 3.2 and 3.3), but also to focus on the individual FAA and amino acid groups to identify similarities and differences. These are discussed in detail in the comments R#2-2 3), R#2-2 4), R#2-21, R#2-27. Furthermore, for the first time the FAA concentrations and compositions in cloud water were determined (section 3.3) and compared with the other two compartments (seawater and aerosol particles) in section 3.4.

In the revised version, we stronger elaborated the significance of the amino acids, as outlined in the Introduction: "Amino acids, either free (FAA) or in combined form (CAA), contribute to the global nitrogen and carbon cycle and to the atmosphere-biosphere nutrient cycle (Zhang and Anastasio, 2003;Wedyan and Preston, 2008). They are produced in the ocean and are reported to be in the upper layer of the ocean, the sea surface microlayer (SML) (Kuznetsova et al., 2004;Reinthaler et al., 2008;van Pinxteren et al., 2012;Engel and Galgani, 2016). The SML, as the direct interface between the ocean and the atmosphere, may play an important role as a source of organic matter (OM) in aerosol particles within the marine environment (Cunliffe et al., 2013;Engel et al., 2017;Wurl et al., 2017). Specific organic groups of compounds, including nitrogenous OM (Engel and Galgani, 2016) can be strongly enriched in the SML. From the ocean, amino acids as part of the class of proteinaceous compounds can be transferred into the atmosphere via bubble bursting (Kuznetsova et al., 2005;Rastelli et al., 2017). These proteinaceous compounds are often analyzed as sum parameter 'proteins' using an analytical staining method with Coomassie blue developed by Bradford (1976) and often applied in previous studies (Gutiérrez-Castillo et al., 2005;Mandalakis et al., 2011;Rastelli et al., 2017). Despite their attribution to proteins the FAAs are better utilizable forms of nitrogen instead of proteins for an aquatic organism such as phytoplankton and bacteria (Antia et al., 1991;McGregor and Anastasio, 2001)." (page 2, line 14-26)

Furthermore, we have added the following text in the introduction about the significance of amino acids in the atmosphere (page 2, lines 27-30): "Due to their structure and hygroscopic properties, amino acids can act as both ice-forming particles (INP) (Wolber and Warren, 1989;Szyrmer and Zawadzki, 1997;Pandey et al., 2016;Kanji et al., 2017) as well as cloud condensation nuclei (CCN) (Kristensson et al., 2010) in the atmosphere when amino acids such as arginine and asparagine can exist as metastable droplets instead of solid particles at low relative humidity; this showed a laboratory study (Chan et al., 2005)."

*Benefits:* Although FAA measurements in the marine atmosphere have been available so far, there is a lack of measurements considering both the abundance and molecular composition of amino acids simultaneously in the different marine compartments, especially in the tropical Atlantic Ocean. The measurements performed here, together with the interpretation of the data, will provide better insights into the FAA abundance, origins and possible transfer from the seawater and their transport within the atmosphere in the marine environment regarding all important compartments; i.e. the ULW, the SML, the aerosol particles and finally cloud water in the remote tropical North Atlantic Ocean.

We would like to mention that the concerted FAA measurements performed here have already been included in a discussion paper published at "Biogeoscience" as "Free AA concentrations … recently quantified in cloud droplets … on the Cape Verde islands" (Jaber et al., 2020).

2. investigating the drivers of their observations by further relationships
amongst their datasets and incorporating other datasets, e.g. Did variations in
measured aerosol volume drive variations in amino acid concentrations? Did primary
productivity (via remotely sensed chlorophyll-a) influence FAA concentrations in water
and air?

2) We agree that the incorporation of other data sets was partly missing in the manuscript. Consequently, we included additional available parameters with the aim to explain the variation of FAA concentrations.
For seawater, DOC and TDN measurements (in ULW and SML), chl-*a* concentrations (as an indicator for primary productivity) and wind speed were included to investigate the observed variance of FAA concentrations in seawater. However, no statistically significant correlations could be found (in more detail in R#2-5).
For the aerosol particles, wind speed, wind direction, the particulate mass (PM) of the aerosol particles and the chl-*a* concentrations in the seawater were considered to describe the variance of the atmospheric FAA concentrations. However, again no statistically relevant correlations could be found (detailed information in R#2-6).
The surface activity of the individual amino acids was also considered in the revised version. The octanol-water partition coefficient ($K_{OW}$), the topological polar surface area (TPSA) and the density were included (Table S9). However, we found that these simple physico-chemical parameters could not explain the variance of the FAA by statistically relevant correlations. In addition to the parameters representing the surface activity, the mean lifetime $\tau$ of the aerosol particles (Table S13) was also taken into account. However, no statistically relevant correlations were found here either. For a detailed answer to the investigations on surface activity and mean lifetime we would like to refer to the comment R#2-33.

3. further exploration of commonalities amond and differences between their
amino acid data across different sample types.

3) In the revised manuscript version, a clearer focus is put on the comparison of amino acid profiles (in terms of individual amino acids and amino acid groups) in the different marine compartments. After the introduction of the individual marine compartments, seawater (section 3.1), aerosol particles (section 3.2) and cloud water (section 3.3), the composition of the amino acid profiles was examined in more detail in section 3.4. A distinction was made between hydrophilic (section 3.4.1), neutral and hydrophobic (section 3.4.2) and aromatic amino acids (section 3.4.3).
The main results are summarized in the Conclusion (page 22, line 29-34): "The similarities between the FAA composition in the seawater (SML) and on the submicron aerosol particle samples, as described in section 3.4, indicated that a certain FAA contribution, in particular the hydrophilic amino acids Asp and Glu in the submicron aerosol particles at the CVAO, was probably caused by sea spray and might be transferred up to cloud level. The neutral and hydrophobic amino acids were also present in all marine compartments, suggesting some interconnections. Stable amino acids like

Gly are often reported as long-range tracers, but their abundance in seawater and marine air masses prevailing during the sampling period suggest an (additional) oceanic source."
For a detailed overview of the individual discussion of the amino acid profiles (in the individual compartments and the cross-compartment comparison in the marine environment), we would like to refer to the comments R#2-20, R#2-21, R#2-22, R#2-24 and R#2-27.

4. what implications can be drawn the
analyses accomplished and those I suggest? Are there certain ratios of amino acids
in seawater that hold constant in seawater samples but not in the aerosol and clouds?
What could cause this?

The main points (summarized in the revised conclusion of the present study) are:
- A certain FAA contribution, particularly the hydrophilic amino acids Asp and Glu in the submicron aerosol particles at the CVAO, was probably caused by sea spray and might be transferred up to cloud level.
- Stable amino acids like Gly are often reported as long-range tracers, but their abundance in seawater and the marine air masses prevailing during the sampling period suggest an (additional) oceanic source.
- Indications for biological production of amino acids on the aerosol particles (GABA) were observed, supporting recent finding of a high active enzymatic activity on marine aerosol particles.
- The high FAA concentrations and enrichments in cloud water, which have been reported here for the first time. Their composition, together with the high concentrations of inorganic marine tracers (sodium, MSA), indicate at least to some extend to an oceanic transfer and biogenic formation that remains subject to future work.

From these findings, together with the now added additional interpretations to our data, we derived the atmospheric implications (see answer to R#2-1 a) and added the following in the Abstract and in the Conclusion: "Finally, the varying composition of the FAA in the different matrices shows that their abundance and ocean-atmosphere transfer are influenced by additional biotic and abiotic formation and degradation processes. Simple physico-chemical parameters (e.g. surface activity) are not sufficient to describe the concentration and enrichments of the FAA in the marine environment. For a precise representation in organic matter (OM) transfer models, further studies are needed to unravel their drivers and understand their composition." (page 2, line 3-8) and "Altogether, the varying composition of FAAs in the different matrices shows that their abundance and their enrichments in the SML and their atmospheric transfer are not determined by single environmental drivers (e.g. wind speed) and/or simple physico-chemical parameters (e.g. surface activity). The ocean-atmosphere transfer of FAAs is influenced by biotic and abiotic formation and degradation processes. Further studies are required to unravel their drivers and understand their complex composition that, finally, have to be considered in OM transfer models." (page 23, line 17-23)
For more details, we would like to refer to our answer to the referee comment R#2-3.

These results and implications were derived from studying the ratios via the percentage composition of the FAA in the individual compartments together with other information (e.g. environmental parameters and phyico-chemical properties). For a more detailed discussion of the amino acid ratios/ profiles, we would like to refer to our answers to the Review comments R#2-21 R#2-26 and R#2-27, R#2-31.

5. better connecting to past work. How do the observations
compare not just to the values reported in other work but to the conclusions drawn in
other research? How do drivers of amino acid concentrations differ between the ocean
and atmosphere? What are the surface activities of the amino acids and can they
predict their transfer in sea spray aersosol?

We agree that the connection of our work to previous studies was not strong enough. In the revised manuscript version, we have significantly extended the comparison of our observations to the conclusions drawn in earlier studies. We have also outlined the uncertainties of sources and drivers of amino acids in the marine environment and connected our findings to literature studies at several parts of the manuscript as follows:

In the Introduction, we have included the following (page 3, line 3-14): "Based on a cluster and factor analysis, Scalabrin et al. (2012) suggested two possible sources for the amino acids in the ultrafine Arctic aerosol particles. First, the authors mentioned the regional development (isoleucine, leucine, threonine) and long-range transport (glycine) of amino acids from marine areas; secondly, the influence of local sources such as of marine primary production (proline, valine, serine, tyrosine, glutamic acid). A different approach of Mashayekhy Rad et al. (2019) investigated the atmospheric proteinogenic aerosol particles in the Arctic and attributed them to different sources based among others on the reactivity of the distinct amino acids. The authors differentiated here between long-range transport (glycine), terrestrial and marine aerosol particles (proline, valine, serine, tyrosine) and coastal and marine phytoplankton and bacteria (isoleucine, leucine and threonine) as important sources for amino acids (Mashayekhy Rad et al., 2019). In fact, previous studies have assigned individual amino acids to specific marine biogenic sources and used them as biomarkers. Hammer and Kattner (1986) reported correlations between aspartic acid, diatoms and zooplankton in seawater. GABA ($\gamma$-aminobutyric acid) was referred to as an indicator for the microbiological decomposition of OM (Dauwe et al., 1999;Engel et al., 2018) and is probably used as a microbiological proxy in aerosol particles."

In section 3.1, we carefully revised the section on FAA enrichment in the SML and it reads now as follows (page 10, line 16-30): "The results of the individual FAA concentrations in seawater (ULW, SML) and their $EF_{SML}$, listed in Table S3, show clear differences between the individual amino acids and the amino acid classes. The most highly enriched amino acids in the SML are the neutral ones with values of up to 203 compared to the hydrophilic ($EF_{SML}$: 2-98) and hydrophobic ($EF_{SML}$: 1-96) amino acids. This may be related to the fact that Ser, Thr and Gly as part of the neutral amino acids, are main components of cell wall proteins (Hecky et al., 1973). The direct release of FAAs through cell lysis and the associated destruction of the cell wall can thus explain the increased enrichment of neutral amino acids in the SML. Our study confirmed that the SML is often non-uniformly enriched with FAAs as outlined from previous observations (Kuznetsova and Lee, 2002;Reinthaler et al., 2008;van Pinxteren et al., 2012;Engel and Galgani, 2016). Different factors, such as the transport of FAA from the ULW to the SML, the in-situ production by an extracellular hydrolysis of CAA or a direct release of FAA by cell lysis probably cause the observed enrichment of FAA in the SML. Kuznetsova and Lee (2002) showed that the rapid extracellular hydrolysis of CAA in the SML was not the cause of the non-uniformly enrichment in SML. Moreover, they suggested that the intracellular pools of organisms rich in DFAA and DCAA compared to seawater can be leached out by stressed microorganisms, resulting in the release of DFAA which in turn influences the pools of both DFAA and DCAA in seawater. Based on previous studies, the transportation and releasing mechanisms seem most likely to be the reasons for the observed enrichment of FAA."

And in section 3.2.1, we revised the discussion as follows (page 14, line 21-27): "Following this hydropathy classification, the submicron aerosol particles consisted on average of 5 % hydrophobic, 15 % hydrophilic and 80 % neutral amino acids, while the supermicron aerosol particles contained on average only 7 % hydrophobic and 93 % neutral amino acids (Table S7). During the campaign, an increase in the contribution of hydrophilic amino acids was observed with a maximum of 55 % on 7/10/2017. Barbaro et al. (2015) reported that hydrophilic components were predominant (60 %) in locally produced marine Antarctic aerosol particles, whereas hydrophobic compounds were rather dominate aerosol particles collected at the continental station (23 % and 27 %). According to

the conclusions by Barbaro et al. (2015), the relatively high content of hydrophilic FAA found here points at least at some influence of local oceanic sources."

As mentioned above (R#2-2, point 2), we included a variety of environmental factors and also considered substance-specific properties as well as the mean lifetime $\tau$ to explain the amino acid concentrations in seawater or on the aerosol particles and to predict the transfer. However, no statistically relevant correlations could be found for all additionally investigated parameters. Therefore, the drivers of amino acid concentrations between ocean and atmosphere could not be distinguished.

R#2-3) Ultimately I would like the authors to demonstrate that they have done more than make some measurements and that we have gained new insight into the ocean-atmosphere system beyond an inventory of amino acids. The manuscript needs to set up what was done on top of our current understanding of sea spray aerosol formation and the transfer of different compounds/-classes into sea spray aersosol. By building on past research the present work is prepared to more clearly communicate its findings and to also contribute new knowledge to the field.

Based on the new measurement data obtained in our work, we have now added many more additional interpretations to our data. Our novel approach of the concerted measurements (ULW, SML, aerosol particles and cloud water) – which is clearly more than 'some measurements - together with the analysis at molecular level provided a rich data set of the ambient amino acids. These measurements allowed to study similarities and differences of FAA in the various compartments and indicated that a certain FAA contribution, in particular the hydrophilic amino acids Asp and Glu in the submicron aerosol particles at the CVAO, was probably caused by sea spray and might be transferred up to cloud level. We showed that the neutral and hydrophobic amino acids were also present in all marine compartments, suggesting some interconnections. Stable amino acids like Gly are often reported as long-range tracers, but their abundance in seawater and the marine air masses prevailing during the sampling period suggest an (additional) oceanic source. The oceanic link is supported by a high atmospheric concentration of ocean-derived compounds (sodium, MSA), a high fractional residence time of the air masses above water and a low-to-medium impact of other non-marine sources (based e.g. on the mass concentration of trace metals). In addition, we could derive some indications for the biological production of amino acids on the aerosol particles (GABA), supporting the recent finding of a high active enzymatic activity on marine aerosol particles. We found that aromatic amino acids are either not transferred from the ocean into the atmosphere or react very quickly, in any case they are present in low concentrations close to the LOQ. By distinguishing between submicron and supermicron aerosol particles, differences in the chemical composition of these aerosol particle size classes could be identified, which show a much higher complexity of the FAA composition in the submicron aerosol particles. FAA were present in the size range for aerosol particles associated with CCN activity and cloud water, and might be connected to CCN activity due to their hygroscopicity and soluble character, but this effect was not investigated here.
Regarding the enrichment factor (see our answer to the comment R#2-1 b).
These aspects are summarized in the conclusions of the revised version: "In a simplified approach, considering only a possible transfer from the ocean onto the aerosol particles and cloud water (neglecting e.g. atmospheric processing), the aerosol enrichment factor was calculated. A high FAA enrichment in the submicron aerosol particles of $EF_{aer(\sum FAA)}$: $2 \cdot 10^1$-$6 \cdot 10^3$ and a medium enrichment on supermicron aerosol particles $EF_{aer(\sum FAA)}$: $1 \cdot 10^1$-$3 \cdot 10^1$ were observed. Applying the same concept to cloud water, an enrichment of $4 \cdot 10^3$-$1 \cdot 10^4$ was obtained. The high FAA concentrations (11.2-489.9 ng m$^{-3}$) and enrichments in cloud water were reported here for the first time. Their composition, together with the high concentrations of inorganic marine tracers (sodium, MSA),

indicate at least to some extend an oceanic transfer and biogenic formation that remains subject to future work. Altogether, the varying composition of FAAs in the different matrices shows that their abundance and their enrichments in the SML and their atmospheric transfer are not determined by single environmental drivers (e.g. wind speed) and/or simple physico-chemical parameters (e.g. surface activity). The ocean-atmosphere transfer of FAAs is influenced by biotic and abiotic formation and degradation processes. Further studies are required to unravel their drivers and understand their complex composition that, finally, have to be considered in OM transfer models. To the best of our knowledge, this study was the first that simultaneously analyzed the FAA in all marine compartments - seawater including the ULW and the SML, size-segregated aerosol particles and cloud water – in such detail to obtain indications on their sources and interconnections." (page 23, line 10-24).

Altogether, we believe that the results mentioned above provide new insight into the ocean-atmosphere system beyond an inventory of amino acids.

R#2-4) Introduction. page 3. line 22. Barbaro et al. (2015) investigated FAA in size-segregated Antarctic aerosol particles to gain information about FAA as possible tracers of primary biological production in Antarctic aerosol particles line 24. Although there are several studies in different marine regions, there is a lack of ambient measurements of FAA simultaneously in seawater and in size-segregated aerosol particles in the tropical Atlantic Ocean It would strengthen the justification for the work if it was stated why we must know this information specifically for the Atlantic Ocean and atmosphere.

As stated correctly by the reviewer, there are several studies of amino acids in marine samples. However, source attributions of amino acids are still not clear and might vary between different marine locations. In addition, there is mostly a lack of measurements that regard the abundance and molecular composition of amino acids simultaneously in marine compartments - in seawater and in the atmosphere – especially in the tropical Atlantic Ocean. Such studies are, however, crucial to learn more about sources and fate of the amino acids in the ambient marine environment. Besides, cloud water studies of amino acids are lacking. However, such studies are needed to investigate if these important compounds (see answer to the comment R#2-2) are transferred or maybe produced in marine clouds.

We referred to these points more clearly in the introduction and reads as follows (page 3, line 22-30): "Despite several studies of FAAs also conducted in the marine environment, there is still a huge uncertainty to the question whether FAAs are of marine origin or not. Matsumoto and Uematsu (2005) showed that the long-range transport of land-derived sources largely contributes to the amino acid concentration in the North Pacific. On the other hand, based on a positive correlation between amino acids in seawater and the atmosphere, Wedyan and Preston (2008) pointed out the particulate amino acids in the Southern Ocean to be of marine origin. These findings are likely due to regional varying source strengths, given different meteorological and biological conditions, which require further measurements in distinct marine regions necessary. Unfortunately, measurements are lacking that regard the abundance and molecular composition of amino acids in both seawater and size-segregated aerosol particles, especially in the tropical Atlantic Ocean."

The region investigated here, the NATR region around the Cape Verde Islands (see R#2-5), is an interesting but rarely studied oligotrophic region. The region of study is of huge interest as it is home to a remote marine time-series observatory with low anthropogenic influences where particles during the campaign were predominantly of marine origin. Therefore, our studies provide a better understanding of the FAA in such a region on a molecular level, in order to be able to describe their sources and fate in the marine environment more precisely.

The main statements of this study concerning amino acids in the atmosphere in this marine region in view of the present state of knowledge and the new main findings of this study are summarized in more detail in answer R#2-8. For the general importance of the amino acids, please see our answer to the referee comment R#2-2.

R#2-5) There is little effort made to investigate the drivers of amino acids data for the ULW and SML. The data are not plotted against other measurements. We are merely offered (page 9, line 19): "On the basis of previous studies, the transportation and the releasing mechanisms seems to be most likely for the observed enrichment of FAA." At this point, we were not given a strong motivation for the measurements and we haven't learned anything from them. Then it is explained that the data are in agreement with other datasets and then this one dataset is extrapolated to the entire North Atlantic Ocean. I wouldn't extrapolate data from one location to the entire North Atlantic Ocean.

We agree that not sufficient effort was made to investigate the drivers of amino acids data for the ULW and SML and have consequently included correlations to environmental parameters in the revised version. However, correlations of the FAA data with either wind speed, chl-*a*, DOC or TDN could not explain the variance in FAA concentrations or enrichments. That was, however, in agreement to results of previous studies.
The transportation and releasing mechanisms is discussed in section 3.1. We found high enrichments of FAA in the SML that we considered interesting. The environmental parameters could not explain the observed FAA enrichment in the SML of neither the individual nor the amino acid groups. However, we could observe that the neutral amino acids in particular are more abundant than the hydrophilic or hydrophobic ones. This may be related to the fact that neutral amino acids are the main components of cell wall proteins and are directly released by cell lysis, as we concluded by comparing our findings to results from the literature.
The motivation for the measurements are summarized in the revised introduction, please see our answer to the referee comments R#2-2 and R#2-8.

Several changes according to these aspects were done in the revised manuscript. For a better visualization, we have included the temporal variation of the respective additional parameters (wind speed, chl-*a*, DOC and TDN) in Figure 1. The discussion of the correlations to environmental parameters was added and reads (page 10, line 12-14): "Nevertheless, the variance of the ∑FAA concentrations in the SML or ULW observed here could neither be explained by the variance of DOC or TDN values, nor by wind speed and chl-a concentrations (see Fig. 1, Table S2, S5), since no correlation between these parameters and the concentration or enrichment of FAA was found."
For explanations of the observed enrichment of the FAA in the SML we want to refer to the referee comment R#2-2 point 5).

R#2-5_1) Then it is explained that the data are in agreement with other datasets and then this one dataset is extrapolated to the entire North Atlantic Ocean. I wouldn't extrapolate data from one location to the entire North Atlantic Ocean.

We agree that we might not extrapolate the local data to the entire North Atlantic Ocean, as it appeared in the first version of the manuscript. In the revised version we have subdivided the marine regions in more detail according to Longhurst (2007). Thus, in the revised manuscript version we refer to the "North Atlantic Tropical Gyral Province (NATR)" region (introduced by Longhurst (2007)). This region is defined as follows in the manuscript (page 4, line 13-16): "In accordance with the classification of Longhurst (2007), the ocean around the Cape Verde Islands belongs to the region "North Atlantic Tropical Gyral Province (NATR)", which is described as the region with the lowest surface chlorophyll in the North Atlantic Ocean having a greater annual variability than seasonality."
We compared our measurements with previous own FAA measurements at the same location (data from 2013, shown here for the first time in Table S4) and the good agreement allowed us to conclude that FAA concentrations shown in this study can be considered as representative for the NATR region surrounding the Cape Verde Islands.
Moreover, we compared the FAA concentrations obtained here to published FAA data from the subtropical Atlantic Ocean the western Mediterranean Sea and concluded that "the FAA concentrations in the NATR region, with its very low surface chlorophyll and a greater annual

variability than seasonality, are in the same order of magnitude compared to other marine regions (i.e. subtropical Atlantic and western Mediterranean Sea (Reinthaler et al., 2008)).” (page 11, line 5-7).

R#2-6) Similar to the seawater measurements, the atmospheric research would be stronger if it explored the drivers of the amino acid data. Particularly lacking from the aerosol work are aerosol sizing measurements. In their absence, we do not know if the variability in amino acid concentrations in the air are driven by changes in particle concentrations. Similarly, amino acid concentrations in the air could be related to wind direction.

Following the comment of the reviewer, and similar to the seawater results, we have tested to correlate the particulate FAA concentrations with wind speed, wind direction, chl-*a* data in seawater as well as the particulate mass concentrations of the aerosol particle samples to explain the variance of amino acid concentrations in the aerosol particle samples. However, statistically relevant correlations could not be found. In the manuscript the text now reads (page 12, line 10-12): “Correlations between the ∑FAA concentrations of the size-segregated aerosol particles (considered as submicron, supermicron and $PM_{10}$) and the wind speed were not observed for here reported data (Fig. 2, Table S2).”
and on page 12, line 18/19: “No significant correlation could be observed between the ∑FAA concentration of size-segregated aerosol particle samples (submicron, supermicron and $PM_{10}$) and the chl-*a* concentration in seawater.”
and on page 14, line 13-15: “However, the reasons for the high concentration of hydrophilic FAAs within these respective sampling days remain unclear, since no change in the environmental parameters determined (e.g. wind speed, wind direction, chl-*a* concentration, Fig. 2a) was observed.”
For a better visualization, we have included the temporal variation of the respective additional parameters (wind speed, wind direction and chl-*a*) in Figure 2.

Other:
R#2-7) Did particle type vary across the different aerosol sampling periods?

We have included the discussion of several meteorological parameters and back trajectories as well as particulate mass concentrations and included the new small chapter “*First indications of aerosol particle origin*“. From these discussed measurements we could not observe changes in the type of particles across the different aerosol sampling periods. In the revised manuscript it reads now as follows: “To obtain a first indication of the particle origin, that might help to explain the differences in the particle composition concerning amino acids, the particles were associated with the origin of the air masses and with marine and dust tracers. Overall, the CVAO station experienced north-easterly trade winds during this campaign, which are typical for this season within this region (Fomba et al., 2014;van Pinxteren et al., 2020). According to physical and chemical specifications such as the air mass origins, particulate MSA concentrations and MSA/sulfate ratios as well as particulate mass concentrations of dust tracers, aerosol particles predominantly of marine origin with low to medium dust influences were observed. It has to be noted that dust generally influences the supermicron particles to a larger extent than the submicron particles (Fomba et al., 2013). Further information on the classification of the air masses are given in the overview paper of this campaign (van Pinxteren et al., 2020).“ (page 11, line 25-33)

R#2-8) If amino acids have already been measured in remote marine air/aerosol, how does the present manuscript advance our knowledge?

Although some studies on amino acids in marine aerosol particles have been published so far, the source of the amino acid as well as their transfer and fate is still not clear. Especially with regard to the connection between ocean and atmosphere, the gap in the simultaneous investigation of both marine compartments is problematic with regard to FAA. We have briefly summarized the current

state of knowledge on this subject in the revised introduction. This now reads as follows (page 3, line 21-30): "Although the study and characterization of amino acids are of paramount importance for atmospheric scientists, the true role and the fate of amino acids in the atmosphere are still poorly understood (Matos et al., 2016). Despite several studies of FAAs also conducted in the marine environment, there is still a huge uncertainty to the question whether FAAs are of marine origin or not. Matsumoto and Uematsu (2005) showed that the long-range transport of land-derived sources largely contributes to the amino acid concentration in the North Pacific. On the other hand, based on a positive correlation between amino acids in seawater and the atmosphere, Wedyan and Preston (2008) pointed out the particulate amino acids in the Southern Ocean to be of marine origin. These findings are likely due to regional varying source strengths, given different meteorological and biological conditions, which require further measurements in distinct marine regions necessary. Unfortunately, measurements are lacking that regard the abundance and molecular composition of amino acids in both seawater and size-segregated aerosol particles, especially in the tropical Atlantic Ocean."

We wanted to investigate the already mentioned knowledge gap about the ocean-atmosphere relationship more closely through our methodological approach and the associated FAA analysis, and to introduce new aspects or support previous assumptions of other studies in this marine region as well. To this end, we have summarized the main objectives of our work in the revised introduction as follows (page 3, line 31 – page 4, line 5): "So, the aim of the present study is to investigate the occurrence of FAA in the marine environment regarding all important compartments; i.e. the ULW, the SML, the aerosol particles and finally cloud water in the remote tropical North Atlantic Ocean at the Cape Verde Atmospheric Observatory (CVAO). Their abundance, origin and possible transfer from the seawater as well as their transport within the atmosphere are studied in particular. Therefore, the FAA are measured on a molecular level and divided into hydrophilic (glutamic acid, aspartic, GABA), neutral (serine, glycine, threonine, proline, tyrosine) and hydrophobic compounds (alanine, valine, phenylalanine, isoleucine, leucine) according to their hydropathy index. Especially the similarities and differences between the amino acid composition in submicron (0.05-1.2 µm) and supermicron (1.2-10 µm) aerosol particles are elucidated. Finally, the potential of individual FAA as proxies or tracers for specific sources of aerosol particles and cloud water in the tropical marine environment is outlined."

The main findings that are beyond the state of knowledge are summarized in the conclusion and have been outlined in our answer to the referee comment R#2-3.

**Specific Comments**
R#2-9) Abstract. page 1. line 20. "The total concentration (PM10) was between 1.8–
6.8âˇL'ngâˇL'm⊡3 and tended to increase during the campaign." Perhaps instead of
"during the campaign", give the time period or relate to a potential/suspected driver of
the observation.

Following the comment of the reviewer the added the time period of the campaign in the Abstract and it reads now (page 1, line 12-14): "Measurements of free amino acids (FAA) in the marine environment to elucidate their transfer from the ocean into the atmosphere to marine aerosol particles and to clouds were performed at the MarParCloud campaign at the Cape Verde islands in autumn 2017."

Introduction.
page 2.
R#2-10) lines _8-12. The marine nitrogen cycle is alluded to but vaguely. "(T)he bulk DON
pool CAN BECOME bioavailable" or "IS bioavailable"? What is remineralized nitrogen?
Perhaps get straight to proteniaceous compounds and amino acids as being large
members of the DON pool.

Due to the revision of the introduction and also following the suggestions from the reviewer's, we focused on the FAA right from the beginning and therefore general parts (as the one mentioned by the reviewer) were deleted from the manuscript.

R#2-11)line 21. much more work has been done on amino acids as ice nucleating entities (and as antifreeze entities) than just Szyrmer and Zawadzki, 1997 and should be ackowledged because it gives importance to this work.

We agree with the comment of the reviewer. We have added more references in the reference to the INP ability of amino acids. It reads now (page2, line 27-30): "Due to their structure and hygroscopic properties, amino acids can act as both ice-forming particles (INP) (Wolber and Warren, 1989;Szyrmer and Zawadzki, 1997;Pandey et al., 2016;Kanji et al., 2017) as well as cloud condensation nuclei (CCN) (Kristensson et al., 2010) in the atmosphere when amino acids such as arginine and asparagine can exist as metastable droplets instead of solid particles at low relative humidity; this showed a laboratory study (Chan et al., 2005)."

R#2-12)line 23. "Amino acids are also present and have been described in the marine environment." This has already been established. The first 3 sentences of this paragraph feel like we are going backwards. In the previous paragraph we went from the ocean to the atmosphere and now we are back to describing amino acids in the ocean.

We agree with the comment of the reviewer and have removed the noted sentences in the course of restructuring the introduction. With the revised introduction, the focus should now be more clearly on amino acids and the structure of the introduction should also be more coherent with the revision (as outlined in the answer to the referee comment R#2-2).

R#2-13)line 28. (Engel et al., 2017) lacking other citations for the microlayer's importance in the ocean and sea spray aerosol.

Following the comment of the reviewer, we added other citations in this sentence. Now it reads (page 2, line 17-19): "The SML, as the direct interface between the ocean and the atmosphere, may play an important role as a source of organic matter (OM) in aerosol particles within the marine environment (Cunliffe et al., 2013;Engel et al., 2017;Wurl et al., 2017)."

page 3.
R#2-14) line 7. "However, chl-a concentration solely does not adequately describe the complete spectrum of biological activity (Quinn et al., 2014)" should reference a paper on marine microbiology, like Azam and Malfatti. 2007. Microbial Sctructuring of Marine Ecosystems. DOI: 10.1038/nrmicro1747

We agree with the comment of the reviewer and have removed the noted sentences in the course of restructuring the introduction. With the revised introduction, the focus should now be more clearly on amino acids and the structure of the introduction should also be more coherent with the revision (as stated in the answer to the referee comment R#2-2).

page 4.
R#2-15) line 13. "During this campaign, concerted measurements were performed including sampling of sizesegregated aerosol particles at the CVAO and seawater sampling at the ocean site (_16_53˝Ĺ30ˊĹN, _24_54˝Ĺ00˝ĹË˝ĹW). The location was carefully chosen with minimal influence of the island and located in wind direction to the CVAO" This second sentence is crucial to the rationale of the study and should be demonstrated. I'm surprised to not see a figure referenced here.

We agree with the comment of the reviewer and added a Figure S1 in the SI. Figure S1 shows an overview of the sampling locations MV, CVAO and seawater sampling site now and additionally the prevailing wind direction was added. In the manuscript, now it reads (page 4, line 13-19): "In accordance with the classification of Longhurst (2007), the ocean around the Cape Verde Islands belongs to the region "North Atlantic Tropical Gyral Province (NATR)", which is described as the region with the lowest surface chlorophyll in the North Atlantic Ocean having a greater annual variability than seasonality. During this campaign, concerted measurements were performed including the sampling of size-segregated aerosol particles at the CVAO and seawater sampling at the ocean site (~16°53'17'N, ~24°54'25'E). The location was carefully chosen with minimal influence of the island and located in wind direction to the CVAO as shown in Fig. S1."

In addition, we would like to point out that a general introduction to the campaign and the setting is given by the (now published) article by van Pinxteren et al., 2020 that we refereed to (for example: "van Pinxteren et al. (2020) provide further details on the MarParCloud campaign.", page 4, line 20/21).

R#2-16) line 20. "The seawater samples were taken with a fishing boat, starting from Bahia das Gatas, São Vicente." A study site map with water and aerosol locations and winds during sampling periods would be good to "connect the dots". Figure S1 does not do this.

We agree with this comment and the previous comment (R#2-15) of the reviewer and added Fig. S1 as on overview of the sampling locations MV, CVAO and seawater sampling site in the SI.

page 6.
R#2-17) line 2. "Since no chiral column was used in the UHPLC separation, it is possible that not only L-amino acids, which were used as the standard, were quantified, but that the here presented concentrations were possibly quantified as the sum of the L- and D-amino acids." So why not simply report them as L- and D-amino acids instead of reporting them as L-amino acids but acknowledging that is not accurate?

We thank the reviewer for this comment and have changed the sentence so that it now reads as follows: "Since no chiral column was used in the UHPLC separation, we cannot differentiate between L- and D- amino acids in our ambient samples" (page 6, line 15/16). Furthermore, the following manuscript does not differentiate between D- and L-amino acids.

R#2-18) 2.2.4 Enrichment Factors
For calculating enrichment factors (EFs) between aerosol and the SML, I wonder if it would be more appropriate to first calculate the aerosol-ULW EF using equation 2 and then apply the SML-ULW EF (EF_SML) from equation 1 for that analyte. So there would be EFs for aerosol relative to the underlying water and relative to the SML and they would differ by the EF_SML (Eq.1). Ideally this would resolve the current inconsistency in invoking the Na+ concentration of the SML when comparing to aerosol to SML (Eq. 2) but not when comparing SML to ULW. The way it stands now, if you calculate aerosol EFs relative to the ULWand the SML they won't differ by the EF_SML and instead will differ by an EF_SML calculated using equation 2.

As mentioned above (answer to the reviewer comment R#2-1b), we strongly revised the here applied concept of the calculation of the EF$_{aer}$ and pointed out the relating caveats. We used the EF$_{aer}$ for comparison purposes and in this regard, we included the EF$_{aer}$ based on ULW (besides the EF$_{aer}$ based on SML). The Na$^+$ concentration in the ULW and SML was very similar (12.45 g L$^{-1}$ in the ULW and 12.53 g L$^{-1}$ in the SML), therefore differences in the EF$_{aer}$ (related to the SML and to the ULW, respectively) are mainly due to the different concentrations, i.e. the high SML enrichments of the FAAs.

To calculate the EF$_{aer}$ (Equation 2), we followed previous studies. In these studies, the enrichment of a substance was calculated by using on the aerosol particles the concentration of the substance in relation to the sodium concentration in relation to the substance concentration and sodium concentrations present in seawater. This was done, for example, under controlled conditions as described in Rastelli et al. (2017), but also in ambient studies such as Russell et al. (2010) and van Pinxteren et al. (2017).

page 10.
R#2-19) line 1. This sentence is confusing: "In the study of Reinthaler et al. (2008) DFAA contributed with _12 % to DOC and with _ 30 % to dissolved organic nitrogen (DON) in the SML of the Atlantic ocean and the western Mediterranean Sea." Is it DFAA in seawater accounting for _12% of DOC (in seawater), and DFAA in SML accounting for _30% of DON in the SML?

We agree with the reviewer and have reworded the sentence (page 11, line 17-19) as follows: "In the SML of the Atlantic Ocean and the western Mediterranean Sea, the DFAA contributed with ~ 12 % of the DOC and ~ 30 % of the dissolved organic nitrogen (DON) (Reinthaler et al., 2008)."

R#2-20) line 28. The characterization of FAA into the hydrophilic, hydrophobic, and neutral classes is nice. What might be the drivers and implications?

The FAA's division into hydrophilic, hydrophobic and neutral amino acid classes was used to ensure better comparability with previous studies. Often different analytical methods are used to study the FAA in different marine compartments, mostly seawater or aerosol particles, and often different individual FAA standards are used. In order to be able to make statements about the FAA with similar physico-chemical properties, the classification of the amino acids is based on the 'hydropathy index'. This classification was used, for example, in previous studies to classify the FAA results in the aerosol particles at the CVAO. This can be read as follows in the revised manuscript (page 14, line 21-27): "Following this hydropathy classification, the submicron aerosol particles consisted on average of 5 % hydrophobic, 15 % hydrophilic and 80 % neutral amino acids, while the supermicron aerosol particles contained on average only 7 % hydrophobic and 93 % neutral amino acids (Table S7). During the campaign, an increase in the contribution of hydrophilic amino acids was observed with a maximum of 55 % on 7/10/2017. Barbaro et al. (2015) reported that hydrophilic components were predominant (60 %) in locally produced marine Antarctic aerosol particles, whereas hydrophobic compounds were rather dominate aerosol particles collected at the continental station (23 % and 27 %). According to the conclusions by Barbaro et al. (2015), the relatively high content of hydrophilic FAA found here points at least at some influence of local oceanic sources."
Moreover, the classification within this study was used to investigate the FAA composition in the different marine compartments (section 3.1, 3.2 and 3.3) and also to compare the compartments with each other. This is done after subdividing the amino acid classes in sections "3.4.1 Hydrophilic amino acids" and "3.4.2 Neutral and hydrophobic amino acids".

page 11
R#2-21) line 11. "However, the presence of Glu, Asp and GABA as part of the hydrophilic species in the submicron aerosol particles (on 22/09/2017, 4/10/2017, 6/10/2017, 7/10/2017) strongly indicated a local oceanic origin." If amino acids can indicate aerosol type or source, this should be central to this work, explained in the introduction, and examined for each sample. Also, how did the amino acid profiles of the different sample types (and individual samples) compare? Were the same relative abundances consisten across the ULW, SML, and different aerosol size classes? Are all the amino acids measured commonly found in the ocean and are they exclusive to the ocean?

There is still a high uncertainly about source attributions of FAA in the marine atmosphere. In the revised manuscript, we addressed this issue in several parts in the revised manuscript as outlined in the following. In the introduction we listed the results concerning the attribution of FAAs to distinct sources from other studies (detailed changes are shown in our response to comment R#2-8) and concluded: "Despite several studies of FAAs also conducted in the marine environment, there is still a huge uncertainty to the question whether FAAs are of marine origin or not. Matsumoto and Uematsu (2005) showed that the long-range transport of land-derived sources largely contributes to the amino acid concentration in the North Pacific. On the other hand, based on a positive correlation between amino acids in seawater and the atmosphere, Wedyan and Preston (2008) pointed out the particulate amino acids in the Southern Ocean to be of marine origin. These findings are likely due to regional varying source strengths, given different meteorological and biological conditions, which require further measurements in distinct marine regions necessary. Unfortunately, measurements are lacking that regard the abundance and molecular composition of amino acids in both seawater and size-segregated aerosol particles, especially in the tropical Atlantic Ocean." (page 3, line 22-30)

In our work we performed a proper comparison of the FAA in the different matrices and concluded: "The high complexity of FAA observed in seawater was also found in the aerosol particles as well as in cloud water, and generally shows a high similarity between FAA in the different compartments. All marine compartments contained Gly, Ser, Glu and Ala as dominant species, i.e. representatives of the hydrophilic, neutral and hydrophobic groups. However, the percentage contribution of the individual FAAs to the ∑FAA varies within the different compartments." (page 18, line 3-7)

The individual FAA are discussed in section 3.4.1, 3.4.2 and 3.4.3.

Important findings from our work show:

"Consequently, the usage of the major FAAs as chemical biomarkers seems to be restricted to some extend due to their lack of source-specifity" (page 14, line 9/10)

Still, we could conclude from our investigations that "…a certain FAA contribution, in particular the hydrophilic amino acids Asp and Glu in the submicron aerosol particles at the CVAO, was probably caused by sea spray and might be transferred up to cloud level. The neutral and hydrophobic amino acids were also present in all marine compartments, suggesting some interconnections. Stable amino acids like Gly are often reported as long-range tracers, but their abundance in seawater and marine air masses prevailing during the sampling period suggest an (additional) oceanic source." (page 22, line 30-34)

This is in agreement with results from Barbaro et al. (2015). "Barbaro et al. (2015) reported that hydrophilic components were predominant (60 %) in locally produced marine Antarctic aerosol particles, whereas hydrophobic compounds were rather dominate aerosol particles collected at the continental station (23 % and 27 %). According to the conclusions by Barbaro et al. (2015), the relatively high content of hydrophilic FAA found here points at least at some influence of local oceanic sources." (page 14, line 24-27)

Furthermore, we found "In addition, some indications for the biological production of amino acids on the aerosol particles (GABA) were observed, supporting the recent finding of a high active enzymatic activity on marine aerosol particles. Aromatic amino acids are either not transferred from the ocean into the atmosphere or react very quickly; in any case, they are present only in small concentrations close to the LOQ." (page 23, line 3-6)

We think that the results summarized here, together with the reported high FAA concentrations in cloud water (see Comment R#2-27), are new and interesting and help to gain better insights into sources and transfer of FAA to the marine environment.

page 12
R#2-22) line 2. "Considering the amino acid classifications from Barbaro et al. (2015)), it can be concluded that the submicron aerosol particles with low averaged percentage of hydrophobic species (5 %) and higher percentages of hydrophilic species (4-55 %, mean

of 15 %) could have local oceanic origin." Similar to previous comment. The different amino acid characterizations: hydropathy index of Kyte & Doolittle 1982; Pommie et al. 2004; and Barbaro et al. 2005 should be explained early on and would add value to the present work.

Following the reviewers' suggestion, we have described the concept of the hydropathy index already in the introduction. It reads now (page 3, line 14-18): "Grouping amino acids as regards their physico-chemical properties ('hydropathy' index (Kyte and Doolittle, 1982)) allows different studies to better compare them what Pommié et al. (2004) suggested pursuant to the partition coefficient between water and ethanol. This divides them into hydrophilic, neutral and hydrophobic amino acids as discussed in Barbaro et al. (2015) for FAA in Antarctic aerosol particles."
And on page 3, line 34 – page 4, line 2: "Therefore, the FAA are measured on a molecular level and divided into hydrophilic (glutamic acid, aspartic, GABA), neutral (serine, glycine, threonine, proline, tyrosine) and hydrophobic compounds (alanine, valine, phenylalanine, isoleucine, leucine) according to their hydropathy index."
Furthermore, this classification of amino acids, introduced in the introduction, is taken up in the discussions of the individual marine compartments (section 3.1, 3.2 and 3.3) and in the comparison of the marine compartments with regard to the FAA classifications (section 3.4.1, 3.4.2, 3.4.3).

R#2-23) line 5. "This is supported by a predominant marine origin of the aerosol particles according to the air masses history, particulate MSA concentrations and MSA/sulfate ratios and particulate concentrations of dust tracers (Table S8)." It would be nice to lay this out because we make it this far into the manuscript wondering about the presence of aerosols from land in these samples. I would characterize the air masses early on. Do the back trajectories differ for the samples collected at 2 different elevations? And which back trajectories were used? Hysplit or Flexpart? Is this mentioned in the Experimental section?

We thank the Reviewer for his comments regarding the missing discussion of the aerosol particle origins and the air mass characterization in the manuscript.
Information about the calculation of the backward trajectories and the validity of those for both sampling stations (CVAO and MV) was added in the experimental part, section 2.2.2 (page 7, line 12-17) and it reads now: "Back trajectory analyses provided information regarding the origins of air masses. Seven-day back trajectories were calculated on an hourly basis within the sampling intervals, using the NOAA HYSPLIT model (HYbrid Single-Particle Lagrangian Integrated Trajectory, http://www.arl.noaa.gov/ready/hysplit4.html, 26.11.16) in the ensemble mode at an arrival height of 500 m ± 200 m (van Pinxteren et al., 2010); van Pinxteren et al. (2020) provide more details. The calculated backward trajectories are representative for both aerosol particle sampling stations (CVAO and MV)."
Moreover, a short characterization of the air masses was added at the beginning of section 3.2.1, which reads now (page 11, line 24-33):
"*First indications of aerosol particle origin*
To obtain a first indication of the particle origin, that might help to explain the differences in the particle composition concerning amino acids, the particles were associated with the origin of the air masses and with marine and dust tracers. Overall, the CVAO station experienced north-easterly trade winds during this campaign, which are typical for this season within this region (Fomba et al., 2014;van Pinxteren et al., 2020). According to physical and chemical specifications such as the air mass origins, particulate MSA concentrations and MSA/sulfate ratios as well as particulate mass concentrations of dust tracers, aerosol particles predominantly of marine origin with low to medium dust influences were observed. It has to be noted that dust generally influences the supermicron particles to a larger extent than the submicron particles (Fomba et al., 2013). Further information on the classification of the air masses are given in the overview paper of this campaign (van Pinxteren et al., 2020)."

R#2-24) line 7. "The higher complexity in the FAA composition on the submicron aerosol particles could only be determined because the analytical method applied here is able to quantify the individual molecular FAA species." Yes, that is good, and I would encourage the authors to leverage this resolution in their data. See previous comments regarding the comparison of amino acid "profiles" for different sample types and what do the presence of each amino acid tell us.

We thank the reviewer for this comment.
In the revised manuscript version, a clearer focus has been placed on the comparison of amino acid profiles (in terms of individual amino acids and amino acid groups). Above all, this comparison is made between the individual marine compartments in section 3.4.1, 3.4.2 and 3.4.3. After introduction of the individual compartments and their FAA specifications in the individual section 3.1 (seawater), 3.2 (aerosol particles) and 3.3 (cloud water). For detailed answers to the amino acid profiles, please refer to the comments R#2-2, R#2-20, R#2-21, R#2-22 and R#2-27.

R#2-25) line 10. "The composition of FAA on the size-segregated aerosol particle samples with focus on the comparison of the submicron with the supermicron aerosol particles as well as the comparison of aerosol composition with the seawater composition will be discussed more detailed in section 3.4." Please simplify this sentence.

During the revision of the manuscript, this reference to section 3.4 was removed.

R#2-26 )3.2.2 Size-segregated aerosol particles at the mountain station (MV) This seciton reports particulate matter (PM) masses for both the MV and CVAO stations. The Experimental section only reports particle volume measured for cloud water sampling at Mount Verde, and not at CVAO, with liquid water content (LWC) somehow derived. I am in favor of reporting what is measured directly. Here, is particle mass dervied from the particle volume measurements or from the mass of filter samples (at CVAO) and cloud water (at MV) recovered? I assume there was a particle volume monitor at both CVAO and MV. I encourage plotting all particle volume data against other particle measurements like FAA. Does FAA abundance track particle volume?

We thank the reviewer for this comment. The measurement technique of the PM was added in the section 2.2.2 and it reads now (page 7, line 11-12): "The particulate mass (PM) of the aerosol particle samples was determined by weighing the filter before and after sampling (van Pinxteren et al., 2015)."
We have explicitly outlined all parameters that were measured in the aerosol particles at both, the CVAO and the MV (e.g. PM, WSOC, $Na^+$ and MSA) in the Table S11 in the SI and referred to this Table in the manuscript.
We also refer to the online particle size distributions (PSND), which were also taken at both sampling stations (CVAO and MV) during the campaign. In the revised manuscript it reads as follows (page 15, line 16-19): "Additional online measurements of particle size number distributions (PSND) at the CVAO and the MV, described in Gong et al. (2020) were in good agreement with one another during cloud-free times. This indicated that, for cloud-free conditions, the aerosol particles measured at ground level represented the aerosol particles at cloud level, i.e. the aerosol particles within the marine boundary layer were well mixed."
Moreover, in the manuscript, in section 3.2.2 we compared the chemical composition of the aerosol particles at the CVAO and at the MV. It reads now (page 15, line 23-32): "The particles at the MV exhibited lower particle masses, as well as lower concentrations of the aerosol particle constituents. The decrease in concentrations of ∑FAA, PM, sodium, MSA and WSOC was reduced by a factor of three to four regarding the submicron aerosol particles. However, no uniform depletion ratio between their concentration at the CVAO and the MV was found for the supermicron aerosol particles (Table S11). While the PM of the supermicron particles was reduced by a factor of four at the MV (similar to the submicron aerosol particles), sodium and WSOC were depleted more strongly (factor of 11-12) compared to their respective concentrations at the CVAO. This suggests

that the submicron particles were rather uniformly affected and depleted, likely by cloud processes, while the supermicron particles were influenced by clouds, and potentially other sources, in a non-uniform way. Nevertheless, the abundance of the marine tracers (sodium, MSA), together with the presence of FAA in the aerosol particles (which mainly had a similar composition compared to the oceanic and ground-based particulate FAA) indicated an oceanic contribution to the aerosol particles at cloud level."

page 13
R#2-27) line 24. "In cloud water samples with _FAA <65 ng m-3, usually Gly was dominant followed by Ser. Cloud water samples with _FAA >290 ng m-3 showed a higher complexity in the FAA composition, especially towards the end of the campaign, including the appearance of Asp." Did the relative abundances of the FAA vary, indicating different FAA profiles, or were they similar, indicating a consistent FAA profile? What was the profile of hydrophobic, hydrophyllic, and neutral amino acids and how did that compare to the aerosol samples?

We thank the reviewer for this comment. Not only the FAA concentrations in cloud water varied during the campaign, but also the composition of these. Samples with a total FAA concentration of <65 ng m$^{-3}$ showed a low variance in composition, because Gly was dominant, followed by Ser. A higher complexity of the FAA composition could be found at FAA concentrations in cloud water with >290 ng m$^{-3}$. Besides Gly and Ser, Ala and Asp played an important role and Thr, Leu and Ile also contributed to the FAA complexity. Thus, it is not possible to speak about a uniform FAA profile over the whole campaign, but a certain variance was observed.
Looking at the amino acid classes in the cloud water, it can be seen that the neutral FAA were dominant in the first part of the campaign (27/09/2017-05/10/2017) and towards the end of the campaign (06/10/2017-08/10/2017) the proportion of hydrophilic amino acids increased significantly. Similar to the observations in the submicron aerosol particles at CVAO. However, here above all GABA was higher concentrated FAA in submicron aerosol particles at the CVAO. But GABA, for example, was not detected in cloud water.
A detailed comparison of the individual amino acids/amino acid classes in the individual compartments, also with regard to aerosol particles and cloud water, can be found in sections 3.4.1, 3.4.2 and 3.4.3.
In order to clarify and discuss the composition of the individual amino acids and amino acid groups in the cloud water sample, Figure 3 was extended and the hydrophilic, neutral and hydrophobic amino acid groups were clearly pointed out. The variability of FAA composition in cloud water of the individual amino acids and amino acid classes were addressed as follows in the revised manuscript (page 16, line 16 – page 17, line 4): "In cloud water samples with ∑FAA <65 ng m$^{-3}$, Gly was usually dominant, followed by Ser. However, cloud water samples with ∑FAA >290 ng m$^{-3}$ showed a higher complexity in FAA composition, including the concentrations of Asp and Ala. Other abundant FAA were Thr, Leu and Ile. In terms of the hydropathy classification, the first part of the campaign (27/09/2017-5/10/2017) was dominated by neutral FAAs, whereas a sudden increase of the hydrophilic FAAs was observed in its second part (06/10/2017-08/10/2017). Comparative studies on the FAA composition of cloud water in the marine environment are lacking, but especially in the second part of the campaign, it pointed to a local marine (biogenic) influence."

R#2-28) Were the sodium, sulfate, and MSA measurements made on cloud water also made on the aerosol samples? Other than FAA (in section 3.4) which other data are common to the two sample types - aerosol and cloud water - that would allow us to compare them?

Sodium, sulfate and MSA measurements were made for both, cloud water and aerosol particle samples. The results of the measurements of inorganic ions in cloud water are listed in Table S12 and are discussed in sections 3.3 and 3.4. For aerosol particles at the CVAO, the results of inorganic ion measurements are listed in Table S8 and are discussed in section 3.2. For the comparison

between the aerosols of both stations (CVAO and MV) regarding inorganic ions and PM, the results are listed in Table S11 and are also discussed in section 3.2.

In the manuscript we addressed this topic in section 3.3. It reads now (page 16, line 9-13): "The inorganic marine tracers in cloud water (Na$^+$: 5.7 µg m$^{-3}$, MSA: 25.1 ng m$^{-3}$, Table S12) were also present in higher concentrations compared to the aerosol particle samples at the CVAO (submicron: Na$^+$: 72.3 ng m$^{-3}$, MSA: 6.0 ng m$^{-3}$) and the MV (submicron: Na$^+$: 17.0 ng m$^{-3}$, MSA: 1.8 ng m$^{-3}$, Table S11). The concentrations of cloud water sulfate (average: 2.9 µg m$^{-3}$, Table S12) and sodium were higher than in cloud water samples, collected at East Peak in Puerto Rico, which can be seen in Gioda et al. (2009)."

and on page 17, line 9-12: "The presence of the marine tracers (sodium, MSA) in cloud water supports a coupling to oceanic sources. In addition, the majority of low-level clouds were formed over the ocean and ocean-derived components are expected to have some influence on cloud formation (van Pinxteren et al., 2020). Nevertheless, contributions from the desert and other non-marine sources cannot be excluded."

And on page 2, line 2-3: "The abundance of inorganic marine tracers (sodium, methane-sulfonic acid) in cloud water suggests an influence of oceanic sources on marine clouds."

R#2-29) page 15
line 10. "the reactivity/ mean life time _ of the amino acids" Please explain.

A definition of the mean life time was added in the mansucript. Now it reads: "The mean life time τ of the individual amino acids depends on the pH-dependent rate constant k and the OH radical concentration of the different atmospheric scenarios (SI, Eq. (3))." (page 18, line 11-13)
Additionally in the SI (page 17) the equation for the calculation as well as a short discussion about the mean lifetime was added.

R#2-30) line 22. "The mean lifetime _ of Glu (remote aerosol case: 0.02 d" Thats 29 minutes.
Is that considered long?

Comparing the mean lifetime τ of the individual amino acids in the remote aerosol case (valid for conditions at the CVAO) listed in Table S13, it can be shown that amino acids with a mean lifetime τ of 1.20 h (Ala) and 0.48 h (Gly, Glu) have a comparatively longest lifetime τ. Whereas some amino acids have a much shorter mean lifetime with e.g. 0.007 h (Tyr) or 0.014 h (Phe). The estimation method for the mean lifetime is discussed in detail in the SI on page 17. Thus, it can be concluded that when comparing FAA under remote aerosol conditions, FAA with e.g. a mean lifetime of 0.48 h have a longer mean lifetime than most other FAA.

We have integrated the discussion of mean lifetime τ into the discussion about Gly (τ: 0.48 h), because Gly is often used as a long-range tracer with comparatively high stability (within the FAA). This can be read as follows in the revised manuscript (page 20, line 10-16): "Compared to other amino acids, Gly and Ser have a very low atmospheric reactivity (McGregor and Anastasio, 2001) and therefore a higher mean lifetime τ (Gly: 0.48 h, Ser: 0.24 h; remote aerosol case, Table S13). Due to its atmospheric stability, Gly is proposed as an indicator for long-range transport (Barbaro et al. (2015) and references therein) and has a very low atmospheric reactivity (McGregor and Anastasio, 2001). However, our results clearly show that Gly and Ser are also present in seawater to a high extend, likely resulting from the siliceous exosceleton of diatom cell walls (e.g. Hecky et al. (1973)). Hence, besides long-range transport, a transfer from the ocean via bubble bursting might be an additional likely source of the stable, long-lived FAA in the atmosphere."

R#2-31) line 30. "The presence of GABA on the submicron aerosol particles pointed out that (marine) microorganisms were present on the aerosol particles and produced GABA via microbiological decarboxylation of Glu." Until the authors demonstrate that GABA cannot exist outside of a (marine) microorganism, this statement is unfounded. The opposite is a safe assumption: that any compound produced my marine microorganisms will also be found in the sewawater, either by adtive release by the living microorganism

or via release of the dead microorganism (residence times will vary).

We have carefully revised the discussion on hydrophilic FAA, including GABA. Our interpretation approach is based on previous studies regarding GABA in the marine environment (Dauwe et al., 1999;Engel et al., 2018) as already introduced in the Introduction (page 3, line 13/14). They reported GABA as an indicator for microbiological OM degradation and GABA can therefore be considered as a microbiological proxy in aerosol particles in the marine environment. Thereby GABA can be actively released by microorganisms on the aerosol particles or passively, e.g. by the death of the microorganisms. Based on these previous studies, the finding that the air masses were mainly marine (indicating non-marine sources of minor importance) and the high GABA concentrations on the submicron aerosol particles at CVAO, we have addressed the discussion in the revised manuscript as follows (page 18, line 17-33): "A conspicuous finding is the high concentration of GABA, which is present exclusively in the submicron aerosol particles (B1 and B2: 0.05-0.42 µm) at the CVAO. Despite the relatively high LOQ of GABA in seawater (Table S1), a major abundance of GABA in seawater would be detectable. GABA is a metabolic product of the decarboxylation of Glu, which has been detected in all marine compartments. Furthermore, it can be produced by microorganisms (Dhakal et al., 2012) and is considered as an indicator for the microbiological decomposition of OM (Dauwe et al., 1999;Engel et al., 2018). The abundance of GABA on the submicron aerosol particles suggests that (marine) microorganisms were present on the aerosol particles and likely produced GABA via microbiological decarboxylation of Glu. Microbial processes on marine particles have recently been reported by Malfatti et al. (2019). The authors observed a diverse array of microbial enzymes transferred from the ocean into the atmosphere with an even higher activity on the particles compared to seawater. On this basis, they hypothesized that active enzymes can dynamically influence the composition of marine aerosol particles after ejecting from the ocean. The high GABA concentrations on the aerosol particles reported here are well in line with this hypothesis. Interestingly, GABA was not detected in cloud water samples, although bacteria were found during the campaign in cloud water (van Pinxteren et al., 2020) whose presence has been reported in the literature (Jardine, 2009;Vaïtilingom et al., 2013;Jiaxian et al., 2019). It remains speculative whether GABA was degraded in cloud water despite its rather long lifetime (remote cloud case: 28.8 h, Table S13) or whether it was not produced by the bacteria in cloud water."

R#2-32) page 17
line 2. "the presence of bacteria in cloud waters has been reported in the literature
(Jiaxian et al., 2019)." Microorganisms have been documented in the air since at least
Darwin's HMS Beagle voyages so a few more citations here would be appropriate. A
few of note:
Jardine, B. Between the Beagle and the barnacle: Darwin's microscopy, 1837-1854.
Stud. Hist. Philos. Sci. Part A 40, 382–395 (2009).
Salisbury, J. H. On the Cause of Intermittent and Remittent Fevers. Am. J. Med. Sci.
51–75 (1866).
M. Vaïtilingom et al., Potential impact of microbial activity on the oxidant capacity and
organic carbon budget in clouds. Proceedings of the National Academy of Sciences of
the United States of America. 110, 559–564 (2013).

Following the comment of the reviewer we included more citations regarding bacteria in cloud water as recommended. Now it reads (page 18, line 29-31): "Interestingly, GABA was not detected in cloud water samples, although bacteria were found during the campaign in cloud water (van Pinxteren et al., 2020) whose presence has been reported in the literature (Jardine, 2009;Vaïtilingom et al., 2013;Jiaxian et al., 2019)."

R#2-33) 3.4.2 Neutral and hydrophobic amino acids How do the surface activities vary across
the different amino acids? Does abundance in aerosol correlate to surface activity? Or
a combination of surface activity and reactivity/lifetime?

We thank the reviewer for his comment. To consider the surface activity of each individual amino acid, the $K_{OW}$, the TPSA and the density as listed in Table S9 were considered. These parameters were used to study the variation of the atmospheric FAA concentration on the aerosol particles. However, these simple physico-chemical parameters could not explain this variance by statistically relevant correlations.

In addition to the surface activity parameters, the mean lifetime $\tau$ of the aerosol particles (Table S13) was also considered, as proposed, to study the variance of the FAA concentrations on the aerosol particles. However, no statistically relevant correlations were found here either. But for example, Gly, with a long mean lifetime $\tau$, occurs dominantly on the aerosol particles. Aromatic amino acids with a shorter mean lifetime $\tau$, were not found on the aerosol particles.

Therefore, neither the surface activity parameters nor the mean lifetime $\tau$ of the individual amino acids are suitable to explain the variance of FAA concentrations on the aerosol particles in the marine region investigated here by statistically relevant correlations.

These points were addressed as follows in the revised manuscript (page 14, line 15-18): "In addition, we considered further FAA physico-chemical parameters such as the octanol-water partition coefficient ($K_{OW}$), the topological polar surface area (TPSA), which describes the surface activity, and the density (Table S9) to describe the concentration changes. However, no statistically relevant correlations between the FAA concentration or composition and physico-chemical parameters were found here either."

page 18.
R#2-34) line 13. "a possible transport from other than marine sources is included in this parameter." The language is (not on purpose) vague and should clearly state that the waters measured and used for the enrichment factors have not been demonstrated to be the source of the aerosols measured.

As mentioned above (response to comment R#2-1b) we have carefully revised the discussion of the $EF_{aer}$, showed the uncertainties related to calculating the $EF_{aer}$ in an open system.

R#2-35) line 16. "Regarding the transfer of OM from the ocean into ambient aerosol particles, solely organic carbon as a sum parameter has been regarded to date and no distinction of single organic matter classes for ambient measurements has been performed." Although not written altogether clearly, this statement seems to state that compound classes nor compounds have been resolved from ambient aerosol. This is false as there is a number of studies that have accomplished this. See:
Molecular diversity of sea spray aerosol particles: Impact of ocean biology on particle composition and hygroscopicity RE Cochran, O Laskina, JV Trueblood, AD Estillore, HS Morris, ... Chem 2 (5), 655-667
Quinn, P. K., Collins, D. B., Grassian, V. H., Prather, K. A., & Bates, T. S. (2015). Chemistry and Related Properties of Freshly Emitted Sea Spray Aerosol. Chemical Reviews. American Chemical Society. https://doi.org/10.1021/cr500713g

As part of the restructuring of the discussion part related to the $EF_{aer}$ (review comment R#2-1b), section 3.4.4, this sentence was removed from the manuscript

R#2-36) Figure 5. Please include in the figure the size range for each Berner stage.

Following the comment of the reviewer we included in Figure 5 the size range of each Berner stage. Figure 5 can be found on page 21.

R#2-37) page 19.
line 7. "Previous studies showed that organic material ejected into the atmosphere during bubble bursting, resulting in sea spray aerosol particles containing similar organic material to that of the SML (Russell et al. (2010);Cunliffe et al. (2013) and references

therein)." This - the basics of sea spray aerosol formation - need to be brought up later
and the investigation of hydrophyllic/-phobic amino acids in differnt particle types needs
to be established in this context.

We thank the reviewer for his comment. The current state of knowledge on primary marine aerosol formation by the bubble bursting process (incl. jet and film droplets) was briefly summarized in section 3.4.4, since this section deals with a possible transfer of FAA from the ocean into the atmosphere.
It should be noted that our ambient investigations cannot provide detailed mechanistic investigations of the bubble transfer. For this purpose, tank experiments under controlled conditions are necessary (that are currently performed in our group).
However, due to the FAA composition in the SML and on the submicron aerosol particles and the similar percentage contribution of FAA to DOC and WSOC in these two compartments, we suggested that film droplets contributed to the transfer of FAA.
The summary of the state of knowledge on primary marine aerosol formation via bubble bursting and the proposed interpretation of our observation is now included on page 22, line 10-19, and reads as follows in the revised manuscript: "Previous studies have shown that OM ejected into the atmosphere during bubble bursting, results in the formation of sea spray aerosol particles containing OM similar to SML (Russell et al. (2010);Cunliffe et al. (2013) and references therein). Especially the film droplets have been reported to be enriched in OM and are suggested to transfer OM from the SML onto submicron aerosol particles (Wilson et al., 2015). The supermicron aerosol particles tend to form from the larger jet droplets and thus represent the ULW composition (Blanchard, 1975;Wilson et al., 2015). We cannot derive mechanistic transfer characterizations from the ambient measurements performed here. Nevertheless, the constant FAA enrichment in the SML together with the strong FAA enrichment in the submicron aerosol particles strongly suggest that film droplets form the submicron particles. However, Wang et al. (2017) showed that jet drops (which transfer OM from the ULW) also have the potential to contribute significantly to the formation of submicron sea spray aerosol particles, so, jet droplets can also contribute to FAA formation."

R#2-38) line 29. "In situ-formation of FAA in cloud water, maybe due to biogenic formation or enzymatic degradation of proteins, selective enrichment processes as well as pH dependent chemical reactions might be potential sources." site Malfatti, F., Lee, C., Tinta, T., Pendergraft, M. A., Celussi, M., Zhou, Y., : : : Prather, K. A. (2019). Detection of Active Microbial Enzymes in Nascent Sea Spray Aerosol: Implications for Atmospheric Chemistry and Climate. Environmental Science and Technology Letters, 6(3), 171–177. https://doi.org/10.1021/acs.estlett.8b00699

We thank the reviewer for pointing out this interesting paper. In fact, the results, suggesting microbial activity on aerosol particles, shown by the transfer of enzymes, fits well with the observations made here and might explain the abundance of GABA on the aerosol particles. Please also note our reply to R#2-31.
We included this reference in the revised manuscript in the following context:
"The abundance of GABA on the submicron aerosol particles suggests that (marine) microorganisms were present on the aerosol particles and likely produced GABA via microbiological decarboxylation of Glu. Microbial processes on marine particles have recently been reported by Malfatti et al. (2019)." (page 18, line 23-25)
and "Altogether, the in-situ formation of FAA in cloud water by chemical processes in the cloud or by atmospheric biogenic formation or enzymatic degradation of proteins, as proposed by Malfatti et al. (2019), as well as by selective enrichment processes and pH dependent chemical reactions might be potential sources." (page 17, line 14-17)

**Technical Corrections**

Introduction.

R#2-39) page 2. line 8. "surface global ocean" to "global surface ocean" line 16. "utilizable sources of nitrogen" to "utilizable FORMS of nitrogen"

Due to the revision of the Introduction and also following the suggestions from reviewer 2, we focused stronger on the FAA from the beginning and therefore general parts (as the one mentioned by the reviewer) were deleted from the manuscript.

On page 2, line 24-26 the suggested correction was implemented. Now it reads: "Despite their attribution to proteins the FAAs are better utilizable forms of nitrogen instead of proteins for an aquatic organism such as phytoplankton and bacteria (Antia et al., 1991;McGregor and Anastasio, 2001)."

R#2-40) page 3. line 31. "underline seawater" to "underlying seawater"

Following the commentary of reviewer 1 (R#1-4), we have used the abbreviations in the manuscript continuously after the introduction: This sentence (page 3, line 31-32) reads now: "So, the aim of the present study is to investigate the occurrence of FAA in the marine environment regarding all important compartments; i.e. the ULW, the SML, the aerosol particles and finally cloud water in the remote tropical North Atlantic Ocean at the Cape Verde Atmospheric Observatory (CVAO)."

R#2-41) page 4. line 4. "as proxies or tracer" to "as proxies or tracerS"

Following the comment of the reviewer we changed the sentence on page 4, line 4/5. And it reads now: "Finally, the potential of individual FAA as proxies or tracers for specific sources of aerosol particles and cloud water in the tropical marine environment is outlined."

R#2-42) In the Experimental section perhaps change "analytics" to "analyses"

We agree with the comment of the reviewer and changed in the experimental section "analytics" to analyses, e.g. page 5, line 25/26: "2.2 Analyses" and "2.2.1 Seawater analyses"; page 6, line 25 "2.2.2 Aerosol particle filter analyses" and page 7, line 19: "2.2.3 Cloud water analyses".

R#2-43) 2.2.1 Seawater sample analytics Was the standard addition method applied to samples to assess for recovery efficiency of the entire process?

The standard addition method was also used to evaluate the recovery efficiency in the development of desalination methods and analytical measurement methods. The recovery efficiency stated here is based on the fact that a defined concentration of amino acids was added to a seawater sample prepared from milli-Q water using sea salt standard (Sigma-Aldrich, Germany) and this sample was treated using the same procedure as the real seawater samples. The FAA concentration in the seawater sample was then determined after sample preparation. To consider the recovery efficiency, the percentage of the measured concentrations to the added FAA concentration was calculated. The recovery rate thus determined is consistent with tests using standard addition methods and proved to be less complex and more practical.

R#2-44) page 6. line 23. "All here presented values" to "All values presented here" line 29.

Following the comment of the reviewer we changed the sentence on page 7, line 4 to "All values presented here for aerosol particle samples are field blank corrected."

R#2-45) "The cloud water samples were operated the same as seawater samples". change "operated" to "handled" or "processed".

We agree with the reviewer's comment and changed the sentence on page 7, line 20/21 to "The cloud water samples were processed the same as seawater samples for the analysis of DOC/TDN and inorganic ions (section 2.2.1)."

R#2-46) line 31. "syringe filters filters" to "syringe filters" or "syringe tip filters"

We agree with the reviewer's comment and changed the sentence to "After the filtration with 0.2 µm syringe filters (Acrodisc-GHP; 25 mm, Pall Corporation, New York, USA), an aliquot of the prepared cloud water was derivatized based on the AccQ-Tag™ precolumn derivatization method (Waters, Eschborn, Germany)." (page 7, line 22-24)

R#2-47) page 10. line 12. Remove "It is obvious that" as it is confusing language.

Following the comment of the reviewer we removed "it is obvious that". Now it reads (page 12, line 4-7): "Whilst the concentration ∑FAA varied between 0.2 ng m$^{-3}$ (6/10/2017) and 1.4 ng m$^{-3}$ (22/09/2017) in the supermicron size range, the highest atmospheric concentrations of ∑FAA were found in the submicron aerosol particles (mean of 3.2 ng m$^{-3}$) compared to the supermicron ones (mean of 0.6 ng m$^{-3}$)."

R#2-48) line 14-15.
Reword. Use "neither" instead of "both".

Following the comment of the reviewer we changed "both" to "neither". Now it reads (page 12, line 33/34): "∑FAA included all investigated amino acids (listed in 2.2.1) except for Met and Gln, analytes which were neither detected in the size-segregated aerosol particle samples."

R#2-49) page 12 line 32. "aerosol particles" is redundant. aerosols are particles. say just "aerosol" or "particle", here and elsewhere.

Since the term "aerosol" refers to the entirety of a gas with the particles suspended in it, we would prefer to use the term "aerosol particles" in the manuscript and in the SI. The term "aerosol particle" is also common in other publications, for example (Cochran et al., 2017;Forestieri et al., 2016;Frossard et al., 2019;Koulouri et al., 2008).

R#2-50) page 17. line 7. "Neutral amino acis" to "acids"

We changed the typo. Now it reads (page 20, line 8/9): "Neutral amino acids were generally the amino acid group with the highest concentration in all investigated marine compartments, accounting for more than 50% of the FAA total (Fig. 4a-d)."

R#2-51) line 10. "A further explanation approach" remove "approach"

Following the comment of the reviewer we restructured the sentence. Now it reads (page 20, line 15/16): "Hence, besides long-range transport, a transfer from the ocean via bubble bursting might be an additional likely source of the stable, long-lived FAA in the atmosphere."

**Additional changes performed by the authors**

When discussing the mean lifetime $\tau$ of individual amino acids (section 3.4 and Table S13), the unit of $\tau$ was changed from days (d) to hours (h).

The acknowledgement was also revised to thank the people from the OSCM. The added sentence is now as follows: "We further acknowledge the professional support provided by the Ocean Science Centre Mindelo (OSCM) and the Instituto do Mar (IMar)" (page 23, line 25-26)

The measured data were published on PANGAEA. The data availability statement was therefore updated and reads as follows: "Data availability. The data are available through the World Data Centre PANGAEA under the following link: https://doi.pangaea.de/10.1594/PANGAEA.914220." (page 23, line 14/15)

The previous citation of van Pinxteren (submitted 2019) was updated to van Pinxteren et al. (2020) in the revised manuscript and supporting information.

[revised manuscript text omitted]

---

## Referee Report (RR1)

Review of revised manuscript: acp-2019-976-manuscript-version4.pdf

General comments

I read in the response to reviews language along the lines of "this amino acid can
be ejected by the ocean and end up in sea spray aerosol" and this implies a
misunderstanding of sea spray aerosol basics, or even aerosol basics. Compounds are
emitted from the ocean either in sea spray aerosol or in gases.  Gases can condense
onto aerosols. Otherwise, a compound found in sea spray aerosol was ejected from the
ocean in a sea spray aerosol. It (the compound) didn't get ejected from the ocean in
some ambiguous, undefined phase and then somehow "end up" in (a) sea spray aerosol.

What I also consider a misunderstanding of sea spray aerosol basics is that the
presence in sea spray aerosol of a substance produced by microorganisms does not
mean a microorganism was present in the aerosol. The substance can be produced in
the seawater and then transfer in sea spray aerosol without the microorganism
present.

While implying the presence of microorganisms in submicron aerosol, the authors may
want to consider and mention that a 1 um (or, here, 1.2 um) size cutoff excludes, by
size, many microorganisms, even many marine bacteria. Perhaps they should state
which marine microorganisms they think might be in their submicron aerosol and what
are the size ranges of those microorganisms. For example, viruses are small enough,
are abundant in the ocean, and are well documented in sea spray aerosol. And since
the aerosol isn't exclusively sea spray aerosol, are there non-marine microorganisms
that may be present?

I commend the authors on making it clear in the Introduction that amino acids in
aerosols can come from diverse sources. As such, I think the paper needs to clearly
explain how amino acids in aerosols can be attributed to sea spray aerosol.

Please make sure all tables clearly state the sample type to which the data
correspond. There is at least one table in the supplemental material that does not
do this.

Specific comments

Introduction

p. 3 line 13.
GABA (γ-aminobutyric acid) was referred to as an indicator for the microbiological

decomposition of OM (Dauwe et al., 1999;Engel et al., 2018) and is probably used as a microbiological proxy in aerosol particles.

I am guessing you want to remove "probably", cite the studies that have used GABA in this capacity, and then perhaps make it a little more clear on how GABA was used.

p. 3 line 14.
free and combined amino acids are introduced but not defined. Perhaps don't specify "free" or "combined" until you want to also define them and just refer to "amino acids" (no free or combined) here.

p. 3 line 16.
There is a disconnect in this sentence at "...what Pommie..."

2.1.2

p. 5 line 4
"By heating the sampled air, the high relative humidity of the ambient air before collecting the aerosol particles was reduced to 75-80%."
Was reduced to 75-80% from what? Plunging tanks have RH about 85%. How high was the ambient RH?

p. 5 line 12
A minor note on: "the Berner impactors ran continuously, thus the impactor on the MV sampled aerosol particles also during cloud events. However, due to the pre-conditioning unit, the cloud droplets were efficiently removed before the aerosol particles were collected on the aluminium foils."
If functioning as intended, the pre-conditioning unit (dryer) would only remove water and any species more volatile than water, and should leave a dried aerosol. These samples could be analyzed and compared against the cloud water samples collected at the same time.

2.2 Analyses

Could you please more clearly define & describe the filtration steps involved in the FAA process? For example, the DOC/TDN seawater samples start with a 0.45 um syringe tip filtration. Was this also done for the subsamples analysed for FAA? For both seawater and aerosol samples?
I don't think FAA vs CAA is ever clearly defined in the manuscript. Does FAA strictly include singular, individual amino acids, not linked to other amino acids in a protein? If not, then what is the upper size range of proteins and proteinaceous particles that this method includes in the analysis of FAA?

3.2.1 Size-segregated aerosol particles at the CVAO

p. 11 line 28

"According to physical and chemical specifications such as the air mass origins, particulate MSA concentrations and MSA/sulfate ratios as well as particulate mass concentrations of dust tracers, aerosol particles predominantly of marine origin with low to medium dust influences were observed."

I think this work adds a lot of strength to the manuscript. But I also think the authors should spend just a few more sentences on this, stating what are the values of the different indicators that imply sea spray aerosol or desert dust. Without doing that, the authors seem to say "We have the data. Trust us on our interpretation." which I don't think is the best approach, especially with the study site being located in a part of the atmosphere that can contain high dust levels, being just offshore and downwind, at times, from the Sahara Desert (and at times downwind of other deserts and large dust sources). Plotting the data along with nominal desert dust levels and sea spray aerosol levels from the literature would be an improvement on tabulated data in the supplemental information.

p. 11 line 30
"It has to be noted 30 that dust generally influences the supermicron particles to a larger extent than the submicron particles (Fomba et al., 2013)."
Perhaps state why this has to be noted (I assume it comes up later on).

Free amino acids in size-segregated aerosol particles: Composition

p. 14 line 26.
"According to the conclusions by Barbaro et al. (2015), the relatively high content of hydrophilic FAA found here points at least at some influence of local oceanic sources."
If "Barbaro et al. (2015) reported that hydrophilic components were predominant (60 %) in locally produced marine Antarctic aerosol particles" and here hydrophilic FAA in submicron = 15% and supermicron = 0%, then I would clarify that only the periods of hydrophilic FAA much higher than the mean values would be deemed by Barbaro el al. (2015) as indicative of sea spray aerosol.

3.2.2 Size-segregated aerosol particles at the MV

p. 15
The following statements do not agree:
"The submicron aerosol particles at the MV had an averaged $\Sigma$FAA concentration of 1.5 ng m-3 (0.8-1.9 ng m-3) and were about three times lower compared to the $\Sigma$FAA concentration at the CVAO." (line 13)
"the aerosol particles measured at ground level represented the aerosol particles at cloud level" (line 18)
I am guessing that the implication being made here is that it is the same particle population sampled at CVAO and MV but that aerosol ageing or processing is happening between the two sites and causing the significant change in FAA concentration. This is a guess because it is not stated. What are the wind speeds and transport times between the two sites?

p. 15 line 18
"the aerosol particles measured at ground level represented the aerosol particles at cloud level, i.e. the aerosol particles within the marine boundary layer were well mixed."
There was no aerosol sampling at ground level, correct? CVAO sampling ocurred at 30 m and MV at 744 m, right?
The argument made in the methods section is vague but I think it is that 30 m (CVAO) and 744 m (MV) are in the same layer of the atmoshpere. Its not clear if this is within the internal boundary layer, or above it, within the marine boundary layer, or otherwise. Please clarify, including marine boundary layer vs. internal boundary layer.

p. 15 line 20
"The concentration and composition of the aerosol particles can therefore be affected by the clouds that formed and disappeared consistently during the sampling period of the aerosol particles at the Mt. Monte Verde"
I like that analytes detected in cloud water are reported as a concentration per volume of air, instead of per volume of cloud water (LWC), thus providing the aerosol content of the air mass, and being unaffected by varying levels of liquid water content.

p. 15 line ~25

I wonder if it would be better to not discuss cloud processing until after the data from the cloud water have been presented.

3.3 Cloud water samples

p. 16 line 3.
"The individual atmospheric concentration of FAA in cloud water was calculated based on the measured liquid water content (LWC)"
OK. Very good. [concentration in cloud water] x [cloud water volume] / [air volume sampled]

p. 17 line 13
"The reason for the high concentrations of FAA in cloud water (compared to the oceanic and aerosol particle concentrations) remain speculative to date and will be subject of further studies."
Probably the most significant finding of this study is not pursued.

p. 17 line 14
"Altogether, the in-situ formation of FAA in cloud water by chemical processes in the cloud or by atmospheric biogenic formation or enzymatic degradation of proteins, as proposed by Malfatti et al. (2019), as well as by selective enrichment processes and pH dependent chemical reactions might be potential sources."

I am very surprised by the term "in-stu formation of FAA in cloud water" here. Are you (suddenly) claiming that any or all of the amino acids detected in cloud water were created in the atmosphere? None of them existed in the particles when the particles were produced?

I fear a misunderstanding of aerosol basics here, unless this is just a language issue. Sea spray aerosol contains subsamples of seawater constituents that are small enough to become aerosolized; this includes amino acids. Enzymes do not create amino acids; they consist of amino acids and aminopeptidase enzymes cleave (or catalyse the hydrolysis) amino acids from larger molecules. All cloud droplets form on aerosols, thus, from the time they are formed, cloud droplets contain more than just water. If the water from the cloud droplets evaporate, the cloud nuclei remain. Rain will remove the cloud nuclei and other particles below. Was there any rain in the area during or just prior to the sampling periods?

To consider atmospheric processing or ageing, I wanted to know the transport time between CVAO and MV. The downwind distance between them is about 2 km. Wind speed on average was about 5.5 m/s. So the average transport time would be about 6 minutes.

3.4.1. Hydrophilic amino acids

p. 18 line 15
"The hydrophilic amino acids (Asp, Glu, GABA) comprised a significant fraction in the ULW and the SML, as well as in the (submicron) aerosol particles and in cloud water (Fig. 4a-d). They were not detected in the supermicron aerosol particles."
This is relevant to a major theme in sea spray aerosol research: the organic fraction of SSA across SSA sizes. The research would be more valuable if this were discussed.

p. 18 line 22
"The abundance of GABA on the submicron aerosol particles suggests that (marine) microorganisms were present on the aerosol particles and likely produced GABA via microbiological decarboxylation of Glu."
So it is impossible for GABA to transfer from the ocean into the atmosphere in a nascent sea spray aerosol?
Which microorganisms would be small enough to be found in submicron aerosol? Small marine bacteria? These sizes are of the dried particles, so the particle sizes in the atmosphere were larger, and perhaps the aerosol population that were submicron after drying contained supermicron aerosols before drying and were big enough to contain marine bacteria.

p. 18 line 24
The connection to Malfatti et al. (2019) here is confusingly redundant.

p. 19 Figure 4
Thank you for plotting the data together in this figure. It appears the cloud water and seawater have similar profiles. Are there others?

3.4.2 Neutral and hydrophobic amino acids

p. 20
Some very low lifetimes are reported for amino acids in the atmosphere: ~0.5-1 hour.
Relate this to the sampling period lengths and the duration of time between sampling
and analyses. Did the  samples/analytes need to be isolated from air/atmosphere
between collection and analyses? Should we question the amino acid residence times?
Or are they accurate and is it possible that the amino acids detected are largely
from intact microorganisms collected in the aerosol?

3.4.4 Transfer of amino acids from the ocean into the atmosphere

p. 21 line 7
"The high similarity concerning the main FAA species within the different
compartments, together with the high concentration of ocean-derived compounds (Na+,
MSA) in the aerosol particles and cloud waters, suggest a coupling between the FAA
in the ocean and the atmosphere."
I don't think this "high similarity" was established. The GABA in submicron CVAO is
a high dissimilarity with SML and ULW.
It would be useful to have FAA profiles of dust (or other aerosol types) for
comparison. If those profiles are very different, then perhaps the profiles
presented in this work would appear more similar.

p. 23 line 8
"FAAs were present in the size range for aerosol particles associated with CCN
activity and cloud water, and might be connected to CCN activity due to their
hygroscopicity and soluble character, but this effect was not investigated here."
It is disappointing to read this. The authors found something very interesting: very
high atmospheric concentrations of amino acids in cloud water, relative to the
ambient aerosol population. But that finding is ignored.

Supplemental Information: acp-2019-976-supplement-version3.pdf

p. 16 Back Trajectory
It is great that you ran back trajectories for all your sampling periods. Having
done so, please share them all. Why not share them all when they support the
science?

---

## Author Response (AR2)

We see this remark under 'Suggestion for revision or reason for rejection:

I am disappointed by the work. It seems the manuscript was written before the data were even acquired; it was preordained. This is especially disappointing to come from established researchers. The first version read like a first draft and I strongly oppose using reviewers as "first draft editors". The response to the initial reviews was confusingly long. Was that a filibuster?
I appreciate the significant work put into the manuscript after the first set of reviews. That work gave more credibility to certain claims made but did not lead to greater findings and conclusions. This is especially disappointing considering the most interesting finding of the research was not pursued. The highlighted findings seem to already have been done - amino acids have previously been found in aerosol, so how does this work advance the field?
More comments attached and also pasted below.

We identify this a personal comment of this particular reviewer which could be addressed strongly but we just want to state that these comments are strange to be found in a reviewer statement of a journal and which are, possibly, foreseen to be published. We are rejecting all these points listed both with regard to their form as well as to their content.

We thank the reviewer for the careful examination of the manuscript and the supporting information. In the following, please find a point-by-point response to the questions and concerns. All references to the manuscript (e.g. page and line numbers) listed in our replies refer to the clean version of the manuscript (without track changes).

**General comments**

R#1-1) I read in the response to reviews language along the lines of "this amino acid can be ejected by the ocean and end up in sea spray aerosol" and this implies a misunderstanding of sea spray aerosol basics, or even aerosol basics. Compounds are emitted from the ocean either in sea spray aerosol or in gases. Gases can condense onto aerosols. Otherwise, a compound found in sea spray aerosol was ejected from the ocean in a sea spray aerosol. It (the compound) didn't get ejected from the ocean in some ambiguous, undefined phase and then somehow "end up" in (a) sea spray aerosol.

The reviewer rightly stated that amino acids are compounds emitted from the ocean in sea spray aerosol. We agree that the mentioned sentence was, unfortunately, not clear and we have reworded it. Now it reads as follows (page 2, line 20-22): "From the ocean, amino acids as part of the class of proteinaceous compounds can be transferred into atmospheric particles via bubble bursting (Kuznetsova et al., 2005;Rastelli et al., 2017)."

R#1-2) What I also consider a misunderstanding of sea spray aerosol basics is that the presence in sea spray aerosol of a substance produced by microorganisms does not mean a microorganism was present in the aerosol. The substance can be produced in the seawater and then transfer in sea spray aerosol without the microorganism present. While implying the presence of microorganisms in submicron aerosol, the authors may want to consider and mention that a 1 um (or, here, 1.2 um) size cutoff excludes, by size, many microorganisms, even many marine bacteria. Perhaps they should state which marine microorganisms they think might be in their submicron aerosol and

are the size ranges of those microorganisms. For example, viruses are small enough, are abundant in the ocean, and are well documented in sea spray aerosol. And since the aerosol isn't exclusively sea spray aerosol, are there non-marine microorganisms that may be present?

We agree that substances can be produced in the seawater and then transferred in sea spray aerosol without the microorganism present there. However, we would like to point out, that previous studies have shown the presence of microorganism on aerosol particles: Aller et al. (2005) concluded that 'Marine aerosols are formed primarily by the eruption of rising bubbles through the sea-surface microlayer (SML), and aerosol formation is the main vector for transport of bacteria and viruses across the air–sea interface.' Pósfai et al. (2003) reported that 'a few bacteria typically occur in all of the samples examined with TEM' (transmission electron microscopy). The study of Pósfai et al. (2003) was able to show single bacterial cells with sizes smaller than 1 µm by electron microscopy (Figure 1, 2a and 2c of the study by Pósfai et al. (2003)). In the study of Ervens and Amato (2020) it was discussed that 'the sizes of bacteria-containing particles usually exceed several hundred nanometers and thus can all be considered CCN'. Jaber et al. (2020) have measured the biotransformation rates of amino acids with four active bacterial strains isolated from atmospheric samples. Furthermore, Xia et al. (2015) investigated the high diversity of bacterial communities on marine aerosols. They found that 19 bacterial orders were present in the aerosol particles. And the results of Rastelli et al. (2017) indicated that '15–25% of the total aerosol viruses and 10–20% of total aerosol prokaryotes were exclusively associated to the fine aerosol fraction (<1.2 µm).'
Even if we cannot make any statements about whether and which microorganisms were present on our size-resolved aerosol particles, previous studies could show that microorganisms can indeed be found on (marine) aerosol particles, even in the submicron range (Rastelli et al., 2017). Whether marine and non-marine microorganisms can be present on the investigated aerosol particles would have to be investigated by further taxo-specific studies with source attribution.

In the manuscript, we addressed the possible presence of microorganisms in the marine environment and their influence on the composition of amino acids in the sections 3.1 and section 3.4.1. In addition, we added the following sentence in the revised manuscript on page 17, line 25-27: "In previous studies, the transfer of microorganisms from the ocean to the aerosol particles could be reported (Aller et al., 2005;Pósfai et al., 2003) and even on submicron marine aerosol particles viruses and prokaryotes were present (Rastelli et al., 2017)."
Furthermore, in the context to the reviewer comment R#1-21, we elucidated this topic in more detail when discussing the occurrence of GABA on the aerosol particles.

R#1-3) I commend the authors on making it clear in the Introduction that amino acids in aerosols can come from diverse sources. As such, I think the paper needs to clearly explain how amino acids in aerosols can be attributed to sea spray aerosol.
Please make sure all tables clearly state the sample type to which the data correspond. There is at least one table in the supplemental material that does not do this.

We agree that this is important to state that the amino acids on aerosol particles can come from different sources. In the revised Introduction we have already clearly stated that amino acids on the aerosol particles can come from different sources and how amino acids can contribute to the sea spray aerosol. This can be read in the MS as follows:
"In general, previous studies have shown that amino acids in aerosol particles can have both natural and anthropogenic sources. Having being detected in volcanic emissions (Scalabrin et al., 2012) and during biomass burning events (Chan et al., 2005;Feltracco et al., 2019), amino acids can be produced by plants, pollens, fungi, bacterial spores and algae (Milne and Zika, 1993;Zhang and Anastasio, 2003;Matos et al., 2016)." (page 1, line 30 – page 2, line 2)

"Although the study and characterization of amino acids are of paramount importance for atmospheric scientists, the true role and the fate of amino acids in the atmosphere are still poorly understood (Matos et al., 2016). Despite several studies of FAAs also conducted in the marine environment, there is still a huge uncertainty to the question whether FAAs are of marine origin or not." (page 3, line 21-24)

"On the other hand, based on a positive correlation between amino acids in seawater and the atmosphere, Wedyan and Preston (2008) pointed out the particulate amino acids in the Southern Ocean to be of marine origin." (page 3, line 25-27).

Moreover, we stated: "Their (FAA) abundance, origin and possible transfer from the seawater as well as their transport within the atmosphere are studied in particular." (page 3, line 33/34).

We also checked the tables in the SI and made sure that the data of the tables can be clearly stated to the different sample types. Therefore, we have added the sampling location (CVAO) to Table S10.

**Specific comments**

Introduction

R#1-4) p. 3 line 13.
GABA (γ-aminobutyric acid) was referred to as an indicator for the microbiological
decomposition of OM (Dauwe et al., 1999;Engel et al., 2018) and is probably used as
a microbiological proxy in aerosol particles.
I am guessing you want to remove "probably", cite the studies that have used GABA in
this capacity, and then perhaps make it a little more clear on how GABA was used.

According to the reviewer's suggestion, we removed 'probably' in this sentence. This sentence reads now as follows (page 3, line 13/14): "GABA (γ-aminobutyric acid) was referred to as an indicator for the microbiological decomposition of OM (Dauwe et al., 1999;Engel et al., 2018) and is used as a microbiological proxy in aerosol particles."
In the study of Dauwe et al. (1999), the author refers to the microbiological formation of GABA and β-alanine from their protein precursors, which was previously described as a degradation state indicator for marine sediments (Dauwe et al. (1999) and references therein). And in the study of Engel et al. (2018) GABA is described as an amino acid, which derived from Glu and has often been used as an indicator for microbial decomposition of OM (Engel et al. (2018) and references therein).

R#1-5) p. 3 line 14.
free and combined amino acids are introduced but not defined. Perhaps don't specify
"free" or "combined" until you want to also define them and just refer to "amino
acids" (no free or combined) here.

We agree and defined 'free' and 'combined' amino acids at the beginning of the introduction. It now reads as (page 2, line 15/16): "They can be divided into free single amino acids (FAA) and combined amino acids (CAA), which include proteins, peptides or other combined forms (Mandalakis et al., 2011)."

R#1-6) p. 3 line 16.
There is a disconnect in this sentence at "...what Pommie..."

According to the reviewer's suggestion, we have reworded this sentence for s better understanding. It now reads as (page 3, line 14-17):" To facilitate the comparison of amino acids in different studies, one

possibility is to group them as regards their physio-chemical properties of amino acids ('hydropathy index' (Kyte and Doolittle, 1982)) as Pommié et al. (2004) suggested based on the partition coefficient between water and ethanol."

R#1-7) 2.1.2
p. 5 line 4
"By heating the sampled air, the high relative humidity of the ambient air before collecting the aerosol particles was reduced to 75-80%."
Was reduced to 75-80% from what? Plunging tanks have RH about 85%. How high was the ambient RH?

Most of the time of the campaign, the ambient relative humidity (RH) at the MV station was ~ 100 % (Gong et al., 2020). By heating the sampled air, its relative humidity was set to 75-80%. We agree with the reviewer that 'was reduced' is the wrong word choice, so we changed this to 'set'. This now reads as follows (page 5, line 4/5): "By heating the sampled air, the high relative humidity of the ambient air before collecting the aerosol particles was set to 75-80 %."

R#1-8) p. 5 line 12
A minor note on: "the Berner impactors ran continuously, thus the impactor on the MV sampled aerosol particles also during cloud events. However, due to the pre-conditioning unit, the cloud droplets were efficiently removed before the aerosol particles were collected on the aluminium foils."
If functioning as intended, the pre-conditioning unit (dryer) would only remove water and any species more volatile than water, and should leave a dried aerosol. These samples could be analyzed and compared against the cloud water samples collected at the same time.

The drying unit in front of the Berner impactors in fact worked as the reviewer states.
At the Mt. Verde station, aerosol particle and cloud water samples were collected only partly within the same sampling interval. The aerosol particle samplers ran for 24 h, whereas the cloud water samples had different collection times (listed in Table S12).

Therefore, to compare amino acid concentration and composition within a similar time period, we defined the 'case study' (section 3.4 including Figure 4) and compared not only the seawater and aerosol particle samples from the CVAO, but also the aerosol samples collected at the Mt. Verde station and a cloud water sample as stated in the manuscript on page 17, line 31-34.
In section 3.4.1, 3.4.2 and 3.4.3 similarities and differences between the different marine compartments (including aerosol particle samples at the MV station and cloud water samples) were discussed in detail.

R#1-9) 2.2 Analyses
Could you please more clearly define & describe the filtration steps involved in the FAA process? For example, the DOC/TDN seawater samples start with a 0.45 um syringe tip filtration. Was this also done for the subsamples analysed for FAA? For both seawater and aerosol samples?
I don't think FAA vs CAA is ever clearly defined in the manuscript. Does FAA strictly include singular, individual amino acids, not linked to other amino acids in a protein? If not, then what is the upper size range of proteins and proteinaceous particles that this method includes in the analysis of FAA?

*Concerning the filtration steps within the sample preparation:*

For seawater samples, the description of the filtration steps is in detail described in the manuscript. For the analysis of inorganic ions and DOC/TDN, the seawater samples were filtered using a 0.45 μm syringe filter as stated in the manuscript on page 5, line 27-29. For the amino acid analysis, the seawater samples were first desalinated, concentrated and then filtered using a 0.2 μm syringe filters as stated on page 5, line 29-34.

The filtration with a 0.2 μm syringe filter are necessary because the (derivatized) amino acids were analyzed using a UHPLC system.

For the aerosol particle samples, an aliquot of the aqueous extract of the aerosol particle samples was filtered using 0.45 μm syringe filters and then used for the analysis of inorganic ions and DOC/TDN as stated in the manuscript on page 6, line 29/30. For the amino acids, a filtration step of the aqueous extract with 0.2 μm filters was also performed, but this handling step was not explicitly mentioned in the manuscript, so we have added the following (page 6, line 32 – page 7, line 1): "The aliquot (1.5 mL) of the aqueous particle extracts for FAA analysis was reduced to several μL with a vacuum concentrator at T=30 °C (miVac sample Duo, GeneVac Ltd., Ipswich, United Kingdom), filtered using 0.2 μm syringe filters and derivatized as well as analyzed using the UHPLC/ESI Orbitrap-MS method as explained in section 2.2.1 for seawater samples."

*FAA vs CAA:*

In this study, FAA refers to single, individual amino acids. For the detailed definition of FAA and CAA we would like to refer to the comment R#1-5. Thus, only FAA that passed through the 0.2 μm syringe filter were considered during this analytical approach. In addition, the UHPLC-Orbitrap-MS method performed included a target analysis for the listed amino acids as stated in the manuscript on page 6, line 1-4), in which not only the retention time of the individual amino acids, but also the *m/z* ratio was used for evaluation.

R#1-10)
3.2.1 Size-segregated aerosol particles at the CVAO
p. 11 line 28
"According to physical and chemical specifications such as the air mass origins, particulate MSA concentrations and MSA/sulfate ratios as well as particulate mass concentrations of dust tracers, aerosol particles predominantly of marine origin with low to medium dust influences were observed."
I think this work adds a lot of strength to the manuscript. But I also think the authors should spend just a few more sentences on this, stating what are the values of the different indicators that imply sea spray aerosol or desert dust. Without doing that, the authors seem to say "We have the data. Trust us on our interpretation." which I don't think is the best approach, especially with the study site being located in a part of the atmosphere that can contain high dust levels, being just offshore and downwind, at times, from the Sahara Desert (and at times downwind of other deserts and large dust sources). Plotting the data along with nominal desert dust levels and sea spray aerosol levels from the literature would be an improvement on tabulated data in the supplemental information.

We thank the reviewer for his comment. Due to the comments of the previous reviews, the discussion of the sources of the aerosol particles in the manuscript has been significantly shortened and is now only briefly and precisely reflected in section 'First indications of aerosol particle origin' (page 11, line 21-30). The detailed discussion concerning the marine (MSA, sulfate, sodium) and dust (titanium, iron) tracers including their concentration and threshold values as well as the backward trajectories can be found in the SI in context with Table S8 in section 'Aerosol particles: dust and marine tracers' (SI page 10/11). In order for readers to understand the interpretation of our data and thus the main statements

listed in the MS, we explicitly refer to the discussion section in the SI and also to the overview paper of the campaign (page 11, line 27-30): "The dust and marine tracers of the aerosol particles considered here are discussed in more detail in SI (Table S8 and in 'aerosol particles: dust and marine tracers'). Further information on the classification of the air masses and distinct concentrations of dust tracers are given in the overview paper of this campaign (van Pinxteren et al., 2020)."

R#1-11) p. 11 line 30
"It has to be noted that dust generally influences the supermicron particles to a larger extent than the submicron particles (Fomba et al., 2013)."
Perhaps state why this has to be noted (I assume it comes up later on).
Free amino acids in size-segregated aerosol particles: Composition

We agree with the reviewer that this sentence was not sufficiently explained in this context. Therefore, we have included the following note in the section "Aerosol particles: dust and marine tracers" (SI, page 11/12), which deals in more detail with the discussion about the origin of the investigated aerosol particles. The discussion about the dust tracers reads now as follows: "In order to estimate potential dust influences during the campaign, mineral dust tracer as iron (Fe) and titanium (Ti) were considered. Considering the time-resolved trend of Fe and Ti values in the size-segregated aerosol particle samples, it could be noticed that the lowest concentration of Fe (7.0 ng m$^{-3}$, submicron size range) was detected on 4/10/2017 (Fe$_{(PM10)}$: 117.2 ng m$^{-3}$). The Ti concentration on that day was 0.1 ng m$^{-3}$ in the submicron aerosol particles and 9.4 ng m$^{-3}$ for PM$_{10}$. When it comes to typical marine background concentrations of trace metals at the CVAO for PM$_{10}$ aerosol particles with <25 ng m$^{-3}$ for Fe and <6 ng m$^{-3}$ for Ti (Fomba et al., 2013), especially the submicron aerosol particles on e.g. 4/10/2017 showed very low or no mineral dust influences. Moreover, it has to be noted that dust generally influences the supermicron particles to a larger extent than the submicron particles (Fomba et al., 2013)." (SI: page 11, line 17 – page 12, line 1)

R#1-12 p. 14 line 26.
"According to the conclusions by Barbaro et al. (2015), the relatively high content of hydrophilic FAA found here points at least at some influence of local oceanic sources."
If "Barbaro et al. (2015) reported that hydrophilic components were predominant (60 %) in locally produced marine Antarctic aerosol particles" and here hydrophilic FAA in submicron = 15% and supermicron = 0%, then I would clarify that only the periods of hydrophilic FAA much higher than the mean values would be deemed by Barbaro el al. (2015) as indicative of sea spray aerosol.

We agree with the reviewer's comment and have added the time constraints of the 'high levels of hydrophilic FAAs' during the campaign as follows (page 14, line 23/24): "The relatively high content of hydrophilic FAAs during certain periods of the campaign points at least at some influence of local oceanic sources."

R#1-13)
3.2.2 Size-segregated aerosol particles at the MV
p. 15
The following statements do not agree:
"The submicron aerosol particles at the MV had an averaged ΣFAA concentration of 1.5 ng m-3 (0.8-1.9 ng m-3) and were about three times lower compared to the ΣFAA concentration at the CVAO." (line 13)
"the aerosol particles measured at ground level represented the aerosol particles at

cloud level" (line 18)
I am guessing that the implication being made here is that it is the same particle
population sampled at CVAO and MV but that aerosol ageing or processing is happening
between the two sites and causing the significant change in FAA concentration. This
is a guess because it is not stated. What are the wind speeds and transport times
between the two sites?

We thank the reviewer for his comment and agree that the differences between MV and CVAO and at the same time their representativeness was not expressed clearly enough. The physical measurements (PSND) showed that there was good mixing of the aerosol particles (page 15, line 13-17 and please see also comment R#1-14). As described in the manuscript, during sampling periods of aerosol particles at the MV station clouds were present, which may have affected the amino acid concentrations and composition at the MV. In addition, as the reviewer rightly mentioned that aging of the aerosol particles on their way from the CVAO to the MV could lead to significant change in FAA concentration. Which of these effects mainly explains the difference in CVAO and MV amino acid composition remains speculative.
To calculate the transport times between two sides the modelled updraft of the vertical winds during the campaign has to be considered (average of ~ 5 cm s$^{-1}$, Figure 23 in van Pinxteren et al. (2020)). Taking into account the height of the Monte Verde (~ 800 m), a transport time of about 4 hours could be calculated. It should be noted that in the considered trade wind region on the Cape Verde Islands the vertical updraft plays a major role in the non-orographic cloud formation, which takes place on the Mt. Verde.
These aspects were discussed and the aging aspects was included in the revised manuscript as follows (page 15, line 18-23): "The concentration and composition of the aerosol particles can therefore be affected by the clouds that formed and disappeared consistently during the sampling period of the aerosol particles at the Mt. Monte Verde (for further details on the frequency of the cloud events see Gong et al. (2020) and van Pinxteren et al. (2020)). Furthermore, ageing processes may occur during the upwind of the aerosol particles from the CVAO to the MV station, which takes about 4 h considering an average vertical wind of 5 cm s$^{-1}$ (van Pinxteren et al. 2020)."

R#1-14)
p. 15 line 18
"the aerosol particles measured at ground level represented the aerosol particles at
cloud level, i.e. the aerosol particles within the marine boundary layer were well
mixed."
There was no aerosol sampling at ground level, correct? CVAO sampling ocurred at 30
m and MV at 744 m, right?
The argument made in the methods section is vague but I think it is that 30 m (CVAO)
and 744 m (MV) are in the same layer of the atmoshpere. Its not clear if this is
within the internal boundary layer, or above it, within the marine boundary layer,
or otherwise. Please clarify, including marine boundary layer vs. internal boundary
layer.

The aerosol particle sampler at the CVAO was installed in a height of 30 m (CVAO tower) and lies within the internal boundary layer (IBL) as stated in the manuscript on page 4, line 33 – page 5, line 2. The height of the marine boundary layer (MBL) was determined by modeling (COSMO-MUSCAT) and by helikite measurements during the campaign. The results of the MBL are discussed in detail in Figure 5 of the MarParCloud campaign overview paper (van Pinxteren et al., 2020) and shows that the Mt. Verde was predominantly in MBL for most of the time during the period considered in this study. In order to include this additional information in the MS, we have reworded the sentence as follows (page 15, line 14-17): "This indicated that, for cloud-free conditions, the aerosol particles measured at ground level (30 m) within the IBL, which is mainly below 30 m (Niedermeier et al., 2014), represented

the aerosol particles at cloud level. Thus, the aerosol particles within the marine boundary layer (MBL) were well mixed and the Mt. Verde was most of the time within the (MBL) (van Pinxteren et al., 2020)."

R#1-15) p. 15 line 20
"The concentration and composition of the aerosol particles can therefore be affected by the clouds that formed and disappeared consistently during the sampling period of the aerosol particles at the Mt. Monte Verde"
I like that analytes detected in cloud water are reported as a concentration per volume of air, instead of per volume of cloud water (LWC), thus providing the aerosol content of the air mass, and being unaffected by varying levels of liquid water content.

We thank the reviewer for his comment. Yes, in fact, for presenting the cloud water concentrations unaffected by varying LWC we provided the calculation of the analyte concentration in cloud water in concentration per volume of air (see also R#1-17).

R#1-16) p. 15 line ~25
I wonder if it would be better to not discuss cloud processing until after the data from the cloud water have been presented.

The cloud water data and its discussion are summarized in section '3.3 cloud water samples'. However, we believe it is helpful to address possible cloud effects already in section 3.2.2 within the discussion of the aerosol concentration and composition of the MV sampling station. In order to provide a possible explanation for the significant change in FAA at the MV (compared to the CVAO). Therefore, we would prefer to keep this short explanation (page 15, line 18-214): "The concentration and composition of the aerosol particles can therefore be affected by the clouds that formed and disappeared consistently during the sampling period of the aerosol particles at the Mt. Monte Verde (for further details on the frequency of the cloud events see Gong et al. (2020) and van Pinxteren et al. (2020))."

R#1-17) 3.3 Cloud water samples
p. 16 line 3.
"The individual atmospheric concentration of FAA in cloud water was calculated based on the measured liquid water content (LWC)"
OK. Very good. [concentration in cloud water] x [cloud water volume] / [air volume sampled]

We thank the reviewer for this positive comment. Using this calculation, we were able to present the analyte cloud water concentrations unaffected by varying LWC.

R#1-18) p. 17 line 13
"The reason for the high concentrations of FAA in cloud water (compared to the oceanic and aerosol particle concentrations) remain speculative to date and will be subject of further studies."
Probably the most significant finding of this study is not pursued.

We agree that the high FAA concentrations in cloud water is a very important finding of this study. We tried to emphasize this at several points in the discussion (e.g. section 3.3, 3.4.1, 3.4.2) as well as in the Abstract and in the Conclusion. To underline this finding, we speculated about the amino acid GABA in

context to microbial sources in the cloud water (e.g. page 20, line 4-7). However, we think that additional studies are necessary to explain the higher concentrations of FAA in cloud water in the marine environment. We believe that this finding alone is worth presenting and noticed that it is already referred to in a new study of Jaber et al. (2020).

R#1-19) p. 17 line 14
"Altogether, the in-situ formation of FAA in cloud water by chemical processes in
the cloud or by atmospheric biogenic formation or enzymatic degradation of proteins,
as proposed by Malfatti et al. (2019), as well as by selective enrichment processes
and pH dependent chemical reactions might be potential sources."
I am very surprised by the term "in-stu formation of FAA in cloud water" here. Are
you (suddenly) claiming that any or all of the amino acids detected in cloud water
were created in the atmosphere? None of them existed in the particles when the
particles were produced?
I fear a misundertanding of aerosol basics here, unless this is just a language
issue. Sea spray aerosol contains subsamples of seawater constituents that are small
enough to become aerosolized; this includes amino acids. Enzymes do not create amino
acids; they consist of amino acids and aminopeptidase enzymes cleave (or catalyse
the hydrolysis) amino acids from larger molecules. All cloud droplets form on
aerosols, thus, from the time they are formed, cloud droplets contain more than just
water. If the water from the cloud droplets evaporate, the cloud nuclei remain. Rain
will remove the cloud nuclei and other particles below. Was there any rain in the
area during or just prior to the sampling periods?
To consider atmospheric processing or ageing, I wanted to know the transport time
between CVAO and MV. The downwind distance between them is about 2 km. Wind speed on
average was about 5.5 m/s. So the average transport time would be about 6 minutes.

We agree with the reviewer that sea spray aerosol contains subsamples of seawater constituents that are small enough to become aerosolized and that this includes amino acids. However, as mentioned in our answer to the comment R#1-2, we believe that amino acid concentration and composition can be modified in the atmosphere, likely due to microbial processes. In this context, Jaber et al. (2020) investigated the abiotic and biotic formation of amino acids in cloud water and we wanted to address these interesting formation processes in this context. We do not assume that all amino acids are formed only in cloud water, but we think that this is an additional possible formation pathway of amino acids that is worth mentioning. This sentence now reads as follows (page 17, line 14-17): "Altogether, the in-situ formation of FAA in cloud water by chemical abiotic processes in the cloud or by atmospheric biogenic formation, as proposed by Jaber Jaber et al. (2020), as well as by selective enrichment processes and pH-dependent chemical reactions might be potential additional sources besides aerosol particles."

Rain was absent during the entire campaign and this was added in the revised manuscript on page 15, line 21: "There was also no rain during the entire campaign".
Considering the transportation times of the aerosol particles between the CVAO and the MV sampling station, a transport time of about 4 hours could be calculated and the consideration that aging processes might occur during this time were included in the revised manuscript (see our answer to comment R#1-13).

R#1-20) 3.4.1. Hydrophilic amino acids
p. 18 line 15
"The hydrophilic amino acids (Asp, Glu, GABA) comprised a significant fraction in

the ULW and the SML, as well as in the (submicron) aerosol particles and in cloud water (Fig. 4a-d). They were not detected in the supermicron aerosol particles."
This is relevant to a major theme in sea spray aerosol research: the organic fraction of SSA across SSA sizes. The research would be more valuable if this were discussed.

In addition to the comparison of the FAA composition between submicron and supermicron aerosol particles in section 3.2.1, where the focus is on the large variety of amino acids in the submicron range and where it is mentioned that the supermicron aerosol particles at the CVAO consist exclusively of Gly, Ser and Ala, we have extended the following discussion regarding the composition of supermicron aerosol particles also in the revised manuscript in section 3.4.2. This now reads as follows (page 20, line 19-23):" It is remarkable that especially the aerosol particles in the larger size range (e.g. supermicron aerosol particles: B4, B5) at both smapling stations are less complex in amino acid composition and almost exclusively dominated by Gly, Ser and Ala (Fig. 4b, 4c). Gly is discussed in the literature as a photochemical degradation product of other existing amino acids and this comparatively more stable amino acid (Gly) thus becomes a major component of the FAA composition (Barbaro et al., 2015)."

R#1-21)
p. 18 line 22
"The abundance of GABA on the submicron aerosol particles suggests that (marine) microorganisms were present on the aerosol particles and likely produced GABA via microbiological decarboxylation of Glu."
So it is impossible for GABA to transfer from the ocean into the atmosphere in a nascent sea spray aerosol?
Which microorganisms would be small enough to be found in submicron aerosol? Small marine bacteria? These sizes are of the dried particles, so the particle sizes in the atmosphere were larger, and perhaps the aerosol population that were submicron after drying contained supermicron aerosols before drying and were big enough to contain marine bacteria.

Based on the ambient study performed here, we cannot totally exclude an oceanic transfer of GABA. However, GABA was not detectable in the seawater and this was not related to a detection problem. This means that within an oceanic transfer GABA would need to be transferred to a very large extend compared to other amino acids. Such an enhanced transfer was not observed for the other hydrophilic amino acids (Glu, Asp). The percentage compositions of Glu and Asp are not strongly different in the ocean and on the aerosol particles. Unless the oceanic transfer of GABA is different compared to the other hydrophilic amino acids, this pathway is not the most probably one. Together with the fact GABA is known indicator for microbiological decomposition of OM, we believe that the formation of GABA on the aerosol particles is a likely explanation (see also our answer to comment R#1-2).
However, as we cannot exclude the oceanic transfer from our ambient measurements, we therefore included this pathway as a possible source besides the microbiological formation of GABA on aerosol particles. It now reads as follows in the revised manuscript (page 18, line 23-30 and page 20, line 1-3): "The abundance of GABA on the submicron aerosol particles suggests that either GABA could have been produced by microbiological decarboxylation of Glu by present (marine) microorganisms on the aerosol particles, or that GABA was transferred from the seawater to the atmosphere. However, GABA could not be found in seawater (ULW and SML) and this is not related to the sensitivity of the analytical method. Hence, a very enhanced oceanic transfer of GABA would be needed to explain this finding. Such an enhanced transfer was, however, not observed for the other hydrophilic amino acids (Glu and Asp), their percentage composition was not strongly different regarding seawater and submicron aerosol particles at the CVAO. Unless the oceanic transfer of GABA is very different compared to other

hydrophilic amino acids, this pathhway does not explain the high abundance of GABA on the submicron aerosol aprticles at the CVAO. Together with the facts that GABA is a known indicator for the microbiological decomposition of OM (Dauwe et al., 1999;Engel et al., 2018), and microorganisms are known to be present on marine aerosol particles even in the submicron size range (Rastelli et al., 2017) the formation of GABA on the aerosol particles might be related to an in-situ formation."

Concerning the marine microorganisms on the aerosol particles we would like to refer to the comment R#1-2, in which we discussed in more detail the possibility that (intact) microorganisms may be present on (submicron) aerosol particles. However, we would like to point out that microorganisms were not examined more closely on the size-segregated aerosol particles considered here.

R#1-22)
p. 18 line 24
The connection to Malfatti et al. (2019) here is confusingly redundant.

We agree with the reviewer's comment and removed this connection to the study of Malfatti et al. (2019) in this context. We added this reference to "Active microbial enzymes on nascent sea spray aerosol have recently been reported by Malfatti et al. (2019)." (page 18, line 22/23)

R#1-23)
p. 19 Figure 4
Thank you for plotting the data together in this figure. It appears the cloud water and seawater have similar profiles. Are there others?

We thank the reviewer for his comment. The similarities between the seawater, the aerosol particles and the cloud water, which are visible in Figure 4, are discussed in section '3.4 Concerted measurements of FAA in the marine compartments (seawater, aerosol particles and cloud water)'. In this comparison, similarities and differences between the marine compartments are identified with focus on the amino acid groups (section 3.4.1, section 3.4.2, section 3.4.3).

R#1-24)
3.4.2 Neutral and hydrophobic amino acids
p. 20
Some very low lifetimes are reported for amino acids in the atmosphere: ~0.5-1 hour. Relate this to the sampling period lengths and the duration of time between sampling and analyses. Did the samples/analytes need to be isolated from air/atmosphere between collection and analyses? Should we question the amino acid residence times? Or are they accurate and is it possible that the amino acids detected are largely from intact microorganisms collected in the aerosol?

We want to emphasize that the calculation of the mean lifetime is based on the fact that all OH radicals in the atmosphere react exclusively with the amino acids, which is why we get these (short) mean lifetimes. These cannot be transferred 1:1 to the real atmosphere, as explicitly mentioned and discussed in the SI (SI: page 17, line 14-18). Our estimation of the mean lifetime of the amino acids on the aerosol particles in the different scenarios serves only to compare the different amino acids among each other and to work out similarities/differences, which are discussed in detail in section 3.4.1, 3.4.2 and 3.4.3 in the manuscript.

For the sample collection time (24 h), we have oriented to the collection times in previous studies. In these studies, collection times for amino acids on aerosol particles are reported as 1-4 days (Helin et al., 2017), 24 h (Mandalakis et al., 2010) or even up to 10 days (Barbaro et al., 2015). These studies included FAA analysis and did not mention any artifacts or biases from which we could conclude that

the technique of aerosol particle sampling we used here causes problems. We cannot completely exclude such problems during sampling, but the aerosol particle samples were frozen immediately after sampling and stored at -20 °C until analysis (as stated in the manuscript on page 5, line 11/12) to minimize possible further reactions or processing. Thus, we have followed common aerosol sampling techniques as well as times in the field.

R#1-25)
3.4.4 Transfer of amino acids from the ocean into the atmosphere
p. 21 line 7
"The high similarity concerning the main FAA species within the different
compartments, together with the high concentration of ocean-derived compounds (Na+,
MSA) in the aerosol particles and cloud waters, suggest a coupling between the FAA
in the ocean and the atmosphere."
I don't think this "high similarity" was established. The GABA in submicron CVAO is
a high dissimilarity with SML and ULW.
It would be useful to have FAA profiles of dust (or other aerosol types) for
comparison. If those profiles are very different, then perhaps the profiles
presented in this work would appear more similar.

We agree with the reviewer's comment and have rephrased this sentence, highlighting differences regarding GABA. It now reads as follows (page 21, line 20-22): "A high similarity regarding the FAA species within the different marine compartments could be observed, although some differences could also be identified (e.g. GABA). Together with the high concentration of ocean-derived compounds (Na$^+$, MSA) in the aerosol particles and cloud water, this indicates a coupling between the FAA in the ocean and the atmosphere."
In our study we did not perform the analysis of an FAA dust profile and we could not find any comparable literature on that subject. Furthermore, to our knowledge, no single (free) amino acid has ever been attributed as a dust tracer. We found only one study by Armstrong et al. (2001) on the analysis of combined amino acids (hydrolyzed proteins and their building blocks "amino acids"). However, this study is not suitable for a profile for FAA in dust as a comparison.

R#1-26) p. 23 line 8
"FAAs were present in the size range for aerosol particles associated with CCN
activity and cloud water, and might be connected to CCN activity due to their
hygroscopicity and soluble character, but this effect was not investigated here."
It is disappointing to read this. The authors found something very interesting: very
high atmospheric concentrations of amino acids in cloud water, relative to the
ambient aerosol population. But that finding is ignored.

The focus of this study was to investigate the individual free amino acids in the different marine compartments (seawater, aerosol particles and cloud water).
Here we have certainly addressed and not ignored the high FAA concentrations in cloud water as an important finding of this study (see also review comment R#1-18 and in the manuscript: Abstract, section '3.3 Cloud water samples' and the Conclusion). We also find the fact that the amino acids were present in the size range for aerosol particles associated with CCN activity interesting, but we think that this is a separate topic for future studies as we stated on page 23, line 28/29 "…., but this effect was not investigated here and should be examined in future studies."

R#1-27)
Supplemental Information: acp-2019-976-supplement-version3.pdf

p. 16 Back Trajectory
It is great that you ran back trajectories for all your sampling periods. Having done so, please share them all. Why not share them all when they support the science?

The data and illustrations of the backward trajectories of the entire MarParCloud campaign have already been published in the study by van Pinxteren et al. (2020). In order to share this information with the scientific community, we have explicitly referred to this overview paper of the MarParCloud campaign as follows (SI: page 16, line 12/13): "An overview of backward trajectories during the entire campaign period can be found in the study by van Pinxteren et al., (2020)."
Reference to this study of van Pinxteren et al. (2020) for further information on air mass classification is also given in the manuscript (page 11, line 29/30): "Further information on the classification of the air masses and distinct concentrations of dust tracers are given in the overview paper of this campaign (van Pinxteren et al., 2020)."

**Additional changes performed by the authors**

We corrected a typo in the revised manuscript on page 4, line 25: "area of 2500 $cm^2$"

[revised manuscript text omitted]